



# Aerosol activation characteristics and prediction at the
# central European ACTRIS research station Melpitz,
# Germany
Yuan Wang[1,2,3*], Silvia Henning[1*], Laurent Poulain[1], Chunsong Lu[2], Frank
Stratmann[1], Yuying Wang[2], Shengjie Niu[2,4], Mira L. Pöhlker[1], Hartmut Herrmann[1],
and Alfred Wiedensohler[1]
1. Leibniz Institute for Tropospheric Research (TROPOS), 04318 Leipzig, Germany.
2. Collaborative Innovation Center on Forecast and Evaluation of Meteorological Disasters,
Nanjing University of Information Science and Technology, 210044 Nanjing, China.
3. Collaborative Innovation Center for Western Ecological Safety, Lanzhou University, 730000
Lanzhou, China.
4. College of Safety Science and Engineering, Nanjing Tech University, 210009 Nanjing, China.
*Correspondence: Yuan Wang (wang_yuan@lzu.edu.cn) and Silvia Henning (henning@tropos.de)
**Abstract:** Understanding aerosol particle activation is essential for evaluating aerosol
indirect effects (AIEs) on climate. Long-term measurements on aerosol particle
activation help to understand the AIEs and narrow down the uncertainties of AIEs
simulation; however, they are still scarce. In this study, more than 4-year aerosol
comprehensive measurements were utilized at the central European research station
Melpitz, Germany, to gain insight into the aerosol particle activation and provide
recommendations on improving prediction. The overall characteristics of aerosol
particle activation at Melpitz are first summarized. For supersaturation ($SS$) levels of
0.1%, 0.2%, 0.3%, 0.5%, and 0.7%, the mean cloud condensation nuclei (CCN) number
concentration ($N_{CCN}$) increases with the increase of $SS$ from 513 to 2477 cm$^{-3}$, which
represents 11% to 52% of the total particle number concentration with diameter ranging
from 10 to 800 nm, while the hygroscopicity factor ($\kappa$) and the critical diameter ($D_c$)
decrease from 0.28±0.08 (mean value ± one standard deviation) to 0.20±0.09 and



from 177$\pm$19 to 54$\pm$8 nm, respectively. Aerosol particle activation is highly variable
across seasons, especially at low $SS$ conditions. At $SS = 0.1\%$, the seasonal mean $N_{CCN}$
is 681 cm$^{-3}$ in winter, which is almost twice higher than the summer value (347 cm$^{-3}$);
the seasonal mean activation ratio (AR) in winter (0.18) is three times higher than the
summer one. Subsequently, size dependency of both $\kappa$ and the state of mixing were
investigated. As the particle diameter ($D_p$) increases, $\kappa$ increases at $D_p$ of ~40 to 100 nm
and almost stays constant at $D_p$ of 100 to 200 nm, whereas the degree of the external
mixture keeps decreasing at $D_p$ of ~40 to 200 nm. The relationships of $\kappa$ vs. $D_p$ and
mixture degree vs. $D_p$ were both fitted well by the power-law function for each season.
Finally, we recommend applying the $\kappa$ - $D_p$ power-law fit for $N_{CCN}$ prediction, which
can narrow down the median uncertainty within 10% for different $SS$ conditions and
seasons at Melpitz; it also could be applied to predict $N_{CCN}$ at other rural and continental
regions with a similar aerosol background. Additionally, the mean $\kappa$ value over $D_p$ of
100 to 200 nm also works well on the $N_{CCN}$ prediction when $SS$ is less than 0.2%.

## 1. Introduction

The specific subset of aerosol particles that serves as nuclei for the condensation
of water vapor, forming activated cloud droplets at a given supersaturation ($SS$)
condition, is known as cloud condensation nuclei (CCN). Aerosol particle activation
affects the aerosol and cloud interactions (ACI), thereby changing the cloud
microstructure (Zhao et al., 2012; Lu et al., 2013; Jia et al., 2019; Wang et al., 2019),
precipitation (Khain, 2009; Wang et al., 2011; Fan et al., 2012, 2018), radiation
(Twomey, 1974, 1977; Albrecht, 1989; Zhao and Garrett, 2015), and by these effects
the global climate (Ramanathan et al., 2001; Wang et al., 2014; Rosenfeld et al., 2019).
The latest sixth assessment report from IPCC (2021) pointed out that aerosol indirect



effects (AIEs) remain the most considerable uncertainty in assessing the anthropogenic
contribution to present and future climate change.

The ambient *SS* and aerosol activation ability are both important for predicting the

number concentration of activated droplets. The classical Köhler theory (Köhler, 1936),
combining the Raoult law with the Kelvin effect, illustrates that the aerosol particle
activation depends on particle size, chemical composition and the given *SS*. Petters and
Kreidenweis (2007) utilized a single hygroscopicity factor $\kappa$ to describe the CCN
activity at each particle diameter ($D_p$), which facilitates studying the activation process
without considering the complex chemical compositions of aerosol particles
(McFiggans et al., 2006).

Different perspectives have been presented on the influence of particle size and

composition on the CCN activation. In terms of a single aerosol particle, the actual
particle size plays a more important role than the chemical composition for activation
because of the reciprocal relationship between $\kappa$ and $D_p{}^3$ at a given *SS*. As for a
population of aerosol particles, Dusek et al. (2006) concluded that particle number size
distribution (PNSD) matters more than the chemical composition distribution, which
has been supported by many experiments. Even sometimes, assuming a constant $\kappa$ still
predicted CCN number concentration ($N_{CCN}$) well (e.g., Sihto et al., 2011; Wang et al.,
2018a). Andreae and Rosenfeld (2008) reviewed the previous studies on aerosol particle
activation and recommended that for modeling purposes, the global $\kappa$ values of 0.3±0.1
and 0.7±0.2 can be representative for continental and marine aerosol, respectively,
which has been widely used to predict $N_{CCN}$. The regional variability should be
underlined because the mean $\kappa$ measured in urban, rural, and forest exhibits significant
differences. For instance, Sihto et al. (2011) suggested an average $\kappa$ of 0.18 to predict
the CCN activation well in boreal forest conditions in Hyytiälä, Finland; a fixed $\kappa$ of



0.31 suffices to calculate the $N_{CCN}$ in a suburban site located in the center of the North
China Plain (Wang et al., 2018a); the mean $\kappa$ is 0.5 in a near-coast and rural background
station (CESAR Tower) in Netherlands (Schmale et al., 2018); the median $\kappa$ ranges
from 0.02 to 0.16 at $SS = 0.1-1.0\%$ in an urban background site in Budapest, Hungary
(Salma et al., 2021). Therefore, the assumption of a constant $\kappa = 0.3$ may not be
appropriate when trying to predict $N_{CCN}$ for different continental regions.

Additionally, some experiments, especially conducted on more diverse particulate

sources, have indicated chemistry does play an important role in $N_{CCN}$ variability (e.g.,
Nenes et al., 2002; Petters and Kreidenweis, 2007; Rose et al., 2010). Not only the bulk
chemical composition with a constant $\kappa$ should be considered for $N_{CCN}$ prediction, but
the size-resolved chemical composition (Deng et al., 2011, 2013; Wu et al., 2016) and
the mixing state should be applied (Su et al., 2010; Zhang et al., 2014). Information on
the organic aerosol fraction improves $N_{CCN}$ prediction considerably (Poulain et al., 2010;
Zhang et al., 2016; Kuang et al., 2020). Freshly formed particles are about 1 nm in
diameter (Kulmala et al., 2012); they must grow to tens of nanometers in diameter to
serve as the effective CCN at a relatively high $SS$ of ~1% (Dusek et al., 2006) and even
larger than 200 nm to be efficient at $SS$ less than 0.1% (Deng et al., 2013). Aerosol
chemical composition changes during the growing and aging processes. For instance, $\kappa$
increases with particle size caused by photochemical processes which enhance
secondary inorganic species formation and go along with an increase in particle size
(Massling et al., 2009; Zhang et al., 2017; Wang et al., 2018b). On the other hand, in
sulfate dominated new particle formation (NPF) events with subsequent particle growth
by condensation of organic vapors, the $\kappa$ of small particles may exceed the $\kappa$ of the
larger ones (Wang et al., 2018a). If the $\kappa$ of organic aerosol increases from 0.05 to 0.15,
the global average aerosol radiative forcing would decrease by $\sim 1\ W\ m^{-2}$, which is in


the same order of magnitude as the overall climate forcing of anthropogenic aerosol
during the industrialization period (Rastak et al., 2017).

To obtain the regional parameters of aerosol particle activation, extensive field

campaigns have been conducted worldwide. Besides the significant difference in spatial,
also the temporal variations of aerosol activation characteristics are essential for $N_{CCN}$
prediction (Andreae and Rosenfeld, 2008). Most of the observations lasted 1–2 months
or even less; they mainly focused on the effects of short-term weather processes or
pollution events on aerosol particle activation, such as the effects of the summer
monsoon (Jayachandran et al., 2017, 2020), wet removal (Croft et al., 2009), NPF
events (Dusek et al., 2010; Wu et al., 2015), biomass burning (Rose et al., 2010), and
aerosol particle aging as well as oxidation processes (Zhang et al., 2016, 2017). The
long-term CCN concentration measurements (of at least one full year) are still rarely
reported, resulting in insufficient knowledge concerning the seasonal and annual cycles
of aerosol particle activation, which are also critical for model predictions and
evaluations. Burkart et al. (2011) reported the particle activation in the urban
background aerosol of Vienna, Austria, based on 11-month aerosol and CCN
concentration measurements. Paramonov et al. (2015) reported a synthesis of CCN
measurements within the EUCAARI (European Integrated project on Aerosol Cloud
Climate and Air Quality interactions) network using the long-term data collected at 14
locations. Pöhlker et al. (2016) presented the climatology of CCN properties of a remote
central Amazonian rain forest site using 1-year measurements. Che et al. (2017)
provided the aerosol-activation properties in the Yangtze River Delta, China, based on
~1-year measurements. Using the long-term (of most > 1 year) aerosol and CCN
concentration measurements from 12 sites, Schmale et al. (2018) presented the spatial
differences in aerosol particle activation for various regional backgrounds. However,




systematic studies focusing on the seasonal cycle of size-resolved particle activation
and respective CCN predictions are still scarce in the central European continent. Such
a study would be of great help for understanding ACI and narrowing down the regional
uncertainties in climate predictions.
In this investigation, more than 4-year comprehensive measurements of aerosol
physical, chemical, and activation properties collected at the ACTRIS (Aerosol, Clouds
and Trace Gases Research Infrastructure, http://www.actris.eu/) site Melpitz, Germany,
are utilized. The major objective is to gain insight into the aerosol particle activation
and provide recommendations on methods for CCN predictions. We present therefore
the long-term observations and seasonal cycles of various particle activation variables
such as CCN number size distribution, $N_{CCN}$, activation ratio, critical diameter, size-
resolved $\kappa$ and mixing state degree. Furthermore, we evaluated the accuracy of $N_{CCN}$
calculated from five different activation schemes and finally provide recommendations
to use a power-law based parameterization for the dependence of $\kappa$ on particles diameter
for long-term $N_{CCN}$ prediction at Melpitz and for other regions with a similar aerosol
background condition.
**2. Methodology**
**2.1 Experiment details**
Atmospheric aerosol measurements were conducted at the Melpitz observatory
(51.54°N, 12.93°E, 86 m above sea level), 50 km to the northeast of Leipzig, Germany.
The aerosol particles observed at Melpitz can be regarded as representative for the
central European rural background conditions (Birmili et al., 2009). The surroundings
of the site are mostly pastures and forests without significant sources of anthropogenic
emissions. More detailed descriptions of the Melpitz site can be found in for example,
Poulain et al. (2020).





This study focuses on the physicochemical properties and the activation ability of
aerosol particles using the data collected at Melpitz from August 2012 to October 2016.
Figure 1 demonstrates the experimental setup. All instruments were in the same
container laboratory and utilized the same air inlet. Ambient aerosol particles were first
pretreated through a $PM_{10}$ Anderson inlet and an automatic aerosol diffusion dryer kept
the relative humidity in sampling lines at a relative humidity less than 40% (Tuch et al.,
2009). Subsequently, the aerosol flow was divided into the different instruments using
an isokinetic splitter. Particle number size distributions (PNSD) were measured using a
Dual-mobility particle size spectrometer (D-MPSS, TROPOS-type; Birmili et al., 1999;
Wiedensohler et al., 2012) within the diameter ranging from 5 to 800 nm. An aerosol
chemical species monitor (ACSM, Aerodyne Inc; Ng et al., 2011) was used to measure
the chemical compositions of near-$PM_1$ non-refractory submicron aerosol particles
(nitrate, sulfate, chloride, ammonium, and organics). A multi-angle absorption
photometer (MAAP, model 5012, Thermo Scientific; Petzold and Schönlinner, 2004)
was used to measure the particle light absorption coefficients and to estimate the
equivalent black carbon (eBC) mass concentration. For simultaneous measurement of
particle and CCN number size distributions, dried aerosol particles were passed through
the bipolar charger to establish charge equilibrium (Wiedensohler, 1988) and then
through a differential mobility analyzer (DMA) for selecting a monodisperse particle
fraction; after the DMA the flow was divided into two parts, respectively passed through
a condensation particle counter (CPC, model 3010, TSI) to measure the total number
concentration of the selected monodisperse condensation nuclei ($N_{CN}$) and through a
cloud condensation nuclei counter (CCNC, model 100, Droplet Measurement
Technologies; Roberts and Nenes, 2005) to measure the $N_{CCN}$. Thus, the size dependent
activated fraction (AF, $N_{CCN}/N_{CN}$) curve, i.e., the AF at a certain diameter ($D_p$) of dry



particles, could be obtained. A total of five different *SS* conditions was set in the CCNC
instrument (0.1%, 0.2%, 0.3%, 0.5%, and 0.7%). A complete *SS* cycle lasted ~2.5 hours.

All the instrumentation was frequently calibrated within the framework of the

European Center for Aerosol Calibration (ECAC, https://www.actris-ecac.eu/). The
ACSM was regularly calibrated according to the manufacturer's recommendations with
350 nm monodispersed ammonium nitrate and ammonium sulfate particles (Freney et
al., 2019; Poulain et al., 2020). The D-MPSS was calibrated following the
recommendations in Wiedensohler et al. (2018). Throughout the campaign, the CCNC
was regularly calibrated following the procedures outlined in Rose et al. (2008). The
measurement uncertainties of these instruments should be noted. The uncertainty in the
MAAP is within 10% (Müller et al., 2011), and those in the D-MPSS and CCNC are
both on the order of 10% (Wiedensohler et al., 2018; Rose et al., 2008). For the *SS*
setting in CCNC, Gysel and Stratmann (2013) pointed out that an achievable accuracy
in *SS* is 10 % (relative) at *SS* > 0.2%, and less than 0.02 % (absolute) at the lower *SS*.
For the ACSM data, the uncertainty in determining the total non-refractory mass is 9%;
while for the individual chemical components, it is 15% for nitrate, 28% for sulfate, 36%
for ammonium, and 19% for organic matter (Crenn et al., 2015).

Due to instrument failures and maintenance operations, missing measurements

occurred during the campaign. Effective data coverage is shown in Figure 2. Overall,
the CCNC, D-MPSS, and ACSM-MAAP captured 45578 AF curves, 103052 PNSDs,
and 26876-hour aerosol chemical measurements, which covered 63%, 92%, and 77%
of the campaign time, respectively. For 42% of the time all these instruments were
measuring together.
**2.2 Methods**

Each AF curve ($N_{CCN}/N_{CN}$ vs. $D_p$) was firstly corrected for multiply charged





particles. Multiply (mostly doubly) charged particles appear in the AF curve as a plateau
or shoulder at small diameters because they have the same electrical mobility diameter
as singly charged smaller particles; thus, they are falsely selected in the DMA (Rose et
al., 2008; Henning et al., 2014). For this was corrected by subtracting the multiply
charged particle fraction as determined from the D-MPSS measurements from each
value of $N_{CCN}/N_{CN}$ in AF. The PNSD from the D-MPSS measurements (5 to 800 nm)
are needed as the DMA-CCNC size range does not cover the large particle fraction,
which is essential for the correction. Subsequently, we obtained the corrected AF curves.
Each corrected AF curve was fitted with a sigmoid function,

$$AF = a + b \Big/ \left(1 + \exp\left(-\frac{D_p - D_c}{\sigma_s}\right)\right) \tag{1}$$

where $a$ and $b$ are the lower and upper limits for calculating critical diameters ($D_c$) at
the set-nominal $SS$, and $\sigma_s$ is a measure for the width of the sigmoid function. This AF
fit was multiplied with the PNSD to gain the CCN number size distribution and by
integrating the total number of CCN, i.e., $N_{CCN}$.
The critical diameter ($D_c$) of dry particles, $\kappa$, and mixing state at each $SS$ condition
can be derived from the AF fit results. Affected by aerosol mixing, the AF rises
gradually from 0 to the max (~1) rather than an intermittent mutation. $D_c$ is defined as
the diameter of the dry particles from which 50% of the particles are activated at the
given $SS$.
The shape of the AF curve, i.e., the relative width of the AF, represents the degree
of external mixture, which can be quantified by the ratio of $(D_{75} - D_{25})/D_c$ (Jurányi et
al., 2013). $D_{75}$ and $D_{25}$ are the diameters at which 75% and 25% of the particles are
activated at the given $SS$. Internal mixture implies that all particles with equal dry size
have equal $\kappa$ with $(D_{75} - D_{25})/D_c = 0$, whereas a distribution of different $\kappa$ can be





observed for externally mixed aerosol with higher $(D_{75} - D_{25})/D_c$ values. Jurányi et al.
(2013) confirmed the reliability of this approach by comparing the $\kappa$ distributions
derived from parallel monodisperse CCN measurements and HTDMA measurements.
According to the derivation of $\kappa$-Köhler theory (Petters and Kreidenweis, 2007),
the $\kappa$ can be calculated from $D_c$ at a given $SS$:

$$\kappa = \frac{4A^3}{27{D_c}^3 \ln^2(1 + SS/100)} \qquad (2a)$$

with

$$A = \frac{4\sigma_{s/a} M_w}{RT\rho_w} \qquad (2b)$$

where $\sigma_{s/a}$ is the droplet surface tension (assumed to be that of pure water, $0.0728\,\mathrm{Nm}^{-2}$),
$M_w$ the molecular weight of water, $R$ the universal gas constant, $T$ the absolute
temperature, $\rho_w$ the density of water, and $A$ can be considered a function of $T$. Thus, the
size-resolved $\kappa$ can be obtained at each $SS$ cycle.
Besides deriving it from the monodisperse CCN measurements, $\kappa$ can be
determined from the ACSM chemical composition measurements ($\kappa_{chem}$) using the
Zdanovskii–Stokes–Robinson (ZSR) mixing rule (Zdanovskii, 1948; Stokes and
Robinson, 1966) combined with $\kappa$-Köhler theory:

$$\kappa_{chem} = \sum_i \varepsilon_i \kappa_i \qquad (3)$$

where $\kappa_i$ and $\varepsilon_i$ mean the $\kappa$ and volume fraction for each component, respectively, and
$i$ is the number of the component in the mixture. The $\varepsilon_i$ was derived from its measured
component $i$ mass concentration and density ($\rho_i$). A simple ion-pairing scheme (Gysel
et al., 2007) was used in this study with the $\kappa_i$ and $\rho_i$ values listed in Table 1 (Wu et al.,
2015). Note that a $\kappa$ of 0.1 is used for particulate organics (Dusek et al., 2010; Gunthe


et al., 2009, 2011); for black carbon, we use a $\kappa$ of 0 (Rose et al., 2011; Schmale et al.,

2018).

When $\kappa$ is given, we can predict the $N_{CCN}$ at each $SS$. Thereto, $D_c(\kappa, SS)$ is
calculated from equation 2a. And, assuming an internal mixture, the predicted $N_{CCN}$ is
the integration of the PNSD from $D_c$, that is,

$$Predicted\ N_{CCN} = \int_{D_c}^{800} PNSD(D_p)dD_p \qquad (4)$$

## 248    3. Results

### 249    3.1 Aerosol activation characteristics

Figure 3a presents the time series of the mean CCN number size distribution at
each $SS$ condition. As $SS$ increases, CCN number size distribution broadens towards
smaller particle sizes, causing an increase in $N_{CCN}$ and activation ratio (AR, i.e., ratio
of $N_{CCN}$ to total aerosol number concentration with diameter ranging from 10 to 800 nm,
$N_{aero}$). At Melpitz, the mean $N_{CCN}$ is 513, 1102, 1466, 2020, and 2477 cm$^{-3}$ at $SS$ of 0.1%,
0.2%, 0.3%, 0.5%, and 0.7%, respectively. The mean AR ranged from 0.11 to 0.52 at
$SS$ = 0.1% to 0.7%. As shown in Table 2, the mean $N_{CCN}$ measured at Melpitz is
generally higher than that measured in more remote rural background stations. For
instance, as $SS$ increased from 0.1% to 1.0%, the mean $N_{CCN}$ increased from 362 to 1795
cm$^{-3}$ in Vavihill, Sweden (Fors et al., 2011) and 274 to 1128 cm$^{-3}$ in Hyytiälä, Finland
(Paramonov et al., 2015); in Southern Great Plains, USA, the mean $N_{CCN}$ at $SS$ = 0.4%
was 1248 cm$^{-3}$ (Liu and Li, 2014); the mean $N_{CCN}$ increased from 118 to 1826 cm$^{-3}$ as
$SS$ increased from 0.1% to 0.94% in Mahabaleshwar, India (Singla et al., 2017).
However, the mean $N_{CCN}$ measured at Melpitz is far lower than that measured in
polluted regions. For example, in a rural site of Guangzhou, China, the mean $N_{CCN}$
increased from 995 to 10731 cm$^{-3}$ as $SS$ increased from 0.068% to 0.67% (Rose et al.,





2010); higher $N_{CCN}$ was observed in Wuqing, China, with the mean $N_{CCN}$ of 2192–12963
cm$^{-3}$ at $SS$ = 0.056–0.7% (Deng et al., 2011); in an urban site of Seoul, Korea, the mean
$N_{CCN}$ increased from 4145 to 6067 cm$^{-3}$ as $SS$ increased from 0.4% to 0.8% (Kim et al.,
2014); in a polluted continental site of Mahabubnagar, India, the mean $N_{CCN}$ at $SS$ = 1.0%
was ~5400 cm$^{-3}$ (Varghese et al., 2016).

At Melpitz, aerosol activation characteristics are highly variable across seasons.

At $SS$ = 0.1%, CCN number size distribution is wider in spring and winter than in
summer and autumn; the mean $N_{CCN}$ at $SS$ = 0.1% is 585, 347, 440, and 681 cm$^{-3}$ in
spring, summer, autumn, and winter, respectively. The mean $N_{CCN}$ at $SS$ = 0.1% in
winter is almost twice as high as that found in summer. The highest mean AR at $SS$ =
0.1% was 0.18 observed in winter, whereas the lowest mean AR (0.06) was observed
in summer. In spring and autumn, the mean AR at $SS$ = 0.1% is 0.1. As $SS$ increases,
CCN number size distribution gradually peaks in summer, especially at $SS$ = 0.5% and
0.7%. At $SS$ = 0.7%, the mean $N_{CCN}$ is 2622, 2530, 2222, and 2495 cm$^{-3}$, and the mean
AR is 0.49, 0.41, 0.51, and 0.68 in spring, summer, autumn, and winter, respectively.

The AR-$SS$ and $N_{CCN}$-$SS$ relationships in each season and all datasets are shown in

Figures 3b and 3c. The two relationships are similar, and both can be fitted well by the
power-law function (Twomey, 1959) and the error function (Pöhlker et al., 2018). The
fit results are shown in Table 3. The error function fits the relationships better than the
power-law function because of more parameters. The power parameter in the power-
law function means the change rate of the controlled variable with the independent
variable, that is the slope in a log-log coordinate system, so it is also called the slope
parameter. In the power-law fits of the two relationships, the slope parameters are
highest in summer and lowest in winter. Therefore, AR and $N_{CCN}$ are most sensitive to
$SS$ in summer, whereas the opposite is true in winter. The coefficients in the power-law



fits represent the AR and $N_{CCN}$ at $SS$ = 1%. The coefficient in AR-$SS$ fit is highest in
winter (0.89) and lowest in summer (0.61). However, the coefficient in $N_{CCN}$-$SS$ fit is
highest in summer (3951 cm$^{-3}$) and lowest in autumn (3136 cm$^{-3}$). Over the whole
period, the mean values of the slope parameter and the coefficient in the $N_{CCN}$-$SS$ power-
law fit are 3497 cm$^{-3}$ and 0.81, respectively, which are within the range of values for
continental aerosol (slope parameter of 600–3500 cm$^{-3}$ and coefficient of 0.4–0.9)
reported in Seinfeld and Pandis (2016).

CCN number size distribution is a part of the particle number size distribution

(PNSD), which approximately corresponds to the part of PNSD with $D_p > D_c$ when
assuming particles to be internally mixed. The schematic diagram in Appendix A shows
the relationship between the PNSD and the CCN number size distribution. Aerosol
chemical composition determines the $\kappa$ through equation 3, thereby changing $D_c$ at a
given $SS$ condition through equation 2a. Thus, we present the time series of the PNSD
and chemical compositions in Figure 4 to explain the variations in aerosol activation
characteristics.

In summer, affected by the frequent NPF events (Ma et al., 2015; Wang et al.,

2017), the Aitken-mode particles with $D_p$ < 100 nm account for the largest portion of
the PNSD, resulting in the highest $N_{aero}$ with a mean value of 6224 cm$^{-3}$ and the smallest
geometric mean diameter ($GMD = \exp\left(\frac{\Sigma_i n_i \times lnD_i}{N_{aero}}\right)$) with a mean value of 50 nm among
the four seasons. On the contrary, in winter, the mean $GMD$ increases to 58 nm, which
is the largest among the four seasons, and the $N_{aero}$ decreases to the lowest with a mean
value of 3686 cm$^{-3}$ because of the rare NPF events. During the NPF events, only a part
of newly formed particles grows to sizes larger than $D_c$ (e.g., ~55 nm at $SS$ = 0.7%),
whereas most of the new particles are still unactivated at $SS \leq$ 0.5%. Therefore, CCN
number size distribution gradually peaks as $SS$ increases in summer, whereas AR keeps



a minimum even at relatively high *SS* conditions as shown in Figure 3a. In winter, the
lowest $N_{aero}$ and the largest *GMD* contribute to the highest AR at each *SS* condition.

Figure 4b shows the average changes of the aerosol particle chemical compositions

over a year and the estimated bulk $\kappa_{chem}$ of submicron aerosol particles. At Melpitz, the
mean value of bulk $\kappa_{chem}$ is 0.36 with one standard deviation of 0.09 over the whole
period; the seasonal mean $\kappa_{chem}$ plus/minus one standard deviation are $0.38\pm0.09$,
$0.29\pm0.08$, $0.36\pm0.08$, and $0.40\pm0.08$ in spring, summer, autumn, and winter,
respectively. Because the $\kappa_{chem}$ depends on aerosol particle chemical composition
through equation 3, we examined the correlation between $\kappa_{chem}$ and the mass fraction
of each component to explain the variations of $\kappa_{chem}$. As shown in Figures 5a and 5b, a
negative correlation between the $\kappa_{chem}$ and the organic mass fraction ($f_{org}$) was observed,
while an opposite trend was found for the nitrate ($f_{nitrate}$). Additionally, the $\kappa_{chem}$ is not
correlated with the sulfate mass fraction ($f_{sulfate}$) and the BC mass fraction ($f_{BC}$), as
shown in Figures 5c and 5d.

In summer, there is the lowest bulk $\kappa_{chem}$ with $0.29\pm0.08$ corresponding to the

highest $f_{org}$ (56% of total mass on average), which could be related to the strong
formation of the secondary organic aerosol. In winter, low temperatures favor the
particulate phase of nitrate (Poulain et al., 2011) with a mean $f_{nitrate}$ of 31%, which might
explain the highest $\kappa_{chem}$ ($0.40\pm0.08$). According to equation 2a, $D_c$ increases as
$\kappa$ decreases at a given *SS* condition. Thus, the lowest $\kappa_{chem}$ results in the narrowest CCN
number size distribution and a decrease in $N_{CCN}$ in summer, especially at relatively low
*SS* conditions (e.g., 0.1% and 0.2%) as shown in Figure 3a.
**3.2 Size-resolved particle hygroscopicity factor and mixing state**

The hygroscopicity factor and the mixing state directly influence the $D_c$ and the



shape of the AF curve, thereby changing the $N_{CCN}$ at a given $SS$ condition. These two
parameters are not constant and both vary with particle size and season.

Figure 6a presents monthly averages of $\kappa$ calculated from monodisperse CCN

measurements ($\kappa_{CCN}$) at each $SS$ condition, and their seasonal mean values are
summarized in Table 4. At Melpitz, the mean $\kappa_{CCN}$ plus/minus one standard deviation
over all datasets are 0.28±0.08, 0.28±0.10, 0.24±0.10, 0.21±0.09, and 0.20±0.09 at
$SS$ = 0.1%, 0.2%, 0.3%, 0.5%, and 0.7%, respectively, where the mean $\kappa_{CCN}$ were all
less than the mean bulk $\kappa_{chem}$ of 0.36. The seasonal variation of $\kappa_{CCN}$ at each $SS$
condition is similar to that of $\kappa_{chem}$. In summer, $\kappa_{CCN}$ is lowest among the four seasons,
with mean values of 0.23, 0.25, 0.21, 0.19, and 0.19 at $SS$ = 0.1%, 0.2%, 0.3%, 0.5%,
and 0.7%, respectively. The highest $\kappa_{CCN}$ at each $SS$ condition was observed in winter,
with mean values of 0.32, 0.32, 0.28, 0.23, and 0.21 at $SS$ = 0.1%, 0.2%, 0.3%, 0.5%,
and 0.7%, respectively. $\kappa_{CCN}$ in spring are slightly lower than that in winter, with mean
values of 0.31, 0.32, 0.27, 0.22, and 0.21 at $SS$ = 0.1%, 0.2%, 0.3%, 0.5%, and 0.7%,
respectively. In autumn, the mean $\kappa_{CCN}$ are 0.27, 0.26, 0.22, 0.19, and 0.19 at $SS$ = 0.1%,
0.2%, 0.3%, 0.5%, and 0.7%, respectively, which is slightly higher than that observed
in summer.

Figure 6b presents the monthly variation of $D_c$ at each $SS$ condition, which shows

the opposite trend to $\kappa_{CCN}$ - $SS$ because of the negative correlation of $D_c^3$ vs. $\kappa$ shown
in equation 2a. The seasonal mean $D_c$ are shown in Table 4. The mean $D_c$ plus/minus
one standard deviation over the whole period are 177±19, 112±14, 91±15, 67±9, and
54±8 nm at $SS$ = 0.1%, 0.2%, 0.3%, 0.5%, and 0.7%, respectively. The largest $D_c$ at
each $SS$ condition were observed in summer, with mean values of 187, 116, 94, 69, and
55 nm at $SS$ = 0.1%, 0.2%, 0.3%, 0.5%, and 0.7%, respectively. Followed by autumn
and spring, the smallest $D_c$ at each $SS$ condition was observed in winter, with mean





values of 168, 107, 86, 64, and 53 nm at $SS$ = 0.1%, 0.2%, 0.3%, 0.5%, and 0.7%,
respectively.

The monthly average of the external-mixing degree is shown in Figure 6c. The

degree of external mixture is quantified by the ratio of $(D_{75} - D_{25})/D_c$. The seasonal
mean $(D_{75} - D_{25})/D_c$ are presented in Table 4. Jurányi et al. (2013) pointed out that the
$(D_{75} - D_{25})/D_c$. ranged from 0.08 to 0.12 for ammonium sulfate calibration
measurements at $SS$ = 0.1−1.0%, which indicated an internal mixture within
measurement accuracy. For our measurements, the mean $(D_{75} - D_{25})/D_c$ over all datasets
range from 0.17 to 0.25 at $SS$ = 0.1−0.7%. In summer, $(D_{75} - D_{25})/D_c$ is lowest ranging
from 0.14 to 0.18 at $SS$ = 0.1−0.7%, implying that aerosol particles were extremely
close to being internally mixed. Followed by spring and autumn, the highest $(D_{75} -$
$D_{25})/D_c$ was observed in winter with values ranging from 0.24 to 0.36 at $SS$ = 0.1−0.7%.
Therefore, the results tend to indicate that the aerosol particles were less internally
mixed in winter among the four seasons at Melpitz. In non-urban locations, initially
externally mixed aerosol particles become an internal mixture on a time scale of ~1 day
(Fierce et al., 2016). In winter, the relatively stable weather patterns increase the
persistence of aerosol (> 5 days) at Melpitz (Schmale et al., 2018). When tracking an
aerosol cluster, the prolonged mixing time should promote the aging process, leading
to an internal mixture. However, we observed a less internally mixed aerosol particle
population in winter. A plausible explanation is mixing in of local pollution.

Essentially, the relationship between $\kappa_{CCN}$ and $SS$ is determined by the $\kappa_{CCN}$ vs. $D_p$

relationship. Identically, the relationship between $(D_{75} - D_{25})/D_c$ and $SS$ depends on the
$(D_{75} - D_{25})/D_c$ vs. $D_p$ relationship. Monodisperse CCN measurements provide the size-
resolved $\kappa$ and $(D_{75} - D_{25})/D_c$. At a given $SS$ condition, $\kappa_{CCN}$ represents the $\kappa$ of particles
at $D_p = D_c$, and the same is true for $(D_{75} - D_{25})/D_c$. It should be noted that our



monodisperse CCN measurements only provide the size-resolved $\kappa$ and $(D_{75} - D_{25})/D_c$
within $D_p$ of ~40−200 nm.

As shown in Figure 7a, $\kappa_{CCN}$ increases with $D_p$ at $D_p$ of ~40 to 100 nm, whereas

$\kappa_{CCN}$ almost stays constant at $D_p$ of 100 to 200 nm for all seasons. Additionally, the
increase $\kappa_{CCN}$ with $D_p$ varies with season. The $\kappa_{CCN}$ vs. $D_p$ relationship is fitted by a
power-law function at each season. Fit results are presented in Table 5. In summer, there
is the lowest slope parameter of 0.19 in the $\kappa_{CCN}$ vs. $D_p$ power-law fit, meaning that the
difference between the $\kappa_{CCN}$ at different particle sizes is smallest among the four seasons.
Followed by autumn with the slope parameter of 0.31, the slope parameter is highest in
spring and winter of 0.36−0.37. Therefore, the $\kappa_{CCN}$ is most sensitive to $D_p$ in spring and
winter.

Figure 7b presents the $(D_{75} - D_{25})/D_c$ vs. $D_p$ relationship. As particle size increases,

$(D_{75} - D_{25})/D_c$ decreases at $D_p$ of ~40 to 200 nm for all seasons, meaning that small
particles are less internally mixed. The reason is that during the aerosol aging process,
not only particle size increases but $\kappa$ becomes more uniform. The $(D_{75} - D_{25})/D_c$ vs. $D_p$
relationship is also fitted well by the power-law function at each season, with fit results
shown in Table 5. The highest absolute value of the slope parameter was observed in
autumn of 0.42, followed by winter of 0.30 and spring of 0.26, and the lowest was 0.20
observed in summer. Thus, the difference between the degree of external mixture at
different particle sizes is largest in autumn, followed by winter and spring, and is
smallest in summer.
**3.3 $N_{CCN}$ prediction at Melpitz**

$N_{CCN}$ plays an important role in modeling the formation and evolution of clouds

(Zhao et al., 2012; Fan et al., 2012, 2018). This section evaluates the accuracy of $N_{CCN}$
predicted from five different schemes. Table 6 introduces the five schemes, which can





be summarized into two categories of $N_{CCN}$ prediction approach. The fit results of $N_{CCN}$
- *SS* relationship and *AR* - *SS* relationship can predict $N_{CCN}$ at the given *SS* conditions,
which belongs to the 1st category approach, corresponding to the 1st and 2nd schemes in
Table 6, respectively. Compared to CCN measurements, it is generally more common
and simpler to obtain the PNSD measurements; thus, we usually predict $N_{CCN}$ using the
real-time PNSD combined with the parameterized CCN activity, which belongs to the
2nd category approach. The last three schemes in Table 6 belong to the 2nd category
approach, but they vary in assuming $\kappa$. The 3rd scheme uses a fixed $\kappa$ of 0.3 without
temporal and size-dependent variations, as recommended for continental aerosol
(Andreae and Rosenfeld., 2008). The 4th scheme uses the bulk $\kappa_{chem}$ calculated from
aerosol chemical composition, which is also non-size-dependent but changes over time.
The 5th scheme uses the $\kappa$ - $D_p$ power-law fit results shown in Table 5, which are size-
dependent without temporal variations at each season. Applying the $\kappa$ - $D_p$ power-law
equation into equation 2a, $D_c$ can be derived as function of *SS*,

$$D_c = \left( \frac{4 \times A^3}{27 \times coef \times \ln^2(1 + SS/100)} \right)^{\frac{1}{slope+3}} \quad (5)$$

where the *slope* and *coef* represent the slope parameter and the coefficient in $\kappa$ - $D_p$
power-law fit. Subsequently, the predicted $N_{CCN}$ can be calculated through equation 4.
The last three schemes all assume that aerosol particles are internally mixed at a
particular $D_p$, as used in many previous $N_{CCN}$ prediction studies (e.g., Deng et al., 2013;
Pöhlker et al., 2016; Wang et al., 2018a).
The prediction results are shown in Figure 8. The linear equation (y = kx) is used
to fit the relationship between the predicted $N_{CCN}$ and the measured one, and its slope
represents the mean ratio of the predicted $N_{CCN}$ to the measured $N_{CCN}$. To make the
results of the predictions comparable for all regression schemes, we also applied a linear



regression to the 1$^{st}$ scheme and forced the linear regression through zero for all
schemes. The relative deviation (RD) equals the ratio of the absolute difference between
the predicted $N_{CCN}$ and the measured one to the measured $N_{CCN}$, i.e., RD = (|predicted
$N_{CCN}$ – measured $N_{CCN}$|)/measured $N_{CCN}$; a large RD represents a large deviation
between prediction and measurement. The slope and RD shown in Figure 8 are both
calculated from all five $SS$ conditions for each season. As shown in Figure 8, the 1$^{st}$ and
2$^{nd}$ schemes only provide rough estimates of the $N_{CCN}$ on account of the pretty high RD
ranging from 64% to 136%. Compared to the 1$^{st}$ category approach (the 1$^{st}$ and 2$^{nd}$
schemes), the 2$^{nd}$ category approach (the 3$^{rd}$, 4$^{th}$, and 5$^{th}$ schemes) predicts $N_{CCN}$ better.
The predicted $N_{CCN}$ correlates well with the measured one for the 3$^{rd}$, 4$^{th}$, and 5$^{th}$
schemes with R$^2$ > 0.97; but $N_{CCN}$ is generally overestimated for the 3$^{rd}$ and 4$^{th}$ schemes
because the fit slopes range from 1.03 to 1.17 for different seasons. The 5$^{th}$ scheme
appears to be the best one for $N_{CCN}$ prediction among the five schemes on account of
the lowest RD ranging from 11% to 17% and the fit slope of ~1 for different seasons. It
should be noted that the fit slope shown in Figure 8 represents the average over all five
$SS$ conditions, which could obscure the performance at each $SS$ condition. Thus, Figure
9 further evaluates the five schemes for the $N_{CCN}$ prediction at each $SS$ condition.

When using the $N_{CCN}$ - $SS$ power-law fit (the 1$^{st}$ scheme) to predict $N_{CCN}$, it causes

significant overestimations of $N_{CCN}$ at $SS$ = 0.1% with median values ranging from 3%
to 29% for different seasons and causes less than 21% underestimations in median at
other larger $SS$ conditions. Additionally, the prediction results are much uncertain at a
given $SS$ condition and season, especially at $SS$ = 0.1%. For instance, one-quarter of the
predicted $N_{CCN}$ are twice higher than the measured values at $SS$ = 0.1% for all datasets.
Thus, this scheme can only be used to provide rough estimations of $N_{CCN}$.

When using the real-time $N_{aero}$ combined with $AR$ - $SS$ power-law fit (the 2$^{nd}$





scheme) to predict $N_{CCN}$, the performances are slightly better than those of the 1st
scheme. The median overestimations of $N_{CCN}$ are less than 17% at $SS = 0.1\%$ for all
seasons, while the median underestimations of $N_{CCN}$ range from 12% to 35% at $SS =$
0.2%−0.7% for all seasons. Similarly, the prediction results remain a high uncertainty
at a given $SS$ condition and season. Thus, this scheme also provides rough estimations
on $N_{CCN}$.
When assuming the real-time PNSD combined with a constant $\kappa$ of 0.3 (the 3rd
scheme) to predict $N_{CCN}$, it causes overestimations of $N_{CCN}$ in most cases. The median
of the overestimation ranges from -3% to 30% at $SS = 0.1\%−0.7\%$ for different seasons.
As shown in Figure 7a, a constant $\kappa$ of 0.3 is almost greater than the $\kappa_{CCN}$ of all particles
with the diameter ranging from ~40 to 200 nm, except for the accumulation-mode
particles ($D_p$ of 100 to 200 nm) in spring and winter. Therefore, besides the well-
predicted $N_{CCN}$ at $SS = 0.1\%$ and 0.2% in spring and winter, $N_{CCN}$ is overestimated at
assuming a constant $\kappa$ of 0.3 as shown in Figure 9c. The largest overestimation occurs
at $SS = 0.1\%$ in summer (30% in median) because of the low $\kappa_{CCN}$ (0.22 in average)
combined with the low measured $N_{CCN}$ (347 cm$^{-3}$ in average). Although the largest
median overestimation reaches to 30%, which is numerically similar to the largest
median overestimation of the 1st scheme (29%) and the largest median underestimation
of the 2nd scheme (35%), the uncertainties of the 3rd scheme are much lower than those
of the 1st and 2nd schemes. For example, when using 3rd scheme, one-quarter of the ratio
of the predicted $N_{CCN}$ to the measured $N_{CCN}$ are larger than 1.31 at $SS = 0.1\%$ for all
datasets as shown in Figure 9c, while the ratio is ~2.0 for both the results of 1st and 2nd
scheme as shown in Figures 9a and 9b. Thus, the 3rd scheme has better predictions on
$N_{CCN}$ compared to the 1st and 2nd schemes.
When assuming the real-time PNSD combined with the real-time bulk $\kappa_{chem}$ (the




4$^{th}$ scheme) to predict $N_{CCN}$, it also causes clear overestimations of $N_{CCN}$ in most cases,
like the prediction results calculated from the 3$^{rd}$ scheme. The median overestimations
are within 7% to 21% at $SS$ = 0.1%−0.7% for different seasons. The reason for the
overestimation is that the $\kappa_{chem}$ is greater than $\kappa_{CCN}$ measured at all the five $SS$
conditions. For instance, the mean $\kappa_{CCN}$ over all datasets ranges from 0.20 to 0.28 at $SS$
= 0.1%−0.7%, whereas the mean $\kappa_{chem}$ over all datasets is 0.36. The largest
overestimation also occurs at $SS$ = 0.1% in summer with 21% in median. Compared to
the 3$^{rd}$ scheme, the uncertainty of the $N_{CCN}$ prediction at a given $SS$ condition and season
is lower in the 4$^{th}$ scheme. Considering the median overestimations of the predicted
$N_{CCN}$ at different seasons and $SS$ conditions and the uncertainty of the predicted $N_{CCN}$ at
each given season and $SS$ condition, we conclude that the performances of the 4$^{th}$
scheme are better than the 3$^{rd}$ scheme.

When assuming the real-time PNSD combined with the $\kappa$ - $D_p$ power-law fit (the

5$^{th}$ scheme) to predict $N_{CCN}$, it can predict the $N_{CCN}$ well at each $SS$ condition for all
seasons. At $SS$ = 0.1%, it causes less than 10% overestimation in median for $N_{CCN}$
prediction for all seasons; at $SS$ = 0.2%−0.7%, the median overestimation ranges from
-3% to 6% for all seasons. The uncertainty of the $N_{CCN}$ prediction at a given $SS$ condition
and season is also smallest among the five schemes, especially at relatively high $SS$
conditions (e.g., 0.5% and 0.7%). For instance, at $SS$ = 0.7% for all datasets, when using
the 5$^{th}$ scheme, one-quarter of the ratio of the predicted $N_{CCN}$ to the measured $N_{CCN}$ are
larger than 1.10, while the ratio ranges from 1.18 to 1.38 for other four schemes.
Therefore, the 5$^{th}$ scheme provides the best $N_{CCN}$ prediction among the five schemes.

Overall, the performance for $N_{CCN}$ prediction is gradually getting better from the

1$^{st}$ to the 5$^{th}$ scheme shown in Table 6. The classic $N_{CCN}$ - $SS$ and $AR$ - $SS$ power-law fits
shown in Table 3 can only be used to provide rough estimates of the $N_{CCN}$. At Melpitz,


using a constant $\kappa$ of 0.3 or the bulk $\kappa_{chem}$ both causes significant overestimations of
$N_{CCN}$ with about 30% in median, especially at $SS = 0.1\%$ in summer. The $\kappa$ - $D_p$ power-
law fit at each season shown in Table 5 is recommended applying for $N_{CCN}$ prediction
at Melpitz, which can narrow down the prediction deviation (ratio of the predicted $N_{CCN}$
to the measured $N_{CCN}$ minus 1) within 10% in median. Additionally, as shown in Figure
10, the $\kappa$ - $D_p$ power-law fit measured at Melpitz is similar to that measured at other
rural and continental regions with similar aerosol background conditions, e.g., the
Vavihill station in Sweden (Fors et al., 2011) and the Xinken station in China (Eichler
et al., 2008), and is also valid for some urban (Ye et al., 2013) and suburb regions
(Mazoyer et al., 2019). Therefore, the $\kappa$ - $D_p$ power-law fit measured at Melpitz could
be applied to predict $N_{CCN}$ for these regions. However, it may cause considerable
deviations for different aerosol background regions, e.g., the polluted suburb station in
Xingtai, China (Wang et al., 2018a), the coast of Barbados (Kristensen et al., 2016), the
amazon rainforest (Pöhlker et al., 2016), and the urban station in Budapest, Hungary
(Salma et al., 2021), because their $\kappa$ - $D_p$ relationships are different from that measured
at Melpitz.

Additionally, it should be noted that the main size dependence of $\kappa$ occurs at $D_p$ of

~40 to 100 nm as shown in Figure 7a, which would be for $SS$ larger than 0.2%. At $D_p$
of 100 to 200 nm corresponding to $SS$ less than 0.2%, $\kappa$ almost stays constant. The mean
value of $\kappa$ is close to 0.3 for spring and winter, and that's where deviations in Figure 9c
are small. However, the mean value of $\kappa$ overestimates the $\kappa$ for $SS$ larger than 0.2% at
each season. We further compare the $N_{CCN}$ predictions between using the seasonally
mean value of $\kappa$ over $D_p$ of 100 to 200 nm and the $\kappa$ - $D_p$ power-law fit. As shown in
Figure 11, at $SS = 0.1$ and 0.2%, the seasonally mean $\kappa$ value over $D_p$ of 100 to 200 nm





and $\kappa$ - $D_p$ power-law fit both predict the $N_{CCN}$ well at each season, while the mean $\kappa$
value leads to significant overestimation of $N_{CCN}$ within 10% on average at $SS = 0.3$,
0.5, and 0.7%. Therefore, to predict the $N_{CCN}$ at a relatively low $SS$ of less than 0.2%
(e.g., in fog and shallow stratiform cloud), the mean $\kappa$ value over $D_p$ of 100 to 200 nm
also works well. The mean value plus/minus one standard deviation are $0.32\pm0.09$,
$0.24\pm0.07$, $0.26\pm0.09$, $0.32\pm0.10$ and $0.28\pm0.09$ for spring, summer, autumn, winter,
and all datasets, respectively.

## 4. Conclusions

Aerosol particle activation plays an important role in determining the number

concentration of cloud droplets, thereby affecting cloud microphysics, precipitation
processes, radiation, and climate. To reduce the uncertainties and gain more confidence
in the simulations on AIEs, long-term measurements on aerosol activation
characteristics are essential; however, still rarely reported. Based on more than 4-year
comprehensive measurements conducted at the central European ACTRIS site Melpitz,
Germany, this study presents a systematic seasonal analysis of aerosol activation
characteristics and $N_{CCN}$ predictions.

Over the whole period, the mean $N_{CCN}$ and AR increased from 513 to 2477 cm$^{-3}$

and 0.11 to 0.52 with $SS$ increasing from 0.1% to 0.7%, respectively. Aerosol activation
characteristics are highly variable across seasons. At $SS = 0.1\%$, the seasonal mean $N_{CCN}$
is 681 cm$^{-3}$ in winter, which is almost twice higher than the summer value (347 cm$^{-3}$);
the seasonal mean AR is 0.18 in winter, which is three times higher than the summer
value (0.06). Aerosol particle activation depends on its physical and chemical properties.
Affected by the frequent NPF events, in summer, the mean $N_{aero}$ is highest (6224 cm$^{-3}$)
and the mean $GMD$ is smallest (50 nm) among the four seasons. On the contrary in
winter, the mean $N_{aero}$ is lowest (3686 cm$^{-3}$) and the mean $GMD$ is largest (58 nm). In



summer, the mean $f_{org}$ (56%) is highest among the four seasons, corresponding to the
lowest $\kappa_{chem}$ with a mean value of 0.29; in winter, the mean $f_{nitrate}$ (36%) is highest
among the four seasons, which might explain the highest mean $\kappa_{chem}$ (0.40). Therefore,
in winter, the highest $\kappa_{chem}$, largest $GMD$, and the lowest $N_{aero}$ cause the highest AR at
each $SS$ condition among the four seasons.

Both $\kappa$ and the mixing state are size-dependent, thereby varying with $SS$. The mean

$\kappa$ is 0.28, 0.28, 0.24, 0.21, and 0.20 at $SS$ = 0.1%, 0.2%, 0.3%, 0.5%, and 0.7%,
respectively. $D_c$ depends on $\kappa$ at a given $SS$ condition. The mean $D_c$ is 177, 112, 91, 67,
and 54 nm at $SS$ = 0.1%, 0.2%, 0.3%, 0.5%, and 0.7%, respectively. For different
seasons, the seasonal mean $\kappa$ varies from 0.23 to 0.32 at $SS$ = 0.1%, and 0.19 to 0.21 at
$SS$ = 0.7%; the seasonal mean $D_c$ varies from 168 nm to 187 nm at $SS$ = 0.1%, and 53
nm to 55 nm at $SS$ = 0.7%. The degree of external mixture is quantified by the $(D_{75} -$
$D_{25})/D_c$, which ranges from 0.17 to 0.25 in average over the whole period at $SS$ =
0.1−0.7%. In summer, aerosol particles were extremely close to being internally mixed
with $(D_{75} - D_{25})/D_c$ ranging from 0.14 to 0.18 at $SS$ = 0.1−0.7%; in winter, particles
were less internally mixed among the four seasons with $(D_{75} - D_{25})/D_c$ ranging from
0.24 to 0.36 at $SS$ = 0.1−0.7%. As $D_p$ increases, $\kappa$ increases at $D_p$ of ~40 to 100 nm and
almost stays constant at $D_p$ of 100 to 200 nm), and $(D_{75} - D_{25})/D_c$ decreases for all
seasons. The relationships of $(D_{75} - D_{25})/D_c$ vs. $D_p$ and $\kappa$ vs. $D_p$ are both fitted well by
the power-law function for each season.

Five activation schemes are evaluated on the $N_{CCN}$ predictions. Compared to using

the classic $N_{CCN}$ - $SS$ or $AR$ - $SS$ power-law fits to predict $N_{CCN}$, the prediction is better
by using the real-time PNSD combined with the parameterized $\kappa$, including a constant
$\kappa$ of 0.3, the bulk $\kappa_{chem}$, and the $\kappa$ - $D_p$ power-law fit. However, assuming a constant $\kappa$





of 0.3 recommended for continental aerosol (Andreae and Rosenfeld., 2008) or the bulk
$\kappa_{\text{chem}}$ calculated from aerosol chemical composition both cause significant
overestimations of the $N_{CCN}$ with about 30% in median, especially at $SS$ = 0.1% in
summer. Generally, the performances of the latter (the bulk $\kappa_{\text{chem}}$) are slightly better
than the former (a constant $\kappa$ of 0.3) on account of the lower uncertainty at each given
season and $SS$ condition. Size-resolved $\kappa$ improves the $N_{CCN}$ prediction. We recommend
applying the $\kappa$ - $D_p$ power-law fit for $N_{CCN}$ prediction, which obtains the best prediction
among the five schemes. At Melpitz, using the real-time PNSD combined with the $\kappa$ -
$D_p$ power-law fit could narrow down the uncertainty of $N_{CCN}$ prediction within 10% in
median for all $SS$ conditions (0.1−0.7%) and seasons. The $\kappa$ - $D_p$ power-law fit
presented in this study could apply to other rural and continental regions with similar
aerosol background conditions. To our knowledge, the $\kappa$ - $D_p$ power-law fit is the first
time applied to predict $N_{CCN}$. Additionally, the mean $\kappa$ value over $D_p$ of 100 to 200 nm
also works well to predict $N_{CCN}$ at a relatively low $SS$ of less than 0.2%.





**Appendix B Notation list**

| | |
|---|---|
| $A$ | comprehensive parameter for $\sigma_{s/a}$, $M_w$, $R$, $T$, and $\rho_w$ in equation (2b) |
| $a$ | lower limit for calculating critical diameters at the set-nominal supersaturations in equation (1) |
| ACI | aerosol and cloud interactions |
| ACSM | aerosol chemical species monitor |
| ACTRIS | Aerosol, Clouds and Trace Gases Research Infrastructure |
| AF | activated fraction, i.e., $N_{CCN}/N_{CN}$ |
| AIEs | aerosol indirect effects |
| AR | activation ratio, i.e., $N_{CCN}/N_{aero}$ |
| $b$ | upper limit for calculating critical diameters at the set-nominal supersaturations in equation (1) |
| BC | black carbon |
| CN | condensation nuclei |
| CCN | cloud condensation nuclei |
| CCNC | cloud condensation nuclei counter |
| $coef$ | coefficient in $\kappa$ - $D_p$ power-law fit |
| CPC | condensation particle counter |
| $D_p$ | diameter of the dry particle |
| $D_c$ | critical diameter of the dry particle |
| $D_X$ | $D$ at which X % of the particles are activated |
| $(D_{75} - D_{25})/D_c$ | degree of external mixture |
| D-MPSS | Dual-mobility particle size spectrometer |
| DMA | differential mobility analyzer |
| eBC | equivalent black carbon |
| $f_{BC}$ | mass fraction of BC in submicron aerosol |
| $f_{nitrate}$ | mass fraction of nitrate in submicron aerosol |
| $f_{org}$ | mass fraction of organics in submicron aerosol |
| $f_{sulfate}$ | mass fraction of sulfate in submicron aerosol |
| $GMD$ | geometric mean diameter of PNSD |
| $M_w$ | molecular weight of water |
| $N_{aero}$ | number concentration of aerosol with $D_p$ ranging 10 to 800 nm |
| $N_{CN}$ | number concentration of CN |
| $N_{CCN}$ | number concentration of CCN |
| NPF | new particle formation |
| $PM_{10}$ | particulate matter with the $D_p < 10$ μm |
| PNSD | particle number size distribution |
| $R$ | universal gas constant |
| $R^2$ | coefficient of determination |
| RD | relative deviation between the predicted $N_{CCN}$ and the measured one |
| $SS$ | supersaturation |
| $T$ | temperature |
| $\sigma_s$ | represent the shape of the sigmoid function |


| | |
|---|---|
| $\sigma_{s/a}$ | droplet surface tension |
| $\kappa$ | hygroscopicity factor of aerosol particle |
| $\kappa_i$ | $\kappa$ *of* each component |
| $\kappa_{CCN}$ | $\kappa$ calculated from the monodisperse CCN measurements |
| $\kappa_{chem}$ | $\kappa$ calculated from the aerosol chemical measurements |
| $\varepsilon_i$ | volume fraction of each component |
| $\rho_w$ | density of the liquid water |


***Data availability.***

The data used in this study are available from Silvia Henning (henning@tropos.de)

upon request and https://doi.org/10.1594/PANGAEA.938215.

***Author contributions.***

AW, SH and LP designed the research. SH and LP collected the data at Melpitz. YW

performed the data analysis and prepared the paper. All co-authors contributed to

interpretation of the results as well as paper review and editing.

***Competing interests.***

The authors declare that they have no conflict of interest.

***Acknowledgments.***

This research has been supported by the H2020 research infrastructures (grant nos.

ACTRIS (262254) and ACTRIS-2 (654109)), the European Cooperation in Science and

Technology (grant no. COLOSSAL CA16109), the German Federal Environment

Ministry (BMU) grants F&E 370343200 (German title: "Erfassung der Zahl feiner und

ultrafeiner Partikel in der Außenluft"), 2008–2010, and F&E 371143232 (German title:

"Trendanalysen gesundheitsgefährdender Fein- und Ultrafeinstaubfraktionen unter

Nutzung der im German Ultrafine Aerosol Network (GUAN) ermittelten

Immissionsdaten durch Fortführung und Interpretation der Messreihen"), 2012–2014,

the Deutsche Forschungsgemeinschaft (*DFG*, German Research Foundation, HE



6770/2), the Second Tibetan Plateau Scientific Expedition and Research Program
(STEP), Grant No. 2019QZKK0602, the National Natural Science Foundation of China
under grant numbers 42075066, 42075063, 42175099, and 42005067. The China
Scholarship Council (no.202008320513) is acknowledged for supporting Yuan Wang
financially. We thank Achim Grüner and René Rabe for the careful maintenance of the
measurements on site.





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



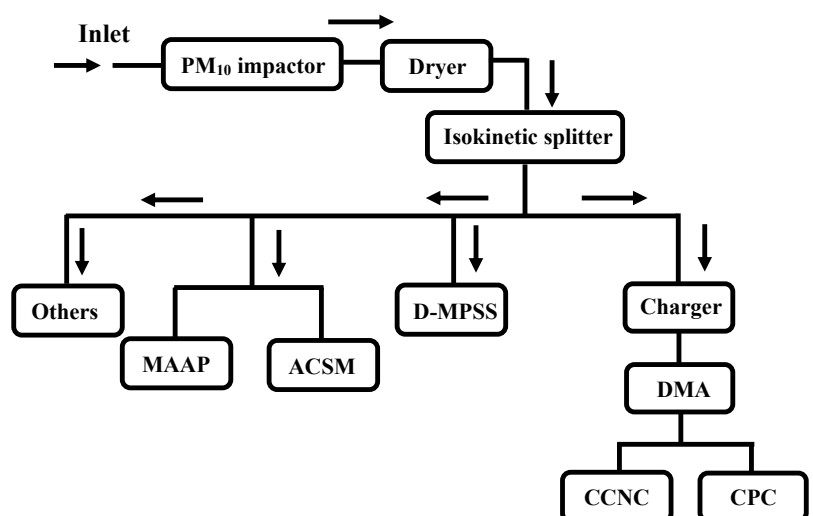


Figure 1. Schematic diagram of the experimental setup. D-MPSS — Dual-mobility particle size
spectrometer, ACSM — aerosol chemical species monitor, MAAP — multi-angle absorption
photometer, DMA — differential mobility analyzer, CPC — condensation particle counter, CCNC
— cloud condensation nuclei counter.



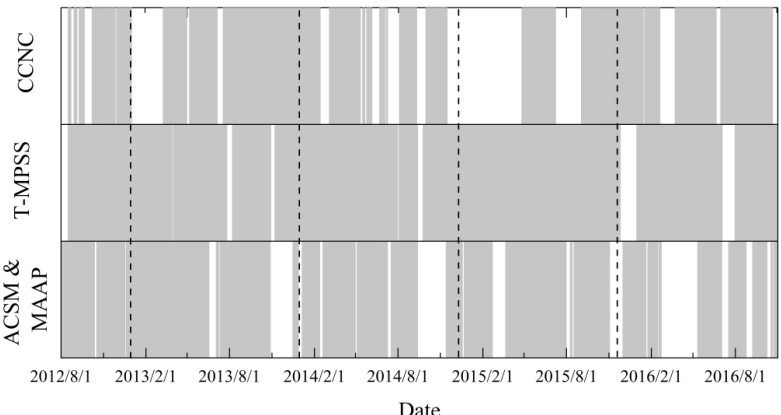


Figure 2. Coverage of the effective data represented by the gray columns.

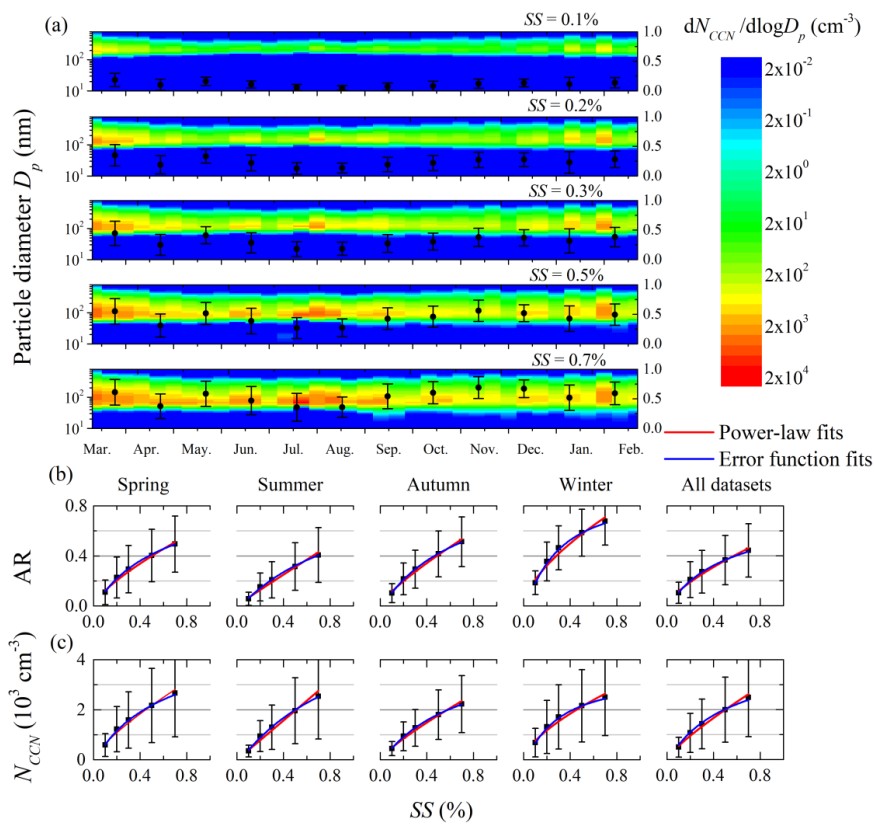


Figure 3. Seasonal variations of (a) CCN number size distributions and activation ratios (AR) at five
different supersaturation ($SS$) conditions, (b) relationship between AR and $SS$ for different seasons,
and (c) relationship between CCN number concentration ($N_{CCN}$) and $SS$ for different seasons. Error
bar means one standard deviation. Red lines and blue lines are the fittings for AR vs. $SS$ and $N_{CCN}$
vs. $SS$ with using the power-law function and the error function, respectively. Fitting results are
shown in Table 3.



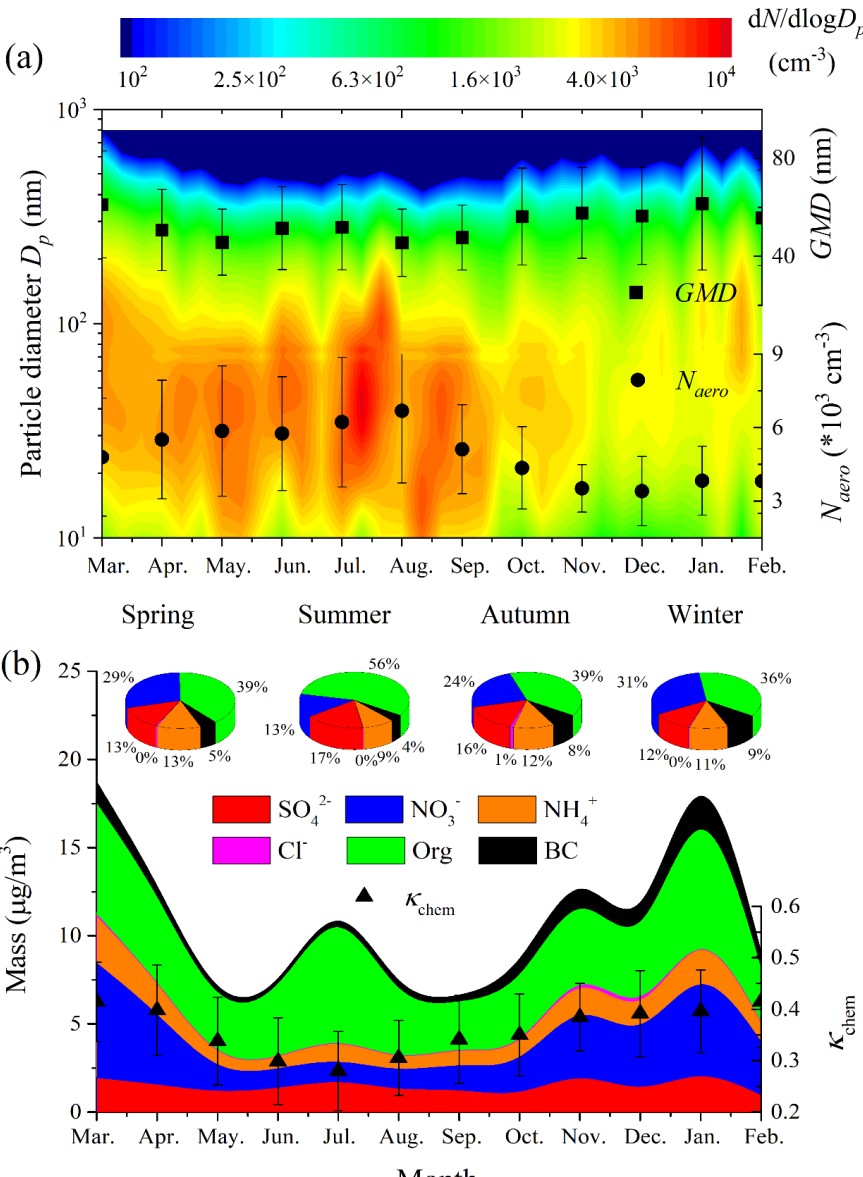


Figure 4. Seasonal variations of (a) aerosol physical and (b) chemical properties. $dN_{aero}/dlogD$

represents the aerosol number concentration at each bin, *GMD* is the geometric mean diameter of

the particles, $N_{aero}$ means total aerosol number concentration with diameter ranging 10 to 800 nm,

$\kappa_{chem}$ is the hygroscopicity factor calculated from the chemical composition. Error bar is one

standard derivation.



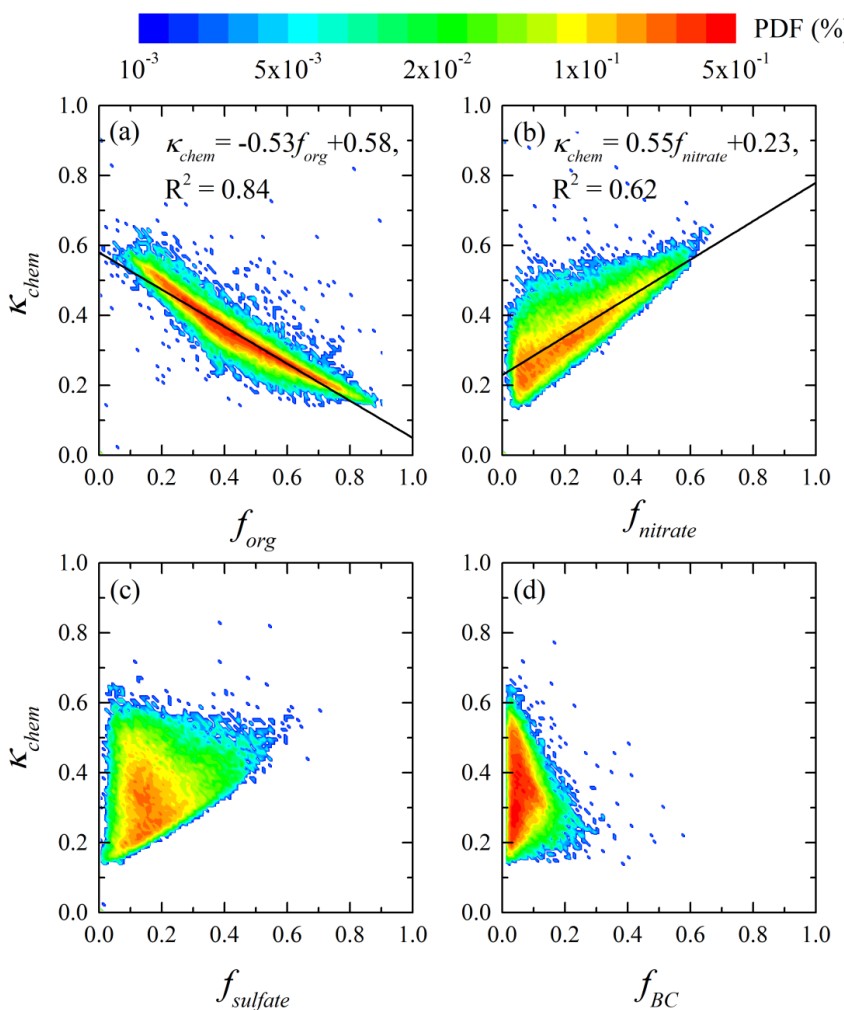

Figure 5. Relationships between (a) aerosol hygroscopicity factor calculated from the chemical composition ($\kappa_{chem}$) and mass fraction of organics ($f_{org}$) in submicron aerosol, (b) $\kappa_{chem}$ vs. mass fraction of nitrate ($f_{nitrate}$), (c) $\kappa_{chem}$ vs. mass fraction of nitrate ($f_{sulfate}$), and (d) $\kappa_{chem}$ vs. mass fraction of black carbon ($f_{BC}$). Color bar represents the probability density function (PDF). Black lines are linear fit lines.

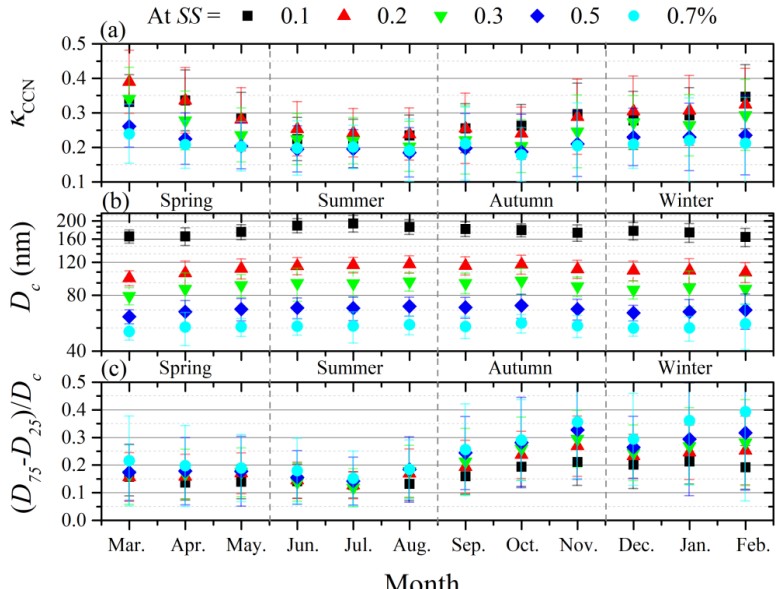

1022

Figure 6. Monthly average of (a) hygroscopicity factor calculated from monodisperse CCN

measurements ($\kappa_{CCN}$), (b) critical diameter of dry particle for activation ($D_c$), and (c) the degree of

external mixture (($D_{75} - D_{25}$)/$D_c$) at five different supersaturation ($SS$) conditions. The definitions

of $D_{75}$ and $D_{25}$ are the $D_p$ at which 75% and 25% of the particles are activated at the given $SS$,

respectively. Error bar is one standard derivation.

1028

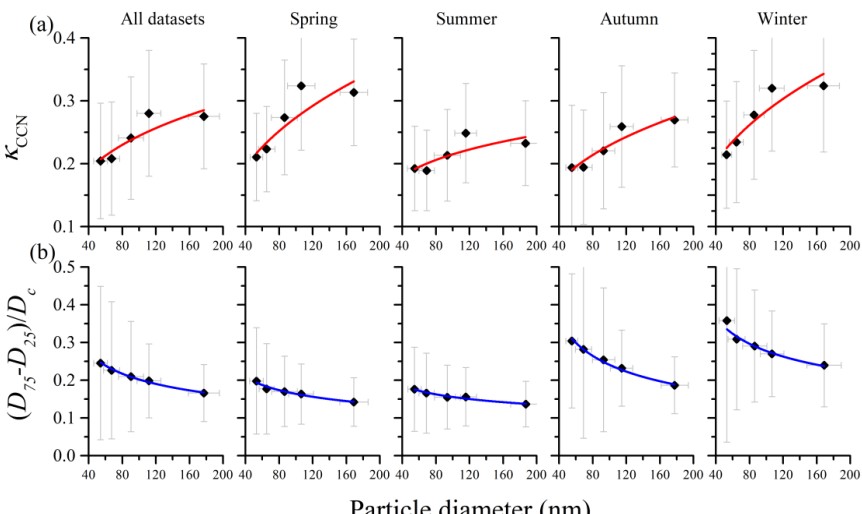

1029

Figure 7. (a) Relationship between the particle diameter ($D_p$) and hygroscopicity factor calculated

from monodisperse CCN measurements ($\kappa_{CCN}$), and (b) $D_p$ vs. degree of external mixture (($D_{75} - D_{25}$)/$D_c$) at each season. The definitions of $D_{75}$ and $D_{25}$ are the $D_p$ at which 75% and 25% of the

particles are activated at the given $SS$, respectively. Red and blue lines are power-law fits for $\kappa_{CCN}$

vs. $D_p$ and ($D_{75} - D_{25}$)/$D_c$ vs. $D_p$.



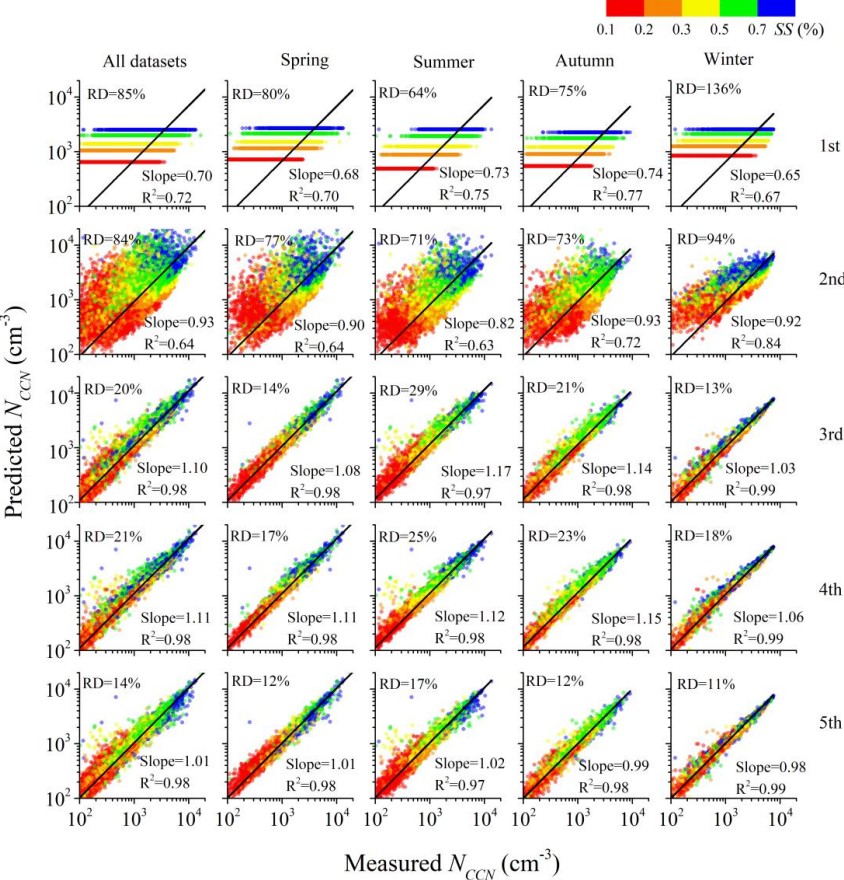

1036

Figure 8. Predicted vs. measured CCN number concentration ($N_{CCN}$) for different seasons. The

Predicted $N_{CCN}$ is calculated from five different schemes with a detailed introduction shown in Table

6. Color bar represents the different supersaturation ($SS$) conditions. Black lines are the linear fits.

The slope and $R^2$ of the linear regression and the relative deviation (RD) of the predicted $N_{CCN}$ (RD=

(|predicted $N_{CCN}$ − measured $N_{CCN}$|)/measured $N_{CCN}$) are shown in each panel. Each row represents

the results at the same scheme in different seasons; each column represents the results at different

schemes in the same season.

1044

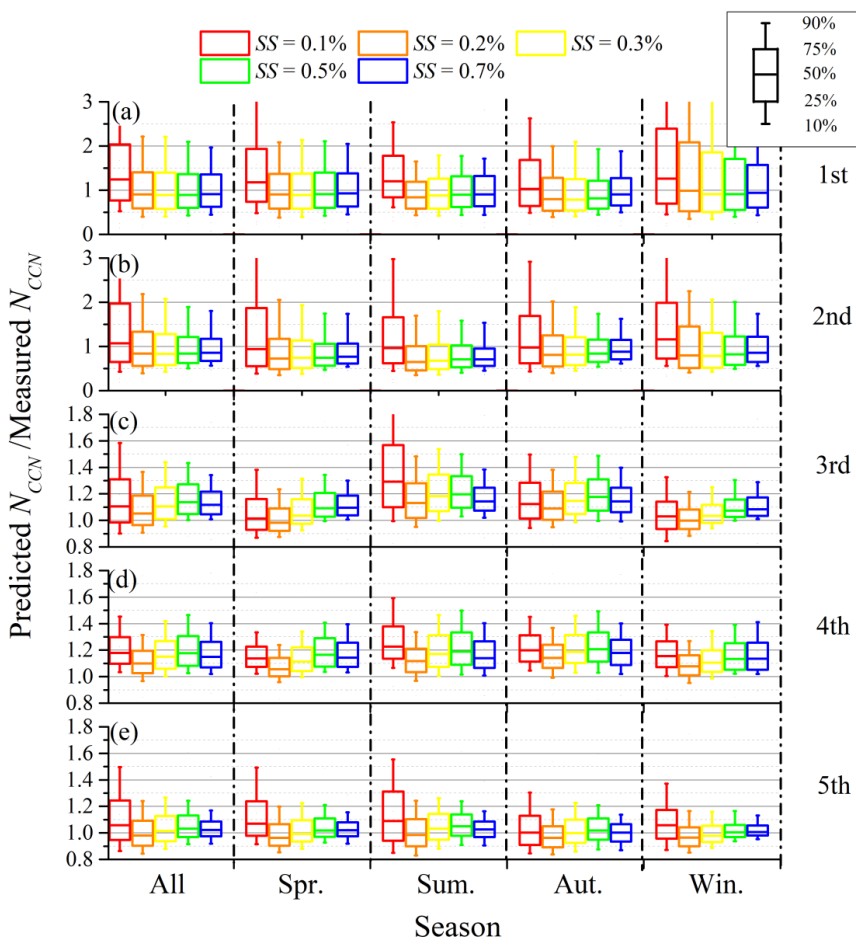

Figure 9. Statistics of the ratio of predicted CCN number concentration ($N_{CCN}$) to the measured one

at different supersaturation (SS) conditions for each season and all datasets. The (a), (b), (c), (d), and

(e) represent the prediction results from the 1st, 2nd, 3rd, 4th, and 5th scheme, respectively.





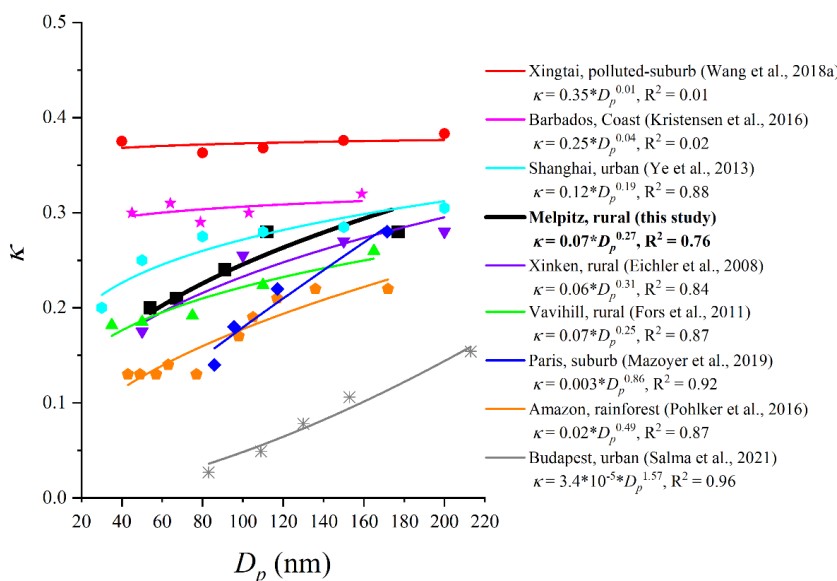

Figure 10. Relationships between the particle hygroscopicity factor ($\kappa$) and diameter ($D_p$) observed

at different aerosol background regions. Lines are the power-law fits of $\kappa$ vs. $D_p$.

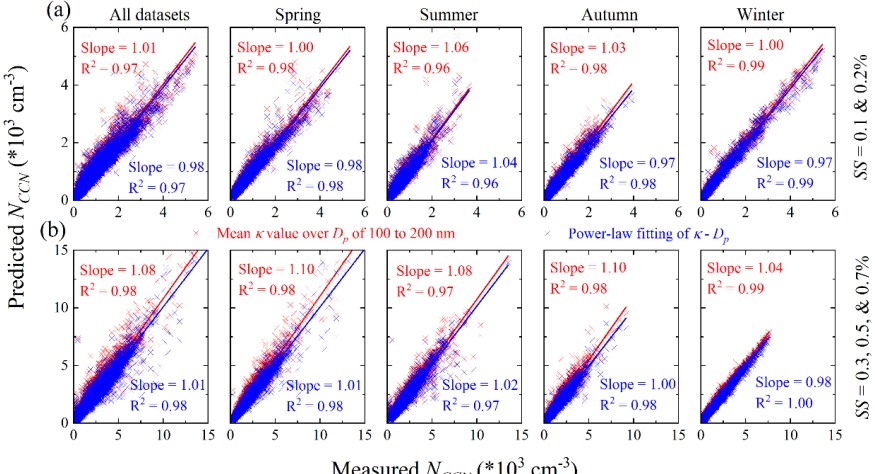


Figure 11. Predicted vs. measured CCN number concentration ($N_{CCN}$) at different supersaturation ($SS$) conditions for different seasons. (a) represents the results at $SS$ = 0.1 and 0.2%; (b) shows the results at $SS$ = 0.3, 0.5, and 0.7%. Red cross represents the predicted $N_{CCN}$ using mean hygroscopicity factor ($\kappa$) over particle diameter ($D_p$) of 100 to 200 nm, while the blue cross represents the predicted $N_{CCN}$ using power-law fit of $\kappa$ and $D_p$. Red and blue lines are the linear fits.






Table 1. Densities ($\rho$) and hygroscopicity factor ($\kappa$) for each component.

| Species | $NH_4NO_3$ | $(NH_4)_2SO_4$ | $NH_4HSO_4$ | $H_2SO_4$ | Organics | BC |
|---|---|---|---|---|---|---|
| $\rho$ (kg m$^{-3}$) | 1720 | 1769 | 1780 | 1830 | 1400 | 1700 |
| $\kappa$ | 0.67 | 0.61 | 0.61 | 0.92 | 0.1 | 0 |






Table 2. Summary of CCN number concentration ($N_{CCN}$) at different supersaturation ($SS$) conditions
measured at different locations.

| Location (coordinates; a.m.s.l) | Type | Period | $SS$ (%) | Mean $N_{CCN}$ (cm⁻³) | Reference |
|---|---|---|---|---|---|
| Melpitz, Germany (51.5°N, 12.9°E; 86 m) | rural, continental | Aug. 2012–Oct. 2016 | 0.1<br>0.2<br>0.3<br>0.5<br>0.7 | 513<br>1102<br>1466<br>2020<br>2477 | Present study |
| Vavihill, Sweden (56.0°N, 13.2 °E; 172 m) | rural | May 2008–Jul 2010 | 0.1–1.0 | 362–1795 | Fors et al., 2011 |
| Southern Great Plains, USA (36.6°N, 97.5°W; 320 m) | rural, agricultural | Sep. 2006–Apr. 2011 | 0.4 | 1248 | Liu and Li, 2014 |
| Hyytiälä, Finland (61.9°N, 24.3°E; 181 m) | rural | Feb. 2009–Dec. 2012 | 0.1–1.0 | 274–1128 | Paramonov et al., 2015 |
| Mahabaleshwar, India (17.9°N, 73.7°E; ~490 m) | rural | Jun. 2015 | 0.1–0.94 | 118–1826 | Singla et al., 2017 |
| Guangzhou, China (23.6°N, 113.1°E; ~21 m) | rural | Jul. 2006 | 0.068–0.67 | 995–10731 | Rose et al., 2010 |
| Wuqing, China (39.4°N, 117.0°E; 7.4 m) | suburban | Dec. 2009–Jan. 2010 | 0.056–0.7 | 2192–12963 | Deng et al., 2011 |
| Seoul, Korea (37.6°N, 127.0°E; ~38 m) | urban | 2004–2010 | 0.4–0.8 | 4145–6067 | Kim et al., 2014 |
| Mahabubnagar, India (17.7°N, 78.9°E; ~490 m) | polluted continental | Oct. 2011 | 1.0 | ~5400 | Varghese et al., 2016 |



Table 3. Power-law function fits and error function fits for the relationships between activation ratio
(AR) vs. supersaturation (*SS*), and CCN number concentration ($N_{CCN}$) vs. *SS* for different seasons.

| Season | *AR* vs. *SS* | | $N_{CCN}$ vs. *SS* | |
| --- | --- | --- | --- | --- |
| | *Power-law* | *Error Function* | *Power-law* | *Error Function* |
| Spring | $AR$ $=0.66SS^{0.73}$, $R^2=0.98$ | $AR$ $=0.5+0.50\mathrm{erf}(\ln(SS/0.72)/2.33)$, $R^2=0.998$ | $N_{CCN}$ $=3679SS^{0.76}$, $R^2=0.97$ | $N_{CCN}$ $=2637+2637\mathrm{erf}(\ln(SS/0.72)/2.33)$, $R^2=0.998$ |
| Summer | $AR$ $=0.61SS^{0.97}$, $R^2=0.97$ | $AR$ $=0.51+0.51\mathrm{erf}(\ln(SS/1.04)/2.15)$, $R^2=0.997$ | $N_{CCN}$ $=3951SS^{1.01}$, $R^2=0.96$ | $N_{CCN}$ $=3162+3162\mathrm{erf}(\ln(SS/1.04)/2.15)$, $R^2=0.997$ |
| Autumn | $AR$ $=0.71SS^{0.79}$, $R^2=0.98$ | $AR$ $=0.56+0.56\mathrm{erf}(\ln(SS/0.84)/2.29)$, $R^2=0.999$ | $N_{CCN}$ $=3136SS^{0.81}$, $R^2=0.98$ | $N_{CCN}$ $=2433+24336\mathrm{erf}(\ln(SS/0.84)/2.29)$, $R^2=0.999$ |
| Winter | $AR$ $=0.89SS^{0.63}$, $R^2=0.96$ | $AR$ $=0.44+0.44\mathrm{erf}(\ln(SS/0.29)/1.83)$, $R^2=0.999$ | $N_{CCN}$ $=3325SS^{0.64}$, $R^2=0.96$ | $N_{CCN}$ $=1624+1624\mathrm{erf}(\ln(SS/0.29)/1.83)$, $R^2=0.999$ |
| All | $AR$ $=0.59SS^{0.71}$, $R^2=0.98$ | $AR$ $=0.40+0.40\mathrm{erf}(\ln(SS/0.59)/2.25)$, $R^2=0.998$ | $N_{CCN}$ $=3497SS^{0.81}$, $R^2=0.98$ | $N_{CCN}$ $=2199+2199\mathrm{erf}(\ln(SS/0.59)/2.25)$, $R^2=0.998$ |



Table 4. At each supersaturation (*SS*) condition, seasonal mean values of the hygroscopicity factor
calculated from monodisperse CCN measurements ($\kappa_{CCN}$), the critical diameter of dry particle for
activation ($D_c$), and the degree of external mixture (($D_{75} - D_{25}$)/$D_c$). The unit of $D_c$ is nm.

| Parameters | *SS* (%) | All datasets | Spring | Summer | Autumn | Winter |
|---|---|---|---|---|---|---|
| | 0.1 | 0.28 | 0.31 | 0.23 | 0.27 | 0.32 |
| | 0.2 | 0.28 | 0.32 | 0.25 | 0.26 | 0.32 |
| $\kappa_{CCN}$ | 0.3 | 0.24 | 0.27 | 0.21 | 0.22 | 0.28 |
| | 0.5 | 0.21 | 0.22 | 0.19 | 0.19 | 0.23 |
| | 0.7 | 0.20 | 0.21 | 0.19 | 0.19 | 0.21 |
| | 0.1 | 177 | 169 | 187 | 178 | 168 |
| | 0.2 | 112 | 107 | 116 | 115 | 107 |
| $D_c$ | 0.3 | 91 | 87 | 94 | 93 | 86 |
| | 0.5 | 67 | 65 | 69 | 69 | 64 |
| | 0.7 | 54 | 53 | 55 | 55 | 53 |
| | 0.1 | 0.17 | 0.14 | 0.14 | 0.19 | 0.24 |
| | 0.2 | 0.20 | 0.16 | 0.16 | 0.23 | 0.27 |
| ($D_{75} - D_{25}$) /$D_c$ | 0.3 | 0.21 | 0.17 | 0.15 | 0.25 | 0.29 |
| | 0.5 | 0.23 | 0.18 | 0.17 | 0.28 | 0.31 |
| | 0.7 | 0.25 | 0.20 | 0.18 | 0.30 | 0.36 |






Table 5. Power-law fit results in Figure 7. The unit of particle diameter ($D_p$) is nm.

|  | $\kappa_{CCN}$ vs. $D_p$ | $(D_{75} - D_{25})/D_c$ vs. $D_p$ |
|---|---|---|
| All datasets | y = 0.07 x$^{0.27}$, R$^2$ = 0.76 | y = 0.92 x$^{-0.33}$, R$^2$ = 0.99 |
| Spring | y = 0.05 x$^{0.37}$, R$^2$ = 0.76 | y = 0.55 x$^{-0.26}$, R$^2$ = 0.97 |
| Summer | y = 0.09 x$^{0.19}$, R$^2$ = 0.56 | y = 0.39 x$^{-0.20}$, R$^2$ = 0.95 |
| Autumn | y = 0.05 x$^{0.31}$, R$^2$ = 0.88 | y = 1.70 x$^{-0.42}$, R$^2$ = 0.99 |
| Winter | y = 0.05 x$^{0.36}$, R$^2$ = 0.82 | y = 1.10 x$^{-0.30}$, R$^2$ = 0.95 |




Table 6. Introduction of five activation schemes. The meaning of the abbreviation can be found in
Notation list.

| Category | Scheme | Introduction |
|---|---|---|
| 1st category:<br><br>$N_{CCN}$ - $SS$ or $AR$ - $SS$<br><br>empirical fit | 1st | $N_{CCN}$ - $SS$ power-law fits shown in Table 3 |
| | 2nd | Real-time $N_{aero}$ combined with $AR$ - $SS$ power-law fits shown in Table 3 |
| 2nd category:<br><br>Real-time PNSD<br><br>combined with the<br><br>parameterized $\kappa$ | 3rd | Real-time PNSD combined with a constant $\kappa$ of 0.3 |
| | 4th | Real-time PNSD combined with the real-time bulk $\kappa_{chem}$ |
| | 5th | Real-time PNSD combined with $\kappa$ - $D_p$ power-law fits shown in Table 5 |







**Appendix A**

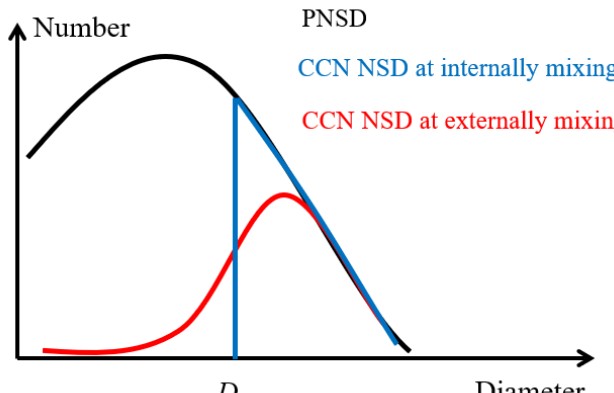


Figure A1. Schematic diagram for the relationship among the particle number size distribution
(PNSD), CCN number size distribution (CCN NSD) at internally mixing, and the CCN NSD at
externally mixing.