# Peer review of "Aerosol activation characteristics and prediction at the 1 central European ACTRIS research station Melpitz, 2 Germany 3 Yuan Wang1,2,3\*, Silvia Henning1\*, Laurent Poulain1, Chunsong Lu2, Frank 4 Stratmann1, Yuying Wang2, Shengjie N"

_Atmospheric Chemistry and Physics, 2022_

## Referee Comment (RC2)

**Review report on Wang et al. 2022, ACPD**

The authors present a very thorough description of a 4 year-long size resolved CCN measurement data set at a central European rural background station. Due to the long dataset, the authors can analyse seasonal trends and test multiple approaches to predict the number of activated particles $N_{CCN}$. This type of data and analysis approach is very valuable for the atmospheric science community. But unfortunately, the authors mostly simply present and describe the measurements and leave the reader wondering what causes these trends and what they may mean in the big picture (e.g., for climate modelling). Adding these necessary discussions constitutes major revisions. I recommend publication after these are done, and the remaining issues listed below have been addressed.

**Main Comments**

1) Throughout the manuscript, the balance between describing the measured values and interpreting them is strongly tilted towards the descriptive side. This data set contains a lot of interesting scientific observations. But in most places the reader is left to wonder what they mean. And if interpretation is provided, it is hidden after all those detailed numbers which are already visualised in the Figures/Tables. By describing every detail, the authors lose the attention of the reader for the really interesting parts. Below is one example to illustrate this issue.

   In Chapter 3.2, the authors describe and name literally all values of k, $D_c$, and $(D_{75}-D_{25})/D_c$ which are written down in Table 4 and shown in Fig 6 (lines 342-376 = 34 lines). Then they provide 8 lines stating that aerosol is least internally mixed in winter with 1 very short sentence saying that that is potentially from local pollution. This is then followed by a general explanation how their type of CCN measurements probe different parts of the particle size distribution (7 lines). This information should have been provided at the start of this chapter as it applies to all values presented here. Next are two paragraphs (9 and 10 lines each) briefly describing the results presented in Fig 7 and Table 5, fortunately in a general way and without providing each individual number.

   After these lengthy descriptions, I ended up with the following questions:

   a) why is $k_{CCN}$ smaller than $k_{chem}$?

   b) why is the aerosol more internally mixed in summer?

   c) what is that local pollution that is a plausible explanation and why is it only relevant in winter?

   d) why is $k_{CCN}$ more sensitive to $D_p$ in winter?

   e) my answer to (c) is: because the winter aerosol is less internally mixed, especially for small particles. Then I wonder: Why are the small particles less externally mixed?

   f) and following that: why are the trends in k ~ $D_p$ and $(D_{75}-D_{25})/D_c$ ~$D_p$ different? Both describe the degree of externally mixing.

   None of these aspect are picked up anywhere. Yes, the reader can come up with their own answers but that is not the point of such a manuscript.

2) The manuscript feels massive in its current state. In addition to the lengthy descriptions (which need to be shortened), there are 11 Figures and 5 Tables. The author should consider which of these are really needed to follow the main story line in their manuscript. Currently, the amount of information is a bit distracting from the main points. In my opinion, the following things can go to the Supplement Information without compromising the content of the manuscript: Fig 1, Fig 2, Fig3a, Fig 5, either Fig 8 or Fig 9, Fig 11, Fig A1, and Table 3 and 5. Table 2 can go to SI as well if the authors follow my suggestion in Specific Comment 15. This together with trimming the detailed descriptions, will make this paper a lot more reader friendly and highlight the interesting scientific findings while still providing all the details in the SI.

3) The authors investigate the mixing state of the aerosol particles. While the definition of the term "internally mixed" is clear (all particles at any given size have the same composition, Seinfeld&Pandis), there are two aspects of external mixtures. It can mean that at a given size, particles can have different compositions. This shows up in size-resolved CCN measurements as a broadening of the AF vs $D_p$ function ($(D_{75}-D_{25})/D_C$ = $IQR/D_C$, IQR: interquartile range). On the other hand, externally mixed can also mean that the composition varies at different sizes (e.g., large dust particles and small SOA particles from NPF). If the CCN measurements probe different parts of the size distribution, the $k_{ccn}$ values will be different according to the size dependent particle composition. The width of the AF vs $D_p$ function can also be affected in this case. In their investigation, the authors only use the $IQR/D_C$ values to infer information about the mixing state and did not link this information with the apparent size dependence of $k_{CCN}$ and the discrepancy to $k_{chem}$. The manuscript will benefit from adding some discussion of these two manifestations of external mixing (see also Specific Comment 29)

4) The authors claim that their findings (i.e., their parameterisation of k ~ $D_P$) can be applied to other locations with a "similar aerosol background". But they do not qualify what they mean by that. How would one determine the similarity? From CCN measurements? Or can composition measurements be used? Fig 10 suggests that Melpitz is more similar to Shanghai than Barbados or the Amazon. So, the categories rural/urban may not always be helpful to identify "similar" behaviour. Discussing what needs to be similar to enable the application of the k ~ $D_P$ parameterisation should be a section in the conclusions chapter.

**Specific comments**

1) line 56 ff: This sounds like Petters and Kreidenweis (2007) came up with a new formula instead of the Köhler theory. But they just parameterized the Raoult term with a single parameter to capture the water activity without needing to know anything about the dissolved compounds. Adjust the sentence to reflect that.

2) line 64: "…because of the reciprocal relationship between k and Dp^3" The word reciprocal is ambiguous in this context. But mainly, I disagree with this statement. The stronger impact comes from the different range in which k and $D_p$ vary and that $D_p$ is cubic. The maximum variation for k is ~0.05 – 0.8. That is a change of factor 16. But the dominant particle size may change from 30 nm to 200 nm. 200^3/30^3 =296.

3) line 156: 40% RH seems still quite high for size selected CCN measurements. For aerosols with a high hygroscopicity, there may be considerable amounts of water in the selected particles which would then create a noticeable bias in the measured $D_C$/$SS_C$ pairs.

4) line 162: What is meant by "near-PM1"?

5) line 177: The total SS cycle is 2.5h. Were all SS steps of the same length? Was all data used or were the first x min omitted while SS stabilised after change? Was there one Dp scan or multiple during one SS setting?

6) line 184: Rose et al. (2008) point out that you need to state which parameterisation/model was used to derive the theoretical SS/Dcrit values. They provide many in their study. Which one was used here? Also, what does "regularly" mean? Every month?

7) line 207ff: What was the size range of the DMA coupled to the CCN-C? Later in Line 391, the authors state that the limit was 40-200nm for $D_p$. This information needs to go into the Methods section. Was the D-MPSS compared with the DMA-CPC to check that the particle losses in each instrument are comparable? Also, why was the upper size limited in the DMA? Was it due to HV? Or was there a physical limit (Impactor, too many bends in line?). I.e., are you sure that the large particles did make it through the DMA in the same way as they reached the D-MPSS?

8) Line 210f/ Eq 1: What is meant with "a and b are the lower and upper limits for calculating Dc"? To my understanding, 'b' is the height of the upper plateau of the sigmoidal function and 'a' is the offset from 0 in the y direction. If b is lower than 1 (after accounting for any instrument discrepancies), this indicates the presence of particles that do not activate (e.g., pure black carbon or dust). As the mixing state is investigated later on, it would be good to know if the b value was lower than 1 during the times with higher external mixing.

9) Line 227 ff: As stated in Petters and Kreidenweis (2007), Eq 2 is an approximation which is only valid if the solution is very dilute at the point of activation. Is this approximation true for the analysed data in this study? I.e. are the kappa values high enough (Petters and Kreidenweis (2007) suggest k>0.2 as the threshold)?

10) Line 230: The authors assume that the surface tension of the droplet is constant, and k is the only unconstrained parameter in their equation. If the surface tension does vary, the effect in measured aerosol particle activation behaviour will then be accounted for by a different k value. Then k is not just representing the hygroscopicity of the aerosol but also the effect of any surface tension change and may differ more from $k_{chem}$ which can only take the composition into account. Since the debate about the importance of surface tension changes and the connected bulk/surface partitioning is still ongoing (e.g.: Ovadnevaite et al. (2017); Vepsäläinen et al. (2021)), I recommend that the authors simply include a brief sentence stating that k may pick up surface tension changes.

11) Line 242: Why was k=0.1 used for organics? In your introduction, you show that $k_{org}$ can vary enough to be relevant for $N_{CCN}$ predictions (e.g., in the range of 0.05 – 0.15 which can be cause by changes in organic composition). Ambient and chamber measurements have shown that $k_{org}$ is a function of the degree of oxidation (e.g., expressed with O:C). What impact will it have if you use such a parameterisation for $k_{org}$ (e.g., $k_{org} = 0.18*O:C + 0.03$ given in (Lambe et al., 2011))

12) Line 245 "When k is given…" For equation 4 to work, only a value for $D_C$ is needed. Size resolved CCN counter measurements provide SS/$D_C$ pairs and Eq 4 can be used directly. If no $D_C$ values are available, $D_C$ can be calculated for any SS from a given k.

13) Average values in Fig 3, 4, 5, and 7: The authors show average values of $N_{CCN}$, AR, etc. and use the standard deviation to indicate the spread of the data. Would it not be better to use the interquartile range for that purpose? So, indicting the mean (or median) value with a marker and then use a error bar/shaded area to indicate the Q25 and Q75 range?

14) There are multiple issues with Fig 3.

   a. The Figure caption does not contain the information about the black markers. in Fig 3a. There is no description of the right-hand axis in Fig 3a.

   b. Neither is there any information about the averaging which seems to be different for the size distribution (two values per month?) and the AR values.

   c. The panels in Fig 3a are so tiny that it is very hard to see, e.g., the differences between SS=0.5% and 0.7%.

   d. The error bars are outside of the y-axis range for some plots in Fig 3b and c.

   e. For me, Fig 3a added very little to understand the description provided in Chapter 3.1. Also, for the overall interpretation of the data, the CCN number size distribution is not as relevant and could easily move to the SI. See next comment for other changes proposed for Fig 3.

15) The description of the AR and $N_{CCN}$ values and trends is very hard to follow in Chapter 3.1. This is cause by the excessive details in the description and the choice of visualisation of the data.

    a.    It is good that the authors compare their values to so many other studies. But due to each study having a different SS range, it is difficult to really understand how the data compares. Table 2 does not provide much more insights. But a simple Figure does (Fig R1 below):

[Figure]

Figure R1: $N_{CCN}$ from this study and studies cited in chapter 3.1.

    b.    They authors look into the seasonal trends of $N_{CCN}$ and AR and again I got lost in all the numbers and the provided visualisations do not help. Looking at the plots in Fig 3b and c, the seasonal behaviour of AR and $N_{CCN}$ look indeed "similar". But when I visualised the given values from lines 273 – 280 as a function of season (see Fig R2 below), I realised that there are some interesting differences. Going from 0.1% to 0.7%, the minimum of the $N_{CCN}$ shifts from autumn to summer (Fig R2 top) while the trends in AR with a minimum in summer are the same for these SS (Fig R 2 bottom. Why is that the case? To me, it is clear that this must be connected to changes in the PNSD. But before the authors get to that, they first dive into the details of the $N_{CCN}$(SS) and AR(SS) relationships. Here already the connection between the set SS and the size range that is probed is important. But that is not mentioned until lines 385-391 (see Specific Comment 28)

[Figure]

Figure R2: Seasonal trends of $N_{CCN}$ (Top) and AR (bottom). Values for SS=0.1% are in blue and use the left-hand y-axis, while the values for SS=0.7% are indicated in orange and use the right-hand axis.

c. I do not understand what the authors mean with "CCN number size distribution gradually peaks in summer". Whatever is meant by that, how is that connected to the seasonal trends in $N_{CCN}$ and AR, especially the summer minimums of AR and $N_{CCN}$(SS=0.1%)?

d. Why are AR and $N_{CCN}$ more sensitive to SS in summer than in winter?

My recommendation for cleaning up this chapter is to change Fig 3 by moving Fig 3a into the SI. Instead, provide a larger version of the "all data set" $N_{CCN}$ vs SS which includes the values for the other studies as shown here in Fig R1. This will make the naming of all the number for the previous studies obsolete and Table 2 can also be moved to SI. If the authors keep the description and deepen the discussion of the seasonal trends, I strongly recommend adding something like Fig R2 either to Fig 3 or into the SI to help the reader follow the descriptions. Shifting the explanations

about the PNSD to follow the description of these trends will then feel more natural. The next few Specific Comment are also related to improving this chapter.

16) line 282ff: The error function used for fitting seems to have 4 free parameters. Each data set that is fitted has 5 values. Some people might say that that is a problem. Or was the original data fitted and not the (seasonal) averages? This is not mentioned in the text.

However, since the error function fits are not used anywhere in the manuscript other than stating that the function fits slightly better, this could be reduced to stating that the fit was also performed with an error function and the fitted parameters and curves are in the SI.

17) line 298: This information about what the CCN number size distribution represents should have appeared at the end of the methods section when EQ 4 is introduced. The schematic diagram is using the assumption that there are no non-CCN active particles at $D_p > D_c$ (see also Specific Comment 8). Is that assumption reasonable? You should be able to estimate that from checking the plateau values in the AF vs Dp plots. How close are these values to 1.0 (after you accounted for different losses in the two instruments/sampling lines)?

18) line 314f: Again, the phrase "CCN number size distribution gradually peaks in summer" occurs without clarifying what is peaking.

19) Lines 306-317: It seems plausible that the presence of a large number of small particles explains the minimum of AR and maximum of $N_{CCN}(SS \leq 0.5\%)$ during summer. But why was the influence of the change in hygroscopicity omitted? Winter and Spring have much higher $k_{CCN}$ values (at least for the larger particles) which will also contribute to the high AR and $N_{CCN}$ during that season.

20) Fig 4 should be improved. The black markers for $k_{chem}$ are difficult to see in front of the dark blue background from NO3. Why does the $k_{chem}$ axis start at 0.2 and not 0? I do not like how the GMD and $N_{aero}$ plot are put over the PNSD graph. Overlaying two panels over the PNSD is not straight forward to read. The intuitive interpretation is that the two sets of black markers are both on an axis extending the full hight of the PNSD plot. The March and February markers are only partially visible.

21) Lines 323-329: While these correlations are interesting, Fig 5 could also be in the SI. Especially, since there is no interpretation of the meaning of the k ~ fX correlations currently. Yet another example where the reader is left to come up with their own conclusions about an interesting observation. Here is my take:

For understanding the relationship between $k_{chem}$ and the individual composition groups, it is important to realise that these groups do not act in the same way in Eq 3. The influence of Org is direct. $k_{org}$ is smaller than $k_{inorg}$. Thus, higher $f_{org}$ means lower $k_{chem}$. But with SO4, NH4, NO3 it is more complicated because they are coupled through the ion balance. The absolute amount of SO4 and NH4 seems pretty stable. But the NO3 amount changes a lot between the seasons. The presence of NO3 shifts the salts from mostly (NH4)2SO4 towards NH4NO3 and NH4HSO4 or even H2SO4. k values are very similar between (NH4)2SO4, NH4HSO4, and NH4NO3, but $k_{H2SO4}$ is much higher. Thus, an increase in NO3

can have a dual impact on k for this data set. The increase in NO3 adds a higher proportion of salt and also increases the k of SO4.

So, if fSO4 decreases because more Org is present, k decreases. If fSO4 decreases because more NO3 is present, k increase. As these two trends are opposite, the correlation of fSO4 and k will be poor. Since this behaviour is opposite the usually assumed "fSO4 increase leads to k increase", it is worth discussing. Also, why is NO3 increasing in winter and spring? Can it really be just the change in ambient T? Could it be linked to the "local pollution" that is mentioned without any explanation in line 384? And to link this to the bigger picture: could the balance between NO3 and the other aerosol constituents be an important factor when comparing aerosol activation behaviour in different regions?

22) Lines 330-337: The changes in the width of the CCN number size distribution are not just related to the hygroscopicity (i.e., the $D_C$ values). The shape of the PNSD plays an equally important role. The $D_C$ value (i.e. the hygroscopicity) determines the edge at the smaller end of the CCN number size distribution. But the shape of the distribution at larger sizes depends more on the shape of the PNSD. I.e. with an identical $k$ / $D_C$ the winter CCN number size distribution will be wider because of the shift to larger sizes in the PNSD. The different shape of the PNSD may also help to explain the stronger sensitivity of $N_{CCN}$ to SS during summer. The PNSD is probably steeper in the 40-150nm size range. Thus, a small shift in $D_c$ will change the $N_{CCN}$ much more than in winter where the PNSD look broader.

23) Chapter 3.2: Throughout the manuscript the authors write as if $k_{CCN}$ and $D_C$ are independently measured variables, while really k is calculated from the measured $SS_c$/$D_C$ pairs. Describing both the $D_c$ and k trends in details is thus redundant. As the authors want to compare the hygroscopicity to the composition, it is sufficient to present the $D_C$ values only in the Table and figure. If a reader is interested in the exact values for $k_{CCN}$ or $D_C$, a Table/Figure is anyway much faster than trying to find the relevant values in the long text.Then focus on the $k_{CCN}$ trends and compare them with the $k_{chem}$ information (see also next Specific Comment). This will make the actual discussion/ interpretation/comparison much more readable. To facilitate the $k_{CCN}$ / $k_{chem}$ comparison, add the $k_{chem}$ values to Fig 6a.

24) Lines 347f: I disagree with the statement that the seasonal variation of $k_{chem}$ and $k_{ccn}$ are similar for all SS. Adding the $k_{chem}$ values to Fig 6a would make this clearer. The season trend in $k_{ccn}$ is much weaker for small particles and I would claim that $k_{CCN}$(SS=0.7%) does not display any trend if its values are 0.19, 0.20, or 0.21, each with a standard deviation of 0.1. This is already a strong sign of a more externally mixed aerosol population during winter and spring. It also shows that $k_{chem}$ is not representative for the smaller particles at this location. See also Specific Comment 28 and 29.

25) Line 339f: k also varies with composition!

26) Line 379: "non-urban locations" Is the point here that the particles are away from strong localised sources? Or is this about the type of aerosol (e.g., anthropogenic vs biogenic)?

27) line 384: This is another example where the manuscript has a lot of description and very little discussion. What is that local pollution? How does it explain the observations? Would this local

pollution have varying effects depending on the particle size? Why is this pollution more important in Winter?

28) lines 385-391: This important explanation needs to come much earlier in the text since it is not only relevant for the IQR/$D_c$ vs $D_P$ relationship but also for all discussion related to the size resolved CCN measurements. Especially, when comparing with $k_{chem}$. As the ACSM is sensitive to mass (and not number) concentration, the bulk composition is dominated by the contribution of the larger particles. Thus, $k_{chem}$ may not be representative for the smaller particles (higher SS) which is exactly what Fig 6 shows.

29) Lines 392-410: Here the authors again just describe the observations without making the interesting connections. The authors do not draw the connection between the change in $D_P$ dependence of $k_{CCN}$ and the change in mixing state (see also Main Comment 3). The higher sensitivity of $k_{CCN}$ to $D_P$ during spring and winter is not an intrinsic property, but it is the direct result of a more externally mixed aerosol population with size dependent composition. In spring and winter, it is more important which part of the size distribution is probed by the CCN measurement because the particle composition changes more with size than in the other seasons. Now, the authors should think about why this is the case? What causes this stronger size dependence of the particle composition? And what does it mean that the IQR/$D_C$ vs $D_p$ relationship is much shallower in spring than in winter?

30) Chapter 3.3: The introduction of the prediction methods is currently a little bit confusing and needs improvement. From the text, I did not understand what the main difference is between the two categories. I eventually work out that the schemes in the first category can be used for data obtained from polydisperse CCN measurements when only $N_{CCN}$ is measured while the second category is based on using some sort of k value to calculate $D_C$. In addition, readability could be enhanced by labelling the schemes using the categories, e.g., N1, N2, and K1, K2, K3.

31) lines 439f: RD is a single value for each case in Fig 8. But |predicted $N_{CCN}$ - measured $N_{CCN}$ |/ measured $N_{CCN}$ provides a number for each measurement point. I guess these values were summed? Please, correct this equation and write it as its own as a proper equation and not "in-line"

32) Lines 443-454 summarise the prediction quality of the 5 schemes. This section is good. But then that is followed by yet another very detailed description of numbers that are presented in Fig 8 and 9 (over 3 and a half pages!). This is extremely tedious to read and again the important conclusions are buried under mountains of numbers. The authors need to trim this section.

Fig 8 and 9 show the same information simply from a different angle. They need to decide which of the figures works better and put the other in the SI. Instead of providing so many numbers for each scheme to say again that the prediction is better/worse, they should focus on the main improvements and features of the schemes which lead to the better worse prediction. E.g., scheme 1 calculates 1 $N_{CCN}$ value for each SS. Thus, the spread depicted by the boxplot in Fig 9a simply reflects the standard deviation of the measurements as shown in Fig 3c (or rather the Interquartile range). For the category 2 schemes, the point is how well the parameterised k describes the measured $k_{CCN}$ value. This then

explains why some seasons are predicted better than others (i.e. if the measured $k_{CCN}$ are closer to the value set in the scheme).

33) lines 510 – 528 provides a good summary of the performance of the schemes and links the power law values to other observations. But what does it mean that the values for Melpitz are similar to some stations (even urban ones) and not to others (see also Main Comment 4). Either here – or better in the conclusions – this should be discussed, and the authors should at least speculate what may be causing the similar behaviour at such different sites.

34) Table 3 is very difficult to read. It is next to impossible to compare the parameters as each entry is spread over multiple lines. How about stating the equations in the Table cation and then providing only the parameters and $R^2$ values in the table. If this table stays in the main text, the error function values should be moved to the SI (see Specific Comment 16).

35) The authors claim that scheme 5 (using the power law $k(D_P)$ approximation) provides an improved prediction of $N_{CCN}$. This is true when compared to schemes 1 and 2. But how much does that improvement really matter when looking at schemes 3-5? From 4 to 5, the slope decreases 0.1 on average. So, the 10% overestimation is reduced. How much will that impact, e.g., the calculation of radiative forcing or prediction of precipitation in a climate model? Is that worth the effort? Some people may argue that operating an ACSM from which $k_{chem}$ can be derived, is more feasible in a measurement station than conducting size resolved CCN measurements which are needed to obtain the k(Dp) relationship.

36) The k value used in scheme 3 is clearly too high. Have the authors tried to run this scheme using the average $k_{CCN}$ value for their data set? How "good" is scheme 3 then?

37) The Conclusions chapter is simply a summary of the previous chapters, repeating many of the numbers that were already stated. These are not "conclusions" as in interpretations or putting their findings into context. There are many things the authors bring up in this chapter. These are a few ideas that spring to my mind (some are already mentioned in other Specific Comments):

    a. How much their improved $N_{CCN}$ prediction may improve modelling results?

    b. How much would using the values from the "wrong" season affect $N_{CCN}$ predictions? or from a wrong location (E.g. using the Budapest values for the Melpitz data set)

    c. If the $k \sim D_P$ prediction works so well, do we really need continuous CCN measurements? Wouldn't it be enough to determine the representative $k \sim D_P$ fit for a few representative locations?

    d. Or playing devil's advocate: Since the $k_{chem}$ based $N_{CCN}$ prediction is much better than the ones based on $N_{CCN} \sim SS$ or $AR \sim SS$, wouldn't it be better to improve composition measurements?

    e. regarding the mixing state: Why is the mixing state different between seasons?

    f. Why is $k(DP)$ and $IQR/D_C(D_p)$ different between the seasons?

38) line 565: these things are also linked to the highest kCCN and the widest spread in kCCN (i.e., least internally mixed)

**Language:**

General: In multiple locations, main clauses are attached with ";" to each other. While this is grammatically possible, it decreases readability by creating "monster sentences". Simply use a full stop and start the second main clause. Examples: line 28ff: second sentence starts at "the seasonal mean activation ratio…"

line 15 "measurements on aerosol particle activation" -> of

line 20 "improving predictions": predictions of what?

line 29 "twice higher" -> either "twice as high as" or "two times higher than"

line 35: "the power law function" sounds as if this is a specific function with the name 'power law' change to "a power law function"

line 44 "activated cloud droplets" -> remove activated. The particles get activated to grow to cloud droplets.

line 72f "should be underlined" -> no, it should not be underlined (unless you speak German ;-). change to "should be emphasised"

line 137 "mixing state degree" -> sounds weird either use "degree of mixing" or "mixing state"

line 149f "can be found in for example, Poulain et al 2020" -> "can be found, for example, in Poulain et al 2020.

line 153: "Figure 1 demonstrates" -> it is not the Figure that does something. Better use "Figure 1 shows/depicts"

line 160: "within the diameter ranging from 5 to 800nm" -> "with a diameter range of 5 – 800 nm"

line 170f "respectively pass through" -> "respectively" cannot be used like that. This is also an example for a ";" monster sentence. Simply start a new sentence. "… monodisperse particle fraction. After the DMA, the flow was split to pass through a CPC […] and a CCN counter […]."

line 200 "was firstly corrected" -> was first corrected

line 203: "thus they are falsely selected in the DMA" -> they are selected in the absolute correct way. It is the assigned diameter that is incorrect. Simply remove this phrase.

line 204 "For this was corrected" -> "To correct for this, the fraction of multiple charged particles […] was subtracted […]"

line 216 "rather than an intermittent mutation" -> do you mean "rather than displaying (?) an intermittent mutation"?

line 235 "determined" -> determined feels a bit strong here. Maybe better "derived" since this is a approximation of the true k value?

line 278 "…gradually peaks in summer…" -> I do not know what "gradually peaks" means in this context

line 282f "the power-law and the error function" -> should be "a".

line 285 "because of more parameters" -> "due to the higher number of parameters".

line 298 "CCN number size distribution" -> missing "The"

line 382: What is meant by "aerosol cluster"?

line 415: "two categories of NCCN prediction approach" -> "approaches" or better "can be divided into two categories"

line 417 and later "category approach" -> only "category" without approach

line 444f "provide rough estimates on account of the pretty high RD" weird. RD is not causing the rough estimate it is the consequence. Better "provide rough estimate which is reflected in the high RD"

line 454: "…Figure 9 further evaluates the model…"It is not the Figure that evaluates the models.

line 458 "results are much uncertain" -> "the results have a high uncertainty"

line 466f "the prediction results remain a high uncertainty" -> ??? "the uncertainty of the prediction results remain high"???

line 475: "$N_{CCN}$ is overestimated at assuming a constant k" -> "when assuming"

line 478f: "the largest median overestimation reaches to 30%" -> no "to"

line 485f: "the 3rd  scheme has better predictions on $N_{CCN}$" ->"provides better predictions of $N_{CCN}$"

line 510: "gradually changes" really? I would not call the big improvement from scheme 1 to 2 to 3 "gradual". For the changes going from schemes 3-4-5, gradual is the correct term.

**References**

Lambe, A. T., Onasch, T. B., Massoli, P., Croasdale, D. R., Wright, J. P., Ahern, A. T., Williams, L. R., Worsnop, D. R., Brune, W. H. and Davidovits, P.: Laboratory studies of the chemical composition and cloud condensation nuclei (CCN) activity of secondary organic aerosol (SOA) and oxidized primary organic aerosol (OPOA), Atmos. Chem. Phys., 11(17), 8913–8928, doi:10.5194/acp-11-8913-2011, 2011.

Ovadnevaite, J., Zuend, A., Laaksonen, A., Sanchez, K. J., Roberts, G., Ceburnis, D., Decesari, S., Rinaldi, M., Hodas, N., Facchini, M. C., Seinfeld, J. H. and O' Dowd, C.: Surface tension prevails over solute effect in organic-influenced cloud droplet activation, Nature, 546(7660), 637–641, doi:10.1038/nature22806, 2017.

Petters, M. D. and Kreidenweis, S. M.: A single parameter representation of hygroscopic growth and cloud condensation nucleus activity, Atmos. Chem. Phys., 7(8), 1961–1971, doi:10.5194/acp-7-1961-2007, 2007.

Rose, D., Gunthe, S. S., Mikhailov, E., Frank, G. P., Dusek, U., Andreae, M. O. and Pöschl, U.: Calibration and measurement uncertainties of a continuous-flow cloud condensation nuclei counter (DMT-CCNC): CCN activation of ammonium sulfate and sodium chloride aerosol particles in theory and experiment, Atmos. Chem. Phys., 8(5), 1153–1179, doi:10.5194/acp-8-1153-2008, 2008.

Vepsäläinen, S., Calderón, S., Malila, J. and Prisle, N.: Droplet activation of moderately surface active organic aerosol predicted with six approaches to surface activity, Atmos. Chem. Phys., 1–25, doi:10.5194/acp-2021-561, 2021.

Seinfeld, J. H., and Pandis, S. N.: Atmospheric chemistry and physics: From air pollution to climate change, Hoboken: John Wiley and Sons, 2016.

---

## Author Comment (AC2)

**Review report on Wang et al. 2022, ACPD**

The authors present a very thorough description of a 4 year-long size resolved CCN measurement data set at a central European rural background station. Due to the long dataset, the authors can analyse seasonal trends and test multiple approaches to predict the number of activated particles NCCN. This type of data and analysis approach is very valuable for the atmospheric science community. But unfortunately, the authors mostly simply present and describe the measurements and leave the reader wondering what causes these trends and what they may mean in the big picture (e.g., for climate modelling). Adding these necessary discussions constitutes major revisions. I recommend publication after these are done, and the remaining issues listed below have been addressed.

Response: Many thanks for your kindly comments. Those detailed comments are valuable and very helpful for improving our manuscript. We have revised our manuscript following your suggestions and answered your questions as follows.

**Main Comments**

1) Throughout the manuscript, the balance between describing the measured values and interpreting them is strongly tilted towards the descriptive side. This data set contains a lot of interesting scientific observations. But in most places the reader is left to wonder what they mean. And if interpretation is provided, it is hidden after all those detailed numbers which are already visualised in the Figures/Tables. By describing every detail, the authors lose the attention of the reader for the really interesting parts. Below is one example to illustrate this issue.

In Chapter 3.2, the authors describe and name literally all values of k, $D_C$, and (D75-D25)/$D_C$ which are written down in Table 4 and shown in Fig 6 (lines 342-376 = 34 lines). Then they provide 8 lines stating that aerosol is least internally mixed in winter with 1 very short sentence saying that that is potentially from local pollution. This is then followed by a general explanation how their type of CCN measurements probe different parts of the particle size distribution (7 lines). This information should have been provided at the start of this chapter as it applies to all values presented here. Next are two paragraphs (9 and 10 lines each) briefly describing the results presented in Fig 7 and Table 5, fortunately in a general way and without providing each individual number.

Response: Thanks for your comment. Exactly, the original manuscript seems to be more

describing the measured values than interpreting them. In the revision, we have reduced descriptive statements and increased the discussion of causes. For instance, Chapter 3.2 has been revised as follows:

[revised manuscript text omitted]

After these lengthy descriptions, I ended up with the following questions:

a)                    why is kCCN smaller than kchem?

Response: Essentially, the relationship between $\kappa_{CCN}$ and $SS$ is determined by the $\kappa_{CCN}$ vs. $D_p$ relationship. The $\kappa_{CCN}$ at $SS$ of 0.1% and 0.7% correspond to the median $D_c$ (i.e., $D_p$) of 176 nm and 54 nm, respectively. As the ACSM is sensitive to particle mass rather than number concentration, the bulk composition is dominated by the contribution of the larger particles. In the median volume size distribution of particle, the peak diameter was at ~300 nm (Poulain et al., 2020). Thus, $\kappa_{chem}$ may be representative for the larger particles (i.e., at lower $SS$ conditions) rather than for the smaller particles (i.e., at higher

*SS* conditions). Owing to the positive correlation between $\kappa$ and $D_p$ (Figure 6a), the $\kappa_{chem}$ representing for the larger particles could be greater than the $\kappa_{CCN}$ for the smaller particles.

[Figure]

Figure 6. (a) Relationship between the hygroscopicity factor calculated from monodisperse CCN measurements ($\kappa_{CCN}$) and particle diameter ($D_p$), and (b) degree of external mixture (($D_{75} - D_{25}$)/$D_c$) vs. $D_p$ at each season. The definitions of $D_{75}$ and $D_{25}$ are the $D_p$ at which 75% and 25% of the particles are activated at the given *SS*, respectively. Red lines are power-law fits. Dots represent the median values. Shaded areas represent the values in the range from 25th to 75th percent.

The text in lines 323 to 332 was changed as follows:

"Essentially, the relationship between $\kappa_{CCN}$ and *SS* is determined by the $\kappa_{CCN}$ vs. $D_p$ relationship. The $\kappa_{CCN}$ at *SS* of 0.1% and 0.7% correspond to the median $D_c$ (i.e., $D_p$) of 176 and 54 nm, respectively. As the ACSM is sensitive to particle mass rather than number concentration, the bulk composition is dominated by the contribution of the larger particles. In the median volume size distribution of particle, the peak diameter was at ~300 nm (Poulain et al., 2020). Thus, $\kappa_{chem}$ may be representative for the larger particles rather than for the smaller particles. Owing to the positive correlation between $\kappa$ and $D_p$ (Figure 6a), the $\kappa_{chem}$ representing for the larger particles could be greater than the $\kappa_{CCN}$ for the smaller particles."

b)                    why is the aerosol more internally mixed in summer?

Response: Here we will suspect two reasons: 1. Less contribution from anthropogenic emissions. 2. In summer, atmospheric chemistry leads to a faster aging process as well as SOA formation, which certainly contribute to make particles more internally mixed. Change in OA composition can be found in Crippa et al. (2014, doi.org/10.5194/acp-14-6159-2014), Poulain et al. (2014, doi:10.5194/acp-14-10145-2014), and Chen et al. (2022, doi.org/10.1016/j.envint.2022.107325).

The following text was added to the manuscript (Lines 345 to 349):

"In summer, the less contribution from anthropogenic emissions and the faster aging process as well as SOA formation caused by atmospheric chemistry certainly contribute to make particles more internally mixed. Changes in organic aerosol (OA) composition can be found in Crippa et al. (2014), Poulain et al. (2014), and Chen et al. (2022)."

c)                    what is that local pollution that is a plausible explanation and why is it only relevant in winter?

Response: In cold seasons, the local pollution (100 km around) is dominated by liquid fuel, biomass, and coal combustions mostly for house heating (e.g., van Pinxteren et al., 2016, DOI: 10.1039/c5fd00228a). Local pollution is mostly related to house heating, which is limited to cold season. During winter long-range transport from the eastern wind bring to the station continental air masses which are strongly influence by anthropogenic emissions (in opposition to western marine air masses). These particles are a mixture of different anthropogenic sources emitted all along the transport as well as including some local/regional sources (most house heating from different methods) all of them at different aging state making the overall particles more externally mixed.

The following text was added to the manuscript (Lines 349 to 357):

"In cold seasons, the local pollution (100 km around) is dominated by liquid fuel, biomass, and coal combustions mostly for house heating (van Pinxteren et al., 2016). During winter long-range transport from the eastern wind bring to the station continental air masses which are strongly influence by anthropogenic emissions (in opposition to western marine air masses). These particles are a mixture of different anthropogenic sources emitted all along the transport as well as including some local and regional sources (most house heating). All of them at different aging state cause the overall particles more externally mixed."

d)                    why is $k_{CCN}$ more sensitive to $D_p$ in winter?

Response: In winter we have a mixture between anthropogenic sources and aged particles leading to a size dependent chemical composition (e.g., van Pinxteren et al.,

2016, DOI: 10.1039/c5fd00228a). In summer, the anthropogenic emissions linked to house heating a strongly reduce which affect the smaller particles, and the dominant < 100 nm particles is associated to NPF and SOA formation. NPF is a complex process which is depending on the availability of condensing material (H2SO4, and organic), as well as pre-existing particles (coagulation and condensation sink parameters). Therefore, same condensing material on the gas phase can either condense on pre-existing particles (usually larger than 100 nm and then detected by ACSM) or lead to NPF formation. A direct consequence of it, is a probable smaller effect of the size dependent chemical composition of the particles. This might explain why kCCN at SS 0.1 and 0.7 % are closer in summer.

The following text was added to the manuscript (Lines 366 to 376):

"Compared to the cold seasons, the anthropogenic emissions linked to house heating strongly reduce in summer which affect the smaller particles, and the dominant small particles ($D_p$ < 100 nm) are associated to NPF and the SOA formation. NPF is a complex process which depends on the availability of condensing material ($H_2SO_4$ and organic), as well as pre-existing particles (coagulation and condensation sink parameters). Therefore, same condensing material on the gas phase can either condense on pre-existing particles (usually larger than 100 nm and then detected by ACSM) or lead to NPF formation. A direct consequence of it is a probable smaller effect of the size dependent chemical composition of the particles. This might explain why $\kappa_{CCN}$ at *SS* of 0.1% and 0.7 % are closer, i.e., the weaker sensitive of $\kappa_{CCN}$ to $D_p$ in summer."

e)                     my answer to (c) is: because the winter aerosol is less internally mixed, especially for small particles. Then I wonder: Why are the small particles less externally mixed?

Response: In winter, small particles must be related to local anthropogenic emissions including house heating which is a combination of liquid fuel, biomass burning and coal burning leading to more externally small particles. While in summer, local emissions reduce and the most important sources of small particles in associated to new particle formation and growth, leading to more internally small particles.

The text has added to the manuscript as shown in Responses (b) and (c).

f)                     and following that: why are the trends in k ~ Dp and (D75-D25)/Dc ~Dp different? Both describe the degree of externally mixing.

Response: k vs Dp is the difference in composition from small to large particles and

(D75-D25)/Dc vs Dp is degree of external mixture at one size. We have monodisperse CCN measurements - meaning we can explore the mixing state of particles of a fixed size. In other words, external mixing would mean particle with the same size can have different chemical composition. In our manuscript, we used the (D75-D25)/DC to quantify the mixing state at a fixed size. External mixtures mean that at a given size, particles can have different compositions rather than composition varies at different sizes. We clarified that in the text (lines 224 to 228) as follows:

"Internal mixture implies that all particles with any given dry size have equal $\kappa$ with $(D_{75} - D_{25})/D_c = 0$, whereas a distribution of different $\kappa$ at a given particle size can be observed for externally mixed aerosol with higher $(D_{75} - D_{25})/D_c$ values. Note that the particle composition varying at different sizes is not defined as external mixing in this study."

As Dp increases, k increases and (D75-D25)/Dc decrease. It means that the large particles have relatively high k and high degree of internal mixing, which could be reasonable due to the aging process.

None of these aspect are picked up anywhere. Yes, the reader can come up with their own answers but that is not the point of such a manuscript.

Response: Thanks for your questions. They have been addressed in the revision and the answers are as above.

2)      The manuscript feels massive in its current state. In addition to the lengthy descriptions (which need to be shortened), there are 11 Figures and 5 Tables. The author should consider which of these are really needed to follow the main story line in their manuscript. Currently, the amount of information is a bit distracting from the main points. In my opinion, the following things can go to the Supplement Information without compromising the content of the manuscript: Fig 1, Fig 2, Fig3a, Fig 5, either Fig 8 or Fig 9, Fig 11, Fig A1, and Table 3 and 5. Table 2 can go to SI as well if the authors follow my suggestion in Specific Comment 15. This together with trimming the detailed descriptions, will make this paper a lot more reader friendly and highlight the interesting scientific findings while still providing all the details in the SI.

Response: Thanks for your suggestion. We have moved the original Fig 2, Fig 3a, Fig 5, Fig 9, Fig 11, Fig A1, Table 2, Table 3, and Table 5 to supporting information. The descriptive content has been shortened in the revision. We would like to keep the Fig 1

in the manuscript to make experiment setting easy for reader to understand. Currently, there are 8 Figures and 3 Tables in the manuscript and 8 Figures and 2 Tables in SI. The word number of the main text has been reduced from 7838 to 6652. The page number of the manuscript has been reduced from 56 to 47.

3)      The authors investigate the mixing state of the aerosol particles. While the definition of the term "internally mixed" is clear (all particles at any given size have the same composition, Seinfeld & Pandis), there are two aspects of external mixtures. It can mean that at a given size, particles can have different compositions. This shows up in size-resolved CCN measurements as a broadening of the AF vs $D_p$ function ((D75-D25)/DC = IQR/DC, IQR: interquartile range). On the other hand, externally mixed can also mean that the composition varies at different sizes (e.g., large dust particles and small SOA particles from NPF). If the CCN measurements probe different parts of the size distribution, the $k_{ccn}$ values will be different according to the size dependent particle composition. The width of the AF vs $D_p$ function can also be affected in this case. In their investigation, the authors only use the IQR/DC values to infer information about the mixing state and did not link this information with the apparent size dependence of $k_{CCN}$ and the discrepancy to $k_{chem}$. The manuscript will benefit from adding some discussion of these two manifestations of external mixing (see also Specific Comment 29)

Response: Thanks for your comment. We have monodisperse CCN measurements - meaning we can explore the mixing state of particles of a fixed size. In other words, external mixing would mean particle with the same size can have different chemical composition. In our manuscript, we used the (D75-D25)/DC to quantify the mixing state at a fixed size, and the external mixtures mean that at a given size, particles can have different compositions rather than composition varies at different sizes. We clarified that in the text (lines 224 to 228) as follows:

"Internal mixture implies that all particles with any given dry size have equal $\kappa$ with $(D_{75} - D_{25})/D_c = 0$, whereas a distribution of different $\kappa$ at a given particle size can be observed for externally mixed aerosol with higher $(D_{75} - D_{25})/D_c$ values. Note that the particle composition varying at different sizes is not defined as external mixing in this study."

4)      The authors claim that their findings (i.e., their parameterisation of k ~ DP)

can be applied to other locations with a "similar aerosol background". But they do not qualify what they mean by that. How would one determine the similarity? From CCN measurements? Or can composition measurements be used? Fig 10 suggests that Melpitz is more similar to Shanghai than Barbados or the Amazon. So, the categories rural/urban may not always be helpful to identify "similar" behaviour. Discussing what needs to be similar to enable the application of the k ~ DP parameterisation should be a section in the conclusions chapter.

Response: As shown in Figure 8, the $\kappa$ and $D_p$ relationships measured at three rural stations (Melpitz, Xinken, and Vavihill) are similar. Thus, we concluded that the $\kappa$ - $D_p$ power-law fit presented in this study could apply to other rural regions. We notice that the power law for Shanghai was also close to what was found for Melpitz, but we only state that our power law can be used for other rural places rather than that is entirely different for all urban places studied. And that we can not answer what environmental properties cause the differences in kappa to Dp.

[Figure]

Figure 8. Relationships between the particle hygroscopicity factor ($\kappa$) and diameter ($D_p$) observed at different stations. Lines are the power-law fits of $\kappa$ vs. $D_p$.

We clarified that in the text (502 to 504 and 510 to 516) as follows:

"The $\kappa$ - $D_p$ power-law fit presented in this study could apply to other rural regions. However, it may cause considerable deviations for different aerosol background regions."

"Although the $\kappa$ - $D_p$ relationships are similar measured in rural stations, but when

comparing the different urban stations (e.g., shanghai vs. Budapest in Figure 8), these relationships are clearly different and the reasons for the difference are still unclear. Thus, long-term monodisperse CCN measurements are still needed not only to obtain the $\kappa$ - $D_p$ relationships for different regions and for different seasons, but furtherly investigate the reasons for the difference of the $\kappa$ - $D_p$ relationships measured at same type of regions."

**Specific comments**

1)        line 56 ff: This sounds like Petters and Kreidenweis (2007) came up with a new formula instead of the Köhler theory. But they just parameterized the Raoult term with a single parameter to capture the water activity without needing to know anything about the dissolved compounds. Adjust the sentence to reflect that.

Response: Thanks for the suggestion. Exactly, they just parameterized the Raoult term with a single parameter to capture the water activity rather than came up with a new formula instead of the Köhler theory. This sentence has been revised as follows.

"Petters and Kreidenweis (2007) parameterized the Raoult term with a single hygroscopicity factor $\kappa$ to capture the water activity without needing to know anything about the dissolved compounds" in lines 54 to 57.

2)        line 64: "…because of the reciprocal relationship between k and Dp^3" The word reciprocal is ambiguous in this context. But mainly, I disagree with this statement. The stronger impact comes from the different range in which k and $D_p$ vary and that $D_p$ is cubic. The maximum variation for k is ~0.05– 0.8. That is a change of factor 16. But the dominant particle size may change from 30 nm to 200 nm. 200^3/30^3 =296.

Response: Thanks for the comment. Exactly, the different range in which k and Dp range plays the critical role in evaluating the effect of particle size and composition on the CCN activation. This sentence has been revised as follows.

"In terms of a single aerosol particle, the actual particle size plays a more important role than the chemical composition for activation because of the different range in which $\kappa$ and particle diameter ($D_p$) vary and the reciprocal relationship between $\kappa$ and the third power of the critical $D_p$ ($D_c^3$) at a given $SS$." in lines 58 to 62.

3)        line 156: 40% RH seems still quite high for size selected CCN measurements.

For aerosols with a high hygroscopicity, there may be considerable amounts of water in the selected particles which would then create a noticeable bias in the measured DC/SSC pairs.

Response: Thanks for your reminder. The RH of the sampling line follows the requirement from the ACTRIS (https://www.actris.eu/sites/default/files/2021-06/Preliminary%20ACTRIS%20recommendations%20for%20aerosol%20in-situ%20measurements%20June%202021.pdf), which were mostly made for MPSS. Exactly, the 40% RH still caused possible bias in the measured DC and SSC pairs. However, drying lower the aerosol is certainly possible but with the risqué of changing its chemical composition. In the revision, we stated that we follow the ACTRIS recommendations as follows:

"Ambient aerosol particles were first pretreated through a $PM_{10}$ Anderson inlet and an automatic aerosol diffusion dryer kept the relative humidity in sampling lines at a relative humidity less than 40% following the ACTRIS recommendations." in lines 148 to 151.

4)      line 162: What is meant by "near-PM1"?

Response: The ACSM does not have an absolute PM1 cut-off since the aerodynamic lenses that equipped the instrument have a transmission efficiency of approximately 40 % at diameter of 1 µm (Liu et al, 2007, doi10.1080/02786820701422278). Therefore, the instrument is commonly considered as measuring near-PM1.

To make the sentence less confusion, it was rewritten as follows:

"An aerosol chemical speciation monitor (ACSM, Aerodyne Inc; Ng et al., 2011) was used to measure the chemical compositions of the non-refractory submicron aerosol particulate matter (nitrate, sulfate, chloride, ammonium, and organics)." in lines 154 to 157.

5)      line 177: The total SS cycle is 2.5h. Were all SS steps of the same length? Was all data used or were the first x min omitted while SS stabilised after change? Was there one Dp scan or multiple during one SS setting?

Response: Thanks for the questions. The "effective" SS steps were all of the same length. The coupling between size selection and CCNC was programmed in a way that the size resolved measurements started only after the temperature and thereby the supersaturation of the CCNC was stabilized. As the diameter scan started after ss

stabilization, the measurement itself was the same length at all supersaturations. At fully stabilized CCNC conditions we did one Dp scan at per SS setting. The size selection was done with a DMA (25 diameters) and a CPC (counting time 30sec per Dp). The slight variations in the 2.5h total SS cycle was only due to the waiting time until the temperature of the CCNC was stabilized. In the text we clarified that in the in the instrumental part (Section 2.1), as follows:

"The coupling between size selection and CCNC was programmed in a way that the size resolved measurements started only after the temperature and thereby the SS of the CCNC was stabilized. As the diameter scan started after SS stabilization, the measurement itself was the same length at all SS conditions. At fully stabilized CCNC conditions we did one Dp scan at per SS setting. A total of five different SS conditions was set in the CCNC instrument (0.1%, 0.2%, 0.3%, 0.5%, and 0.7%). A complete SS cycle lasted ~2.5 hours and the slight variations in the 2.5h total SS cycle was only due to the waiting time until the temperature of the CCNC was stabilized." in lines 171 to 179.

6)      line 184: Rose et al. (2008) point out that you need to state which parameterisation/model was used to derive the theoretical SS/Dcrit values. They provide many in their study. Which one was used here? Also, what does "regularly" mean? Every month?

Response: Thanks for your comment. We used the E-AIM model as based on Clegg et al., (1998, DOI: 10.1021/jp973043j). The CCNC was calibrated once a year during the campaign. We clarified that in the text (lines 185 to 187) as follows:

"Throughout the campaign, the CCNC was calibrated once a year following the procedures outlined in Rose et al. (2008) with using the E-AIM model (Clegg et al. 1998)."

7)      line 207ff: What was the size range of the DMA coupled to the CCN-C? Later in Line 391, the authors state that the limit was 40-200nm for $D_p$. This information needs to go into the Methods section. Was the D-MPSS compared with the DMA-CPC to check that the particle losses in each instrument are comparable? Also, why was the upper size limited in the DMA? Was it due to HV? Or was there a physical limit (Impactor, too many bends in line?). I.e., are you sure that the large particles did make

it through the DMA in the same way as they reached the D-MPSS?

Response: The size range of the DMA coupled to CCNC was 20 to 350 at 0.1% and 20-300nm at all other supersaturations. However, the DMPS scan was continued to the upper limit of 440 nm to be able to compared also for larger particles to the MPSS and check for losses. The ratio of particles larger than 440nm is assumed to be neglectable at Melpitz site. The size-resolved $\kappa$ (pair of $\kappa$ and $D_c$) can be obtained at each $SS$ cycle. Our monodisperse CCN measurements provide the size-resolved $\kappa$ within $D_p$ ($D_c$) of ~40−200 nm, which depends largely on the $SS$ setting of 0.1% to 0.7%.

The losses in both instruments were checked and it was corrected for in the inversion routine. The upper diameter limit is set by the DMA technique (high voltage supply 12.5kV and aerosol to sheath ratio 1/10). No impactor was applied, bends in the lines were omitted as far as possible and for unavoidable losses was corrected. Concerning possible differences in the losses in DMA/CCN and MPSS: We measured rather the shape of the activation curve than the absolute number of particles. We measured the activation curve, inverted it and multiplied it with the MPSS size scan.

The following sentence has been added to Methods section (2.2)

"The losses in both instruments were checked and it was corrected for in the inversion routine." in lines 170 to 171, and

"Our monodisperse CCN measurements provide the size-resolved $\kappa$ within $D_p$ ($D_c$) of ~40−200 nm, which depends largely on the $SS$ setting of 0.1% to 0.7%." in lines 238 to 240.

8)      Line 210f/ Eq 1: What is meant with "a and b are the lower and upper limits for calculating Dc"? To my understanding, 'b' is the height of the upper plateau of the sigmoidal function and 'a' is the offset from 0 in the y direction. If b is lower than 1 (after accounting for any instrument discrepancies), this indicates the presence of particles that do not activate (e.g., pure black carbon or dust). As the mixing state is investigated later on, it would be good to know if the b value was lower than 1 during the times with higher external mixing.

Response: Thanks for the comment. Yes, "a" is the offset from 0 in the y direction and 'b' is the height of the upper plateau of the sigmoidal function. Theoretically, b < 1 means that some of the larger particles cannot be activated (e.g., pure black carbon or dust), which could correspond to the large (D75-D25)/DC (i.e., higher degree of

external mixing). But we examined the relationship between parameter b and (D75-D25)/DC, the results show no correlation as shown in following Figure R1. The reasons could come from the instrument discrepancies and the sigmoid fitting bias. In this study, we chose to use the (D75-D25)/DC rather than parameter b to investigate the mixing state following the previous studies (e.g., Jurányi et al., 2013, https://doi.org/10.5194/acp-13-6431-2013).

[Figure]

Figure R1. Relationship between parameter b and (D75-D25)/DC in sigmoid function. Related sentence (lines 213 to 215) has been revised as follows: "Where $a$ is the offset from 0 in the y direction and $b$ is the height of the upper plateau of the sigmoidal function, $D_c$ is the critical diameter, and $\sigma_s$ is a measure for the width of the sigmoid function."

9)      Line 227 ff: As stated in Petters and Kreidenweis (2007), Eq 2 is an approximation which is only valid if the solution is very dilute at the point of activation. Is this approximation true for the analysed data in this study? I.e. are the kappa values high enough (Petters and Kreidenweis (2007) suggest k>0.2 as the threshold)?

Response: Thanks for your reminder. In this study, the kccn is almost larger than 0.2, especially at relatively low SS conditions. At relatively high SS conditions (e.g., 0.5 and 0.7%), the presence of kccn < 0.2 could cause a slight bias in calculating k. We clarified that in the text (lines 240 to 242) as follows:

"Note that equation 2a is an approximation of $\kappa$-Köhler equation and when $\kappa$ is less than 0.2, it causes a slight bias in calculating $\kappa$ (Petters and Kreidenweis, 2007)."

10)      Line 230: The authors assume that the surface tension of the droplet is constant, and k is the only unconstrained parameter in their equation. If the surface tension does vary, the effect in measured aerosol particle activation behaviour will then be accounted for by a different k value. Then k is not just representing the hygroscopicity of the aerosol but also the effect of any surface tension change and may differ more from $k_{chem}$ which can only take the composition into account. Since the debate about the importance of surface tension changes and the connected bulk/surface partitioning is still ongoing (e.g.: Ovadnevaite et al. (2017); Vepsäläinen et al. (2021)), I recommend that the authors simply include a brief sentence stating that k may pick up surface tension changes.

Response: Thanks for your suggestion. We have added the following sentence to pick up this effect in lines 242 to 245:

"Additionally, the debate about the importance of $\sigma_{s/a}$ changes and the connected bulk/surface partitioning on activation of aerosols is on ongoing (e.g., Ovadnevaite et al., 2017; Vepsäläinen et al., 2022), which is not focused on in this study."

11)      Line 242: Why was k=0.1 used for organics? In your introduction, you show that $k_{org}$ can vary enough to be relevant for $N_{CCN}$ predictions (e.g., in the range of 0.05 – 0.15 which can be cause by changes in organic composition). Ambient and chamber measurements have shown that $k_{org}$ is a function of the degree of oxidation (e.g., expressed with O:C). What impact will it have if you use such a parameterisation for $k_{org}$ (e.g., $k_{org} = 0.18*O:C +0.03$ given in (Lambe et al., 2011))

Response:   Thanks for your good question. Q-ACSM is equipped with a simple quadrupole mass spectrometer having only a unit mass resolution, which is contrary to the HR-ToF-AMS. Therefore, estimation of the O:C ratio can only be done using the fraction of m/z 44 (mostly $CO_2$) to the total OA. For technical issues this ration could be quite instrument dependent (Crenn et al., 2015, https://doi.org/10.5194/amt-8-5063-2015). Making challenging the estimation of the O:C ratio and OC (Poulain et al., 2020, https://doi.org/10.5194/amt-13-4973-2020). Therefore, we keep to a constant $k_{org}$ value of 0.1 in this study.

12)      Line 245 "When k is given…" For equation 4 to work, only a value for $D_C$ is needed. Size resolved CCN counter measurements provide $SS/D_C$ pairs and Eq 4 can

be used directly. If no $D_C$ values are available, $D_C$ can be calculated for any SS from a given $k$.

Response: Thanks for your comment. We fully understand that Size resolved CCN counter measurements provide SS/DC pairs and Eq 4 can be used directly. This sentence has been rewritten as follows:

"The CCN number size distribution is a part of the particle number size distribution (PNSD), which approximately corresponds to the part of PNSD with $D_p > D_c$ when assuming particles to be internally mixed (Figure S2 in SI). The assumption of the internal mixing could be reasonable because the median values of the parameter $b$ and $(D_{75} - D_{25})/D_c$ are 1.0 and 0.18. Thus, $D_c$ plays a critical role on diagnosing $N_{CCN}$ in models, which can be derived from $\kappa$ parameterization at a given $SS$. When $\kappa$ is obtained, $D_c(\kappa, SS)$ is calculated from equation 2a." in lines 256 to 262.

13)     Average values in Fig 3, 4, 5, and 7: The authors show average values of NCCN, AR, etc. and use the standard deviation to indicate the spread of the data. Would it not be better to use the interquartile range for that purpose? So, indicting the mean (or median) value with a marker and then use a error bar/shaded area to indicate the Q25 and Q75 range?

Response: Thanks for your suggestions. We have adopted the interquartile range to show the characteristics of NCCN, AR, etc., in the new version. For instance, the new Figures are as follows.

[Figure]

Figure 3. (a) Relationships between CCN number concentration ($N_{CCN}$) and supersaturation ($SS$), and relationship between activation ratios (AR) and $SS$ for different seasons. (b) Seasonal trends of $N_{CCN}$ and AR at $SS$ = 0.1% and 0.7%. Dots represent the median values of $N_{CCN}$ and AR. Shaded areas represent the values in the range from 25th to 75th percent. Red lines are the power-law fittings for $N_{CCN}$ (and AR) vs. $SS$. Two parameters of the fitting results are shown in brackets.

[Figure]

Figure 4. Seasonal variations of (a) aerosol particle number size distribution ($\mathrm{d}N_{aero}/\mathrm{dlog}D_p$ vs. $D_p$, $D_p$ is particle diameter), (b) total aerosol number concentration with $D_p$ ranging from 10 to 800 nm ($N_{aero}$) and geometric mean diameter of the particles ($GMD$), and (c) mass concentration and ratio of each component in aerosol particle with $D_p$ less than 1 μm and the hygroscopicity factor calculated from the chemical composition ($\kappa_{chem}$). Dots represent the median values. Shaded areas represent the values in the range from 25$^{th}$ to 75$^{th}$ percent.

[Figure]

Figure 5. Monthly variations of (a) hygroscopicity factor calculated from monodisperse CCN measurements ($\kappa_{\text{CCN}}$) at supersaturation ($SS$) of 0.1% and 0.7%, and hygroscopicity factor calculated from particle chemical composition ($\kappa_{\text{chem}}$), (b) critical diameter of dry particle for activation ($D_c$) at $SS$ = 0.1% and 0.7%, and (c) the degree of external mixture (($D_{75} - D_{25})/D_c$) at $SS$ = 0.1% and 0.7%. The definitions of $D_{75}$ and $D_{25}$ are the $D_p$ at which 75% and 25% of the particles are activated at the given $SS$, respectively. Dots represent the median values. Shaded areas represent the values in the range from 25[th] to 75[th] percent.

[Figure]

Figure 6. (a) Relationship between the hygroscopicity factor calculated from monodisperse CCN measurements ($\kappa_{CCN}$) and particle diameter ($D_p$), and (b) degree of external mixture (($D_{75}$ – $D_{25}$)/$D_c$) vs. $D_p$ at each season. The definitions of $D_{75}$ and $D_{25}$ are the $D_p$ at which 75% and 25% of the particles are activated at the given *SS*, respectively. Red lines are power-law fits. Dots represent the median values. Shaded areas represent the values in the range from 25[th] to 75[th] percent.

14)      There are multiple issues with Fig 3.

a.            The Figure caption does not contain the information about the black markers. in Fig 3a. There is no description of the right-hand axis in Fig 3a.

Response: Thanks for your reminder. The caption of the right-hand axis has been added in original Figure 3a (now Figure S2).

b.            Neither is there any information about the averaging which seems to be different for the size distribution (two values per month?) and the AR values.

Response: There are three values per month (~every ten days) for averaging the CCN number size distribution and one value per month for median AR. We clarified that in the title of this Figure as follows:

"The CCN number size distribution was the result of using an average of every ten days. The black dot presents the median AR at each month."

c.            The panels in Fig 3a are so tiny that it is very hard to see, e.g., the differences between SS=0.5% and 0.7%.

Response: We have revised the original Figure 3a (now is Figure S2) as follows:

[Figure]

Figure S2. Monthly variations of CCN number size distributions and activation ratios (AR) at five different supersaturation (*SS*) conditions (a to e). The CCN number size distribution was the result of using an average of every ten days. The black dot presents the median AR at each month.

d.                      The error bars are outside of the y-axis range for some plots in Fig 3b and c.

Response: Thanks for your reminder. After using the Q25 and Q75 values as shaded areas, the original ranges of y-axis in original Fig 3b and 3c (now is Figure 3a) are suitable. The new Figure 3a is as follow:

[Figure]

Figure 3. (a) Relationships between CCN number concentration ($N_{CCN}$) and supersaturation ($SS$), and relationship between activation ratios (AR) and $SS$ for different seasons. (b) Seasonal trends of $N_{CCN}$ and AR at $SS = 0.1\%$ and 0.7%. Dots represent the median values of $N_{CCN}$ and AR. Shaded areas represent the values in the range from 25th to 75th percent. Red lines are the power-law fittings for $N_{CCN}$ (and AR) vs. $SS$. Two parameters of the fitting results are shown in brackets.

e.    For me, Fig 3a added very little to understand the description provided in Chapter 3.1. Also, for the overall interpretation of the data, the CCN number size distribution is not as relevant and could easily move to the SI. See next comment for other changes proposed for Fig 3.

Response: Thanks for your comments. The CCN number size distribution in original Figure 3a has been revised and moved to SI (Figure S2).

15)   The description of the AR and NCCN values and trends is very hard to follow in Chapter 3.1. This is cause by the excessive details in the description and the choice of visualisation of the data.

a.    It is good that the authors compare their values to so many other studies. But due to each study having a different SS range, it is difficult to really understand how

the data compares. Table 2 does not provide much more insights. But a simple Figure does (Fig R1 below):

[Figure]

Figure R1: NCCN from this study and studies cited in chapter 3.1.

Response: Thanks for your great comments. Exactly, such figure is much better than original Table for showing the results. We have added a new figure (Figure 2) in this paper and moved the original Table 2 to SI (Table S1). The new Figure 2 is as follow:

[Figure]

Figure 2. Relationship between CCN number concentration ($N_{CCN}$) and supersaturation ($SS$) measured at Melpitz and other stations.

b.       They authors look into the seasonal trends of NCCN and AR and again I got lost in all the numbers and the provided visualizations do not help. Looking at the

plots in Fig 3b and c, the seasonal behavior of AR and NCCN look indeed "similar". But when I visualized the given values from lines 273 – 280 as a function of season (see Fig R2 below), I realized that there are some interesting differences. Going from 0.1% to 0.7%, the minimum of the NCCN shifts from autumn to summer (Fig R2 top) while the trends in AR with a minimum in summer are the same for these SS (Fig R 2 bottom. Why is that the case? To me, it is clear that this must be connected to changes in the PNSD. But before the authors get to that, they first dive into the details of the NCCN(SS) and AR(SS) relationships. Here already the connection between the set SS and the size range that is probed is important. But that is not mentioned until lines 385-391 (see Specific Comment 28)

[Figure]

Figure R2: Seasonal trends of NCCN (Top) and AR (bottom). Values for SS=0.1% are in blue and use the left-hand y- axis, while the values for SS=0.7% are indicated in orange and use the right-hand axis.

Response: Thanks for your comments. We added the seasonal trend of NCCN (& AR) at SS of 0.1% and 0.7% as a new Figure (Figure 3b). Then we see that the minimum of the NCCN shifts from summer to autumn as SS increasing from 0.1% to 0.7%. The reason related to the seasonal trend of PNSD and kchem. From summer to autumn, the Naero decreases while the GMD and kchem both increases as shown in Figure S6. At relatively low SS condition (e.g., 0.1%), the minimum of NCCN is in summer mainly because of the lack of large particles in PNSD as shown in Figure S4. At relatively high SS condition (e.g., 0.7%), the minimum of NCCN is in autumn mainly because of the low Naero, even though the GMD and kchem higher than those in summer.

We clarified that in the text (lines 295 to 313) as follows:

"To explain the seasonal variations in aerosol activation characteristics, we investigated the PNSD and chemical compositions as shown in Figure 4. In summer, affected by the frequent NPF events (Ma et al., 2015; Wang et al., 2017), the Aitken-mode particles with $D_p < 100$ nm accounted for the largest portion of the PNSD (Figure S4 in SI),

resulting in the highest $N_{aero}$ and the smallest geometric mean diameter ($GMD =$

$\exp\left(\dfrac{\sum_i n_i \times lnD_i}{N_{aero}}\right)$) among the four seasons. Additionally, in summer, there was the lowest

bulk $\kappa_{chem}$ with median value of 0.24 corresponding to the highest organic mass fraction

(56% of total mass), which could be related to the strong formation of the secondary

organic aerosol (SOA). Therefore, the $N_{CCN}$ and AR both kept relatively low values in

summer, especially at low $SS$ conditions (e.g., at $SS = 0.1\%$). On the contrary in winter,

the relatively low number concentration of Aitken-mode particles caused the lowest

$N_{aero}$ and the largest $GMD$ among the four seasons, which could be owing to the rare

NPF events. Meanwhile, in winter, low temperatures favored the particulate phase of

nitrate (Poulain et al., 2011), causing the highest nitrate mass fraction (31% of total

mass) among the four seasons, which might explain the highest $\kappa_{chem}$ (median value of

0.34). Taking all three together, the lowest $N_{aero}$, the largest $GMD$, as well as the highest

$\kappa_{chem}$, contribute to the highest AR value in winter at each $SS$ condition. The

relationships between $\kappa_{chem}$ and each particle component, and the correlations among

seasonal median values of $N_{aero}$, $GMD$, and $\kappa_{chem}$ are in SI (Text S1, Figures S4 and

S5)."

[Figure]

Figure 3. (a) Relationships between CCN number concentration ($N_{CCN}$) and

supersaturation (*SS*), and relationship between activation ratios (AR) and *SS* for different seasons. (b) Seasonal trends of $N_{CCN}$ and AR at *SS* = 0.1% and 0.7%. Dots represent the median values of $N_{CCN}$ and AR. Shaded areas represent the values in the range from 25[th] to 75[th] percent. Red lines are the power-law fittings for $N_{CCN}$ (and AR) vs. *SS*. Two parameters of the fitting results are shown in brackets.

[Figure]

Figure S6. Relationships among seasonal median values of aerosol number concentration with diameter raging 10 to 800 nm ($N_{aero}$), geometric mean diameter of aerosol particles (*GMD*), and particle hygroscopicity parameter calculated from the chemical compositions ($\kappa_{chem}$). The dots represent the median values at each season.

[Figure]

Figure S4. Mean particle number size distribution at each season.

c.       I do not understand what the authors mean with "CCN number size distribution gradually peaks in summer". Whatever is meant by that, how is that

connected to the seasonal trends in $N_{CCN}$ and AR, especially the summer minimums of AR and $N_{CCN}$(SS=0.1%)?

Response: Thanks for your comments. We originally wanted to say that as SS increases, the Dc decreases and the CCN number size distribution peaks in the small particles, which is more noticeable in summer than other seasons. This sentence has been deleted due to the ambiguous statement and weak connection to the seasonal trends in NCCN and AR.

d.      Why are AR and $N_{CCN}$ more sensitive to SS in summer than in winter? My recommendation for cleaning up this chapter is to change Fig 3 by moving Fig 3a into the SI. Instead, provide a larger version of the "all data set" $N_{CCN}$ vs SS which includes the values for the other studies as shown here in Fig R1. This will make the naming of all the number for the previous studies obsolete and Table 2 can also be moved to SI. If the authors keep the description and deepen the discussion of the seasonal trends, I strongly recommend adding something like Fig R2 either to Fig 3 or into the SI to help the reader follow the descriptions. Shifting the explanations about the PNSD to follow the description of these trends will then feel more natural. The next few Specific Comment are also related to improving this chapter.

Response: Thanks for your suggestions. The original Figure 3a has been moved to SI (Figure S2). We add a new Figure to show the difference between the NCCN vs SS relation in this study and other studies (Figure 2). The new Figure 3 combines the original Figures 3b, 3c and the Figure R2, as follows. Additionally, the statement has been revised in Chapter 3.1. We have reduced descriptive statements and increased the discussion of causes.

In summer, the median $N_{CCN}$ and AR are both lowest at $SS = 0.1\%$, which contributed to the largest sensitivity of $N_{CCN}$ and AR to $SS$, i.e., the largest slope parameter in the power-law fitting among the four seasons. Additionally, the shape of the PNSD contributed to explain the sensitivity of $N_{CCN}$ and AR to $SS$. The PNSD in summer was steepest in the 40-200 nm size range among the four seasons (Figure S4 in SI). Thus, in summer, a small shift in $D_c$ will change the $N_{CCN}$ and AR much more than those in winter where the PNSD looks broader, causing the strong sensitivity of $N_{CCN}$ and AR to $SS$.

We clarified that in the text (lines 287 to 294) as follows:

"In summer, the median $N_{CCN}$ and AR are both lowest at $SS = 0.1\%$, which

contributed to the largest sensitivity of $N_{CCN}$ and AR to $SS$, i.e., the largest slope parameter in the power-law fitting among the four seasons. Additionally, the shape of the PNSD contributed to explain the sensitivity of $N_{CCN}$ and AR to $SS$. The PNSD in summer was steepest in the 40-200 nm size range among the four seasons (Figure S4 in SI). Thus, in summer, a small shift in $D_c$ will change the $N_{CCN}$ and AR much more than those in winter where the PNSD looks broader, causing the strong sensitivity of $N_{CCN}$ and AR to $SS$."

[Figure]

Figure 2. Relationship between CCN number concentration ($N_{CCN}$) and supersaturation ($SS$) measured at Melpitz and other stations.

[Figure]

Figure 3. (a) Relationships between CCN number concentration ($N_{CCN}$) and supersaturation ($SS$), and relationship between activation ratios (AR) and $SS$ for different seasons. (b) Seasonal trends of $N_{CCN}$ and AR at $SS = 0.1\%$ and $0.7\%$. Dots represent the median values of $N_{CCN}$ and AR. Shaded areas represent the values in the range from $25^{th}$ to $75^{th}$ percent. Red lines are the power-law fittings for $N_{CCN}$ (and AR) vs. $SS$. Two parameters of the fitting results are shown in brackets.

16)      line 282ff: The error function used for fitting seems to have 4 free parameters. Each data set that is fitted has 5 values. Some people might say that that is a problem. Or was the original data fitted and not the (seasonal) averages? This is not mentioned in the text. However, since the error function fits are not used anywhere in the manuscript other than stating that the function fits slightly better, this could be reduced to stating that the fit was also performed with an error function and the fitted parameters and curves are in the SI.

Response: Thanks for your suggestion. The error function used for fitting is y=a+a*erf(ln(x/b)/c), which has 3 free parameters. It was used to fit the seasonal median values (averages in old version) rather than the original data. The fitting results of error function have moved to SI (Table S2) following your suggestion. In the manuscript, we revised

the statement as follows:

"The fit was also performed with an error function (Pöhlker et al., 2018) and the fitted parameters are in the SI (Table S2)." in lines 279 to 280.

17)      line 298: This information about what the CCN number size distribution represents should have appeared at the end of the methods section when EQ 4 is introduced. The schematic diagram is using the assumption that there are no non-CCN active particles at $D_p > D_c$ (see also Specific Comment 8). Is that assumption reasonable? You should be able to estimate that from checking the plateau values in the AF vs Dp plots. How close are these values to 1.0 (after you accounted for different losses in the two instruments/sampling lines)?

Response: Thanks for your suggestion. This information about what the CCN number size distribution represents has moved to the end of methods section (lines 256 to 258) as follows:

"The CCN number size distribution is a part of the particle number size distribution (PNSD), which approximately corresponds to the part of PNSD with $D_p > D_c$ when assuming particles to be internally mixed (Figure S2 in SI)."

The assumption of the internal mixing could be reasonable because the median values of the parameter $b$ and $(D_{75} - D_{25})/D_c$ are 1.0 and 0.18 as shown in following Figure R2 and Figure R3. We clarified that in the text as follows:

"The assumption of the internal mixing could be reasonable because the median values of the parameter $b$ and $(D_{75} - D_{25})/D_c$ are 1.0 and 0.18." in lines 258 to 260.

[Figure]

Figure R2. Probability distribution of parameter b in sigmoid function at different supersaturation (SS) conditions.

[Figure]

Figure R3. Probability distribution of $(D_{75} - D_{25})/D_c$ at different supersaturation (SS) conditions.

18)      line 314f: Again, the phrase "CCN number size distribution gradually peaks in summer" occurs without clarifying what is peaking.

Response: Thanks for your comment. We originally wanted to say that as SS increases, the Dc decreases and the CCN number size distribution peaks in the small particles, which is more noticeable in summer than other seasons. This sentence has been also deleted due to the ambiguous statement.

19)      Lines 306-317: It seems plausible that the presence of a large number of small particles explains the minimum of AR and maximum of NCCN(SS≤0.5%) during summer. But why was the influence of the change in hygroscopicity omitted? Winter and Spring have much higher kCCN values (at least for the larger particles) which will also contribute to the high AR and NCCN during that season.

Response: Thanks for your comment. Exactly, the hygroscopicity was also an important factor on affecting the AR and NCCN. We used the kchem rather than kccn to analyze the influence of the change in k on AR and NCCN. Finally, we combined the effects of Naero, GMD, and kchem on $N_{CCN}$ and AR as follows:

"To explain the seasonal variations in aerosol activation characteristics, we investigated the PNSD and chemical compositions as shown in Figure 4. In summer, affected by the frequent NPF events (Ma et al., 2015; Wang et al., 2017), the Aitken-mode particles with $D_p < 100$ nm accounted for the largest portion of the PNSD (Figure S4 in SI), resulting in the highest $N_{aero}$ and the smallest geometric mean diameter ($GMD = \exp\left(\frac{\sum_i n_i \times \ln D_i}{N_{aero}}\right)$) among the four seasons. Additionally, in summer, there was the lowest bulk $\kappa_{chem}$ with median value of 0.24 corresponding to the highest organic mass fraction (56% of total mass), which could be related to the strong formation of the secondary organic aerosol (SOA). Therefore, the $N_{CCN}$ and AR both kept relatively low values in summer, especially at low $SS$ conditions (e.g., at $SS = 0.1\%$). On the contrary in winter, the relatively low number concentration of Aitken-mode particles caused the lowest $N_{aero}$ and the largest $GMD$ among the four seasons, which could be owing to the

rare NPF events. Meanwhile, in winter, low temperatures favored the particulate phase of nitrate (Poulain et al., 2011), causing the highest nitrate mass fraction (31% of total mass) among the four seasons, which might explain the highest $\kappa_{chem}$ (median value of 0.34). Taking all three together, the lowest $N_{aero}$, the largest $GMD$, as well as the highest $\kappa_{chem}$, contribute to the highest AR value in winter at each $SS$ condition. The relationships between $\kappa_{chem}$ and each particle component, and the correlations among seasonal median values of $N_{aero}$, $GMD$, and $\kappa_{chem}$ are in SI (Text S1, Figures S4 and S5)." in lines 290 to 313.

20)     Fig 4 should be improved. The black markers for $k_{chem}$ are difficult to see in front of the dark blue background from NO3. Why does the $k_{chem}$ axis start at 0.2 and not 0? I do not like how the GMD and $N_{aero}$ plot are put over the PNSD graph. Overlaying two panels over the PNSD is not straight forward to read. The intuitive interpretation is that the two sets of black markers are both on an axis extending the full hight of the PNSD plot. The March and February markers are only partially visible.

Response: Thanks for the comment. The figure 4 was improved as can be seen below. For the axis of the kchem, it was set in the range 0.2 to 0.5 to make more obvious the monthly variations, which will not be visible when starting to 0.

[Figure]

Figure 4. Seasonal variations of (a) aerosol particle number size distribution ($dN_{aero}/dlogD_p$ vs. $D_p$, $D_p$ is particle diameter), (b) total aerosol number concentration with $D_p$ ranging 10 to 800 nm ($N_{aero}$) and geometric mean diameter of the particles ($GMD$), and (c) mass concentration and ratio of each component in aerosol particle with $D_p$ less than 1 μm and the hygroscopicity factor calculated from the chemical composition ($\kappa_{chem}$). Dots represent the median values. Shaded areas represent the values in the range from $25^{th}$ to $75^{th}$ percent.

21)      Lines 323-329: While these correlations are interesting, Fig 5 could also be in the SI. Especially, since there is no interpretation of the meaning of the k ~ fX

correlations currently. Yet another example where the reader is left to come up with their own conclusions about an interesting observation. Here is my take:

For understanding the relationship between $k_{chem}$ and the individual composition groups, it is important to realise that these groups do not act in the same way in Eq 3. The influence of Org is direct. $k_{org}$ is smaller than $k_{inorg}$. Thus, higher $f_{org}$ means lower $k_{chem}$. But with SO4, NH4, NO3 it is more complicated because they are coupled through the ion balance. The absolute amount of SO4 and NH4 seems pretty stable. But the NO3 amount changes a lot between the seasons. The presence of NO3 shifts the salts from mostly (NH4)2SO4 towards NH4NO3 and NH4HSO4 or even H2SO4. k values are very similar between (NH4)2SO4, NH4HSO4, and NH4NO3, but $k_{H2SO4}$ is much higher. Thus, an increase in NO3 can have a dual impact on k for this data set. The increase in NO3 adds a higher proportion of salt and also increases the k of SO4. So, if fSO4 decreases because more Org is present, k decreases. If fSO4 decreases because more NO3 is present, k increase. As these two trends are opposite, the correlation of fSO4 and k will be poor. Since this behaviour is opposite the usually assumed "fSO4 increase leads to k increase", it is worth discussing. Also, why is NO3 increasing in winter and spring? Can it really be just the change in ambient T? Could it be linked to the "local pollution" that is mentioned without any explanation in line 384? And to link this to the bigger picture: could the balance between NO3 and the other aerosol constituents be an important factor when comparing aerosol activation behaviour in different regions?

Response: Thanks for the comment. We have moved the original Figure 5 to SI (Figure S5). The discussions on the relationship between $k_{chem}$ and the individual composition groups are also presented in SI (Text S1) as follows:

**"Text S1. Relationship between aerosol hygroscopicity factor calculated from the chemical composition and the individual composition groups.**

For understanding the relationship between aerosol hygroscopicity factor calculated from the chemical composition ($\kappa_{chem}$) and the individual composition groups, it is important to realize that these groups do not act in the same way in Eq 3. The influence of organics is direct. As shown in Table 1, the hygroscopicity of

organics ($\kappa_{org}$) is smaller than that of inorganics ($\kappa_{inorg}$). Thus, higher mass fraction of organics ($f_{org}$) means lower $\kappa_{chem}$, as shown in Figure S5a. But with $SO_4$, $NH_4$, $NO_3$ it is more complicated because they are coupled through the ion balance. As shown in Figure 4c, the absolute amount of $SO_4$ and $NH_4$ seems stable, but the $NO_3$ amount changes a lot between the seasons. The presence of $NO_3$ shifts the salts from mostly $(NH_4)_2SO_4$ towards $NH_4NO_3$ and $NH_4HSO_4$ or even $H_2SO_4$. Hygroscopicity factors ($\kappa$) are very similar between $(NH_4)_2SO_4$, $NH_4HSO_4$, and $NH_4NO_3$, but $\kappa$ of $H_2SO_4$ is much higher. Thus, an increase in $NO_3$ can have a dual impact on $\kappa$ for this data set, causing the positive correlation between mass fraction of nitrate and $\kappa_{chem}$ in Figure S5b. The increase in $NO_3$ adds a higher proportion of salt and increases the $\kappa$ of $SO_4$. So, if mass fraction of $SO_4$ ($f_{sulfate}$) decreases because more organics is present, $\kappa$ decreases. If $f_{sulfate}$ decreases because more $NO_3$ is present, k increase. As these two trends are opposite, the correlation of $f_{sulfate}$ and $\kappa_{chem}$ will be poor."

Some elements of answer are as follows:

a) Particulate ammonium nitrate is mostly driven by temperature. Even if concentrations of AN precursors (HNO3 and NH3) are more important in spring and summer (For Melpitz see Stieger et al., 2017, DOI 10.1007/s10874-017-9361-0)

b) From the ACSM aspect, most of the time, the particles are neutralized (means NH4 concentration almost fully explain the concentration of nitrate and sulfate), which is not surprising since the station is surrounding by field which are regularly fertilized. Then we examined the trend of volume fraction of (NH4)2SO4, NH4HSO4, and H2SO4 over the year and the results are as shown in Figure R4. Compared to (NH4)2SO4 and NH4HSO4, the volume fraction of H2SO4 could be negligible. Thus, the kchem is depending on the ratio inorganic/organic independently to the exact salt of inorganic since they all have similar kappa-values.

[Figure]

Figure R4. monthly variations of the mean volume fraction of (NH4)2SO4, NH4HSO4, and H2SO4.

22)   Lines 330-337: The changes in the width of the CCN number size distribution are not just related to the hygroscopicity (i.e., the $D_C$ values). The shape of the PNSD plays an equally important role. The $D_C$ value (i.e. the hygroscopicity) determines the edge at the smaller end of the CCN number size distribution. But the shape of the distribution at larger sizes depends more on the shape of the PNSD. I.e. with an identical $k / D_C$ the winter CCN number size distribution will be wider because of the shift to larger sizes in the PNSD. The different shape of the PNSD may also help to explain the stronger sensitivity of $N_{CCN}$ to SS during summer. The PNSD is probably steeper in the 40-150nm size range. Thus, a small shift in $D_C$ will change the $N_{CCN}$ much more than in winter where the PNSD look broader.

Response: Thanks for your comment. Exactly, the PNSD shape and the $k / D_C$ change the CCN number size distribution, thereby the $N_{CCN}$ and AR, and the sensitivity of $N_{CCN}$ (AR) to SS. We add the seasonal mean PNSD in SI (Figure S6). The statement in the manuscript has been revised as follows.

"Additionally, the shape of the PNSD contributed to explain the sensitivity of $N_{CCN}$ and AR to SS. The PNSD in summer was steepest in the 40-200 nm size range among the four seasons (Figure S4 in SI). Thus, in summer, a small shift in $D_c$ will change the $N_{CCN}$ and AR much more than those in winter where the PNSD looks broader, causing

the strong sensitivity of $N_{CCN}$ and AR to $SS$." In lines 290 to 294.

[Figure]

Figure S6. Mean particle number size distribution at each season.

23)    Chapter 3.2: Throughout the manuscript the authors write as if $k_{CCN}$ and $D_C$ are independently measured variables, while really k is calculated from the measured $SS_C/D_C$ pairs. Describing both the $D_C$ and k trends in details is thus redundant. As the authors want to compare the hygroscopicity to the composition, it is sufficient to present the $D_C$ values only in the Table and figure. If a reader is interested in the exact values for $k_{CCN}$ or $D_C$, a Table/Figure is anyway much faster than trying to find the relevant values in the long text. Then focus on the $k_{CCN}$ trends and compare them with the $k_{chem}$ information (see also next Specific Comment). This will make the actual discussion/ interpretation/comparison much more readable. To facilitate the $k_{CCN}$ / $k_{chem}$ comparison, add the $k_{chem}$ values to Fig 6a.

Response: Thanks for your suggestions.

a)  We understand that describing both the $D_C$ and k trends in details is redundant because k is calculated from the measured $SS_C/D_C$ pairs. In the revised Figure (original Figure 6, now Figure 5), we keep the $D_C$ trends because we would like to show the Dc range when SS increasing from 0.1% to 0.7%. But the describing has been much reduced.

b) In Chapter 3.2, numbers referring to kCCN and DC were removed from the text of the manuscript as suggested by the reviewer.

c) The original Figure 6a (now Figure 5a) has been added the $\kappa_{chem}$ trend.

The statement and Figure are revised as follows:

"Figure 5b depicts the monthly variation of $D_c$ at $SS$ of 0.1% and 0.7%, which shows the opposite trend to $\kappa_{CCN}(SS)$ because of the negative correlation of $D_c^3(SS)$ vs. $\kappa(SS)$ shown in equation 2a. Compared to the $D_c$ at lower $SS$ conditions (e.g., 0.1%), $D_c$ has a more significant seasonal trend at higher $SS$ conditions (e.g., 0.7%). At $SS = 0.7\%$, the low $\kappa_{CCN}$ caused the large $D_c$ in summer, whereas the high $\kappa_{CCN}$ caused the small $D_c$ in spring and winter." In lines 333 to 338.

[Figure]

Figure 5. Monthly variations of (a) hygroscopicity factor calculated from monodisperse CCN measurements ($\kappa_{CCN}$) at supersaturation ($SS$) of 0.1% and 0.7%, and hygroscopicity factor calculated from particle chemical composition ($\kappa_{chem}$), (b) critical diameter of dry particle for activation ($D_c$) at $SS = 0.1\%$ and 0.7%, and (c) the degree of external mixture (($D_{75} - D_{25})/D_c$) at $SS = 0.1\%$ and 0.7%. The definitions of $D_{75}$ and $D_{25}$ are the $D_p$ at which 75% and 25% of the particles are activated at the given $SS$, respectively. Shaded areas represent the values in the range from 25th to 75th percent.

24)     Lines 347f: I disagree with the statement that the seasonal variation of $k_{chem}$ and $k_{ccn}$ are similar for all SS. Adding the $k_{chem}$ values to Fig 6a would make this clearer. The season trend in $k_{ccn}$ is much weaker for small particles and I would claim that $k_{CCN}(SS=0.7\%)$ does not display any trend if its values are 0.19, 0.20, or 0.21, each with a standard deviation of 0.1. This is already a strong sign of a more externally mixed aerosol population during winter and spring. It also shows that $k_{chem}$ is not representative for the smaller particles at this location. See also Specific Comment 28 and 29.

Response: Thanks for the comment. We revised the original Figure 6 (now Figure 5) with adding monthly variation of $\kappa_{chem}$. The season trend in $\kappa_{CCN}$ was exactly much weaker for small particles (high SS) compared to that for larger particles (low SS) and $\kappa_{chem}$. In manuscript, the statement has been revised as follows.

"The seasonal variation of $\kappa_{CCN}$ at $SS$ of 0.1% is similar to that of $\kappa_{chem}$, whereas the seasonal trend in $\kappa_{CCN}$ is much weaker at $SS = 0.7\%$. Essentially, the relationship between $\kappa_{CCN}$ and $SS$ is determined by the $\kappa_{CCN}$ vs. $D_p$ relationship. The $\kappa_{CCN}$ at $SS$ of 0.1% and 0.7% correspond to the median $D_c$ (i.e., $D_p$) of 176 and 54 nm, respectively. As the ACSM is sensitive to particle mass rather than number concentration, the bulk composition is dominated by the contribution of the larger particles. In the median volume size distribution of particle, the peak diameter was at ~300 nm (Poulain et al., 2020). Thus, $\kappa_{chem}$ may be representative for the larger particles rather than for the smaller particles. Owing to the positive correlation between $\kappa$ and $D_p$ (Figure 6a), the $\kappa_{chem}$ representing for the larger particles could be greater than the $\kappa_{CCN}$ for the smaller particles." In lines 323 to 333.

[Figure]

Figure 5. Monthly variations of (a) hygroscopicity factor calculated from monodisperse CCN measurements ($\kappa_{CCN}$) at supersaturation ($SS$) of 0.1% and 0.7%, and hygroscopicity factor calculated from particle chemical composition ($\kappa_{chem}$), (b) critical diameter of dry particle for activation ($D_c$) at $SS$ = 0.1% and 0.7%, and (c) the degree of external mixture (($D_{75}$ − $D_{25}$)/$D_c$) at $SS$ = 0.1% and 0.7%. The definitions of $D_{75}$ and $D_{25}$ are the $D_p$ at which 75% and 25% of the particles are activated at the given $SS$, respectively. Shaded areas represent the values in the range from 25th to 75th percent.

25)    Line 339f: k also varies with composition!

Response: Yes, k varies with composition. We clarified that in the text as follows: "Affected by the variations of particle composition, these two parameters are not constant and both vary with particle size and season." in lines 316 to 317.

26)    Line 379: "non-urban locations" Is the point here that the particles are away from strong localised sources? Or is this about the type of aerosol (e.g., anthropogenic vs biogenic)?

Response: Thanks for your reminder. We have checked the reference and the "non-urban locations" should be "rural locations". However, we have changed the explanation for the less internally mixed particles in winter and the original sentence

has been removed.

27) line 384: This is another example where the manuscript has a lot of description and very little discussion. What is that local pollution? How does it explain the observations? Would this local pollution have varying effects depending on the particle size? Why is this pollution more important in Winter?

Response: Thanks for the comment. The answers are as follows:

a) Local pollution (100 km around) in winter is dominated by liquid fuel, biomass, and coal combustions mostly for house heating (e.g., van Pinxteren et al., 2016, DOI: 10.1039/c5fd00228a) which is more important in cold seasons. These local/regional emissions are also mixed with long-range transport aerosol particles that are quite important during eastern wind.

b) The local emissions mixed with the long-range transport aerosol particles could increase the degree of external mixing.

c) Yes, the local pollution has varying effects depending on the particles size as shown in van Pinxteren et al. (2016, Figure 5).

In manuscript, the statement (lines 346 to 358) has been revised as follows:

"In summer, the less contribution from anthropogenic emissions and the faster aging process as well as SOA formation caused by atmospheric chemistry certainly contribute to make particles more internally mixed. Changes in organic aerosol (OA) composition can be found in Crippa et al. (2014), Poulain et al. (2014), and Chen et al. (2022). In cold seasons, the local pollution (100 km around) is dominated by liquid fuel, biomass, and coal combustions mostly for house heating (van Pinxteren et al., 2016). During winter long-range transport from the eastern wind bring to the station continental air masses which are strongly influence by anthropogenic emissions (in opposition to western marine air masses). These particles are a mixture of different anthropogenic sources emitted all along the transport as well as including some local and regional sources (most house heating). All of them at different aging state cause the overall particles more externally mixed."

28) lines 385-391: This important explanation needs to come much earlier in the text since it is not only relevant for the IQR/$D_C$ vs DP relationship but also for all discussion related to the size resolved CCN measurements. Especially, when comparing

with $k_{chem}$. As the ACSM is sensitive to mass (and not number) concentration, the bulk composition is dominated by the contribution of the larger particles. Thus, $k_{chem}$ may not be representative for the smaller particles (higher SS) which is exactly what Fig 6 shows.

Response: Thanks for your suggestion. We have stated this statement earlier, which has been moved to the beginning of the Section 3.2 where explain that $k_{chem}$ may not be representative for the smaller particles (higher SS). In manuscript, the statements have been revised as follows:

"Essentially, the relationship between $\kappa_{CCN}$ and *SS* is determined by the $\kappa_{CCN}$ vs. $D_p$ relationship" in lines 324 to 325.

"As mentioned above, $\kappa_{CCN}$ (and $(D_{75} - D_{25})/D_c$) vs. $D_p$ relationships determine the relationship between $\kappa_{CCN}$ (and $(D_{75} - D_{25})/D_c$) and *SS*. Monodisperse CCN measurements provide the size-resolved $\kappa$ and $(D_{75} - D_{25})/D_c$. At a given *SS* condition, $\kappa_{CCN}$ represents the $\kappa$ of particles at $D_p = D_c$, and the same is true for $(D_{75} - D_{25})/D_c$." in lines 359 to 362.

29)     Lines 392-410: Here the authors again just describe the observations without making the interesting connections. The authors do not draw the connection between the change in DP dependence of $k_{CCN}$ and the change in mixing state (see also Main Comment 3). The higher sensitivity of $k_{CCN}$ to DP during spring and winter is not an intrinsic property, but it is the direct result of a more externally mixed aerosol population with size dependent composition. In spring and winter, it is more important which part of the size distribution is probed by the CCN measurement because the particle composition changes more with size than in the other seasons. Now, the authors should think about why this is the case? What causes this stronger size dependence of the particle composition? And what does it mean that the IQR/DC vs Dp relationship is much shallower in spring than in winter?

Response: Thanks for your comment.

a) We have monodisperse CCN measurements - meaning we can explore the mixing state of particles of a fixed size. In other words, external mixing would mean particle with the same size can have different chemical composition. In this study, the definition of externally mixed particles is that particles with the same size can have different chemical composition rather than composition varies at different

sizes. We clarified that in the text (lines 225 to 228) as follows:

"Internal mixture implies that all particles with any given dry size have equal $\kappa$ with $(D_{75} - D_{25})/D_c = 0$, whereas a distribution of different $\kappa$ at a given particle size can be observed for externally mixed aerosol with higher $(D_{75} - D_{25})/D_c$ values. Note that the particle composition varying at different sizes is not defined as external mixing in this study."

b) Exactly, the size dependence of the particle composition (kccn) is weaker in summer than other seasons. The reasons are as follows. In winter we have a mixture between anthropogenic sources and aged particles leading to a size dependent chemical composition (e.g., van Pinxteren et al., 2016, DOI: 10.1039/c5fd00228a). In summer, the anthropogenic emissions linked to house heating a strongly reduce which affect the smaller particles, and the dominant < 100 nm particles is associated to NPF and SOA formation. NPF is a complex process which is depending on the availability of condensing material (H2SO4, and organic), as well as pre-existing particles (coagulation and condensation sink parameters). Therefore, same condensing material on the gas phase can either condense on pre-existing particles (usually larger than 100 nm and then detected by ACSM) or lead to NPF formation. A direct consequence of it, is a probable smaller effect of the size dependent chemical composition of the particles. This might explain why kCCN at SS 0.1 and 0.7 % are closer in summer and why kchem better explain kCCN at different SS in summer than winter. We clarified that in the text (lines 367 to 377) as follows:

"Compared to the cold seasons, the anthropogenic emissions linked to house heating strongly reduce in summer which affect the smaller particles, and the dominant small particles ($D_p < 100$ nm) are associated to NPF and the SOA formation. NPF is a complex process which depends on the availability of condensing material ($H_2SO_4$ and organic), as well as pre-existing particles (coagulation and condensation sink parameters). Therefore, same condensing material on the gas phase can either condense on pre-existing particles (usually larger than 100 nm and then detected by ACSM) or lead to NPF formation. A direct consequence of it is a probable smaller effect of the size dependent chemical composition of the particles. This might explain why $\kappa_{CCN}$ at $SS$ of 0.1% and 0.7 % are closer, i.e., the weaker sensitive of $\kappa_{CCN}$ to $D_p$ in summer."

[Figure]

Figure 6. (a) Relationship between the hygroscopicity factor calculated from monodisperse CCN measurements ($\kappa_{CCN}$) and particle diameter ($D_p$), and (b) degree of external mixture (($D_{75} - D_{25}$)/$D_c$) vs. $D_p$ at each season. The definitions of $D_{75}$ and $D_{25}$ are the $D_p$ at which 75% and 25% of the particles are activated at the given $SS$, respectively. Red lines are power-law fits. Dots represent the median values. Shaded areas represent the values in the range from 25th to 75th percent.

30)      Chapter 3.3: The introduction of the prediction methods is currently a little bit confusing and needs improvement. From the text, I did not understand what the main difference is between the two categories. I eventually work out that the schemes in the first category can be used for data obtained from polydisperse CCN measurements when only NCCN is measured while the second category is based on using some sort of k value to calculate DC. In addition, readability could be enhanced by labelling the schemes using the categories, e.g., N1, N2, and K1, K2, K3.

Response: Thanks for your comment. There are five schemes and can be divided into two categories. The 1st category uses the NCCN (AR) – SS empirical formula which obtained from the polydisperse CCN measurements when only NCCN is measured. The 2nd category uses the real-time PNSD combined with the parameterized k(Dc). The labellings of N1, N2, and K1, K2, K3 are used as suggested. The introduction of the prediction methods has been revised as follows.

"Table 3 introduces the five schemes, which can be summarized into two categories. From polydisperse CCN measurements, the $N_{CCN}$ (AR) and $SS$ relationships can be obtained, and their fitting results can be used to predict $N_{CCN}$ at the given $SS$ conditions, which belongs to the 1$^{st}$ category, corresponding to the N1 and N2 schemes in Table 3, respectively. Compared to CCN measurements, it is generally more common and simpler to obtain the PNSD measurements. Thus, we usually predict $N_{CCN}$ using the real-time PNSD combined with the parameterized $\kappa(D_c)$, which belongs to the 2$^{nd}$ category. The 2$^{nd}$ category includes the last three schemes (K1, K2, and K3) in Table 3, but they vary in assuming $\kappa$. The K1 scheme used a fixed $\kappa$ of 0.3 without temporal and size-dependent variations, as recommended for continental aerosol (Andreae and Rosenfeld., 2008), which is also the median value of $\kappa_{chem}$ over all data setting at Melpitz. The K2 scheme used the bulk $\kappa_{chem}$ calculated from aerosol chemical composition, which is also non-size-dependent but changes over time. The K3 scheme used the $\kappa$ - $D_p$ power-law fit results shown in Figure 6a, which are size-dependent without temporal variations at each season." In lines 388 to 403.

Table 3. Introduction of five activation schemes. The meaning of the abbreviation can be found in Notation list.

| Category | Scheme | Introduction |
|---|---|---|
| 1$^{st}$ category: $N_{CCN}$ - $SS$ or $AR$ - $SS$ empirical fit | N1 | $N_{CCN}$ - $SS$ power-law fits shown in Table 3 |
| | N2 | Real-time $N_{aero}$ combined with $AR$ - $SS$ power-law fits shown in Table 3 |
| 2$^{nd}$ category: Real-time PNSD combined with the parameterized $\kappa$ | K1 | Real-time PNSD combined with a constant $\kappa$ of 0.3 |
| | K2 | Real-time PNSD combined with the real-time bulk $\kappa_{chem}$ |
| | K3 | Real-time PNSD combined with $\kappa$ - $D_p$ power-law fits shown in Figure 6a |

31)    lines 439f: RD is a single value for each case in Fig 8. But |predicted NCCN - measured NCCN |/ measured NCCN provides a number for each measurement point. I

guess these values were summed? Please, correct this equation and write it as its own as a proper equation and not "in-line"

Response: Thanks for your reminder. Exactly, the RD corresponds to a number for each measurement point. The median RD value was used to quantify the deviation between measurements and predictions. It has been revised as follows.

"The relative deviation (RD) equals the ratio of the absolute difference between the predicted $N_{CCN}$ and the measured one to the measured $N_{CCN}$,

$$RD = \frac{|predicted\ N_{CCN} - measured\ N_{CCN}|}{measured\ N_{CCN}}. \tag{6}$$

The median RD was used to quantify the deviation between predictions and measurements of each scheme." In lines 412 to 416.

32)     Lines 443-454 summarise the prediction quality of the 5 schemes. This section is good. But then that is followed by yet another very detailed description of numbers that are presented in Fig 8 and 9 (over 3 and a half pages!). This is extremely tedious to read and again the important conclusions are buried under mountains of numbers. The authors need to trim this section.

Fig 8 and 9 show the same information simply from a different angle. They need to decide which of the figures works better and put the other in the SI. Instead of providing so many numbers for each scheme to say again that the prediction is better/worse, they should focus on the main improvements and features of the schemes which lead to the better worse prediction. E.g., scheme 1 calculates 1 NCCN value for each SS. Thus, the spread depicted by the boxplot in Fig 9a simply reflects the standard deviation of the measurements as shown in Fig 3c (or rather the Interquartile range). For the category 2 schemes, the point is how well the parameterised k describes the measured kCCN value. This then explains why some seasons are predicted better than others (i.e. if the measured kCCN are closer to the value set in the scheme).

Response: Thanks for your comment. The original Figure 8 (now Figure 7) was kept and the original Figure 9 was moved to SI (Figure S7). In manuscript, the statement of the 5 schemes' evaluation has been reduced to less than one page, as follows:

"As shown in Figure 7, the N1 and N2 schemes only provide rough estimates of the $N_{CCN}$ which is reflected in the high median RD. The results for N1 and N2 schemes are

similar in that they both predict the overall mean $N_{CCN}$ well (slopes of approximately 1.0) but with large median RDs. Compared to N1 scheme, the N2 scheme is better because of the lower median RD. Compared to the 1st category (the N1 and N2 schemes), the 2nd category (the K1, K2, and K3 schemes) predicts $N_{CCN}$ better because of the lower median RD. The results for K1 and K2 are similar in that they both overestimate $N_{CCN}$ by approximately 10% (slopes of approximately 1.1) with similar median RDs. The reason for the $N_{CCN}$ overestimation is that the constant $\kappa$ of 0.3 and the real-time bulk $\kappa_{chem}$ are both greater than the $\kappa_{CCN}$ at each season. In winter, the $\kappa_{CCN}$ was highest and the difference between the $\kappa_{CCN}$ and the parameterized $\kappa$ in K1 and K2 scheme was lowest, causing the best prediction of $N_{CCN}$ among the four seasons. Owing to the largest difference between the $\kappa_{CCN}$ and the parameterized $\kappa$, the $N_{CCN}$ prediction was worst in summer for K1 scheme and in autumn for K2 scheme. The K3 scheme appears to be the best one for $N_{CCN}$ prediction among the five schemes which is reflected in the lowest median RDs and the fit slope of ~1.0 for different seasons. The evaluations of the five schemes for the $N_{CCN}$ prediction at each $SS$ condition and each season are provided in Figure S7 in SI." In lines 417 to 434.

[Figure]

Figure 7. Predicted vs. measured CCN number concentration ($N_{CCN}$) for different seasons. The Predicted $N_{CCN}$ is calculated from five different schemes with a detailed introduction shown in Table 3. Color bar represents the different supersaturation (*SS*) conditions. Black lines are the linear fits. The slope and $R^2$ of the linear regression and the median relative deviation (RD) between the predicted and measured $N_{CCN}$ are shown in each panel. Each row represents the results using the same scheme in different seasons; each column represents the results using different schemes in the same season.

33)    lines 510 – 528 provides a good summary of the performance of the schemes and links the power law values to other observations. But what does it mean that the values for Melpitz are similar to some stations (even urban ones) and not to others (see also Main Comment 4). Either here – or better in the conclusions – this should be

discussed, and the authors should at least speculate what may be causing the similar behaviour at such different sites.

Response: As shown in Figure 8, the $\kappa$ and $D_p$ relationships measured at three rural stations (Melpitz, Xinken, and Vavihill) are similar. In these rural stations, the slope parameter and the coefficient range from 0.25 to 0.32 and 0.052 to 0.07, respectively. Thus, we concluded that the $\kappa$ - $D_p$ power-law fit presented in this study could apply to other rural regions. We notice that the power law for Shanghai was also close to what was found for Melpitz, but now we only state that our power law can be used for other rural places rather than that is entirely different for all urban places studied. And that we can not answer what environmental properties cause the differences in kappa to Dp. We clarified that in the text as follows:

"The $\kappa$ - $D_p$ relationship measured at Melpitz is similar to that measured at other rural regions with similar $\kappa$ - $D_p$ power-law fitting results, e.g., the Vavihill station in Sweden (Fors et al., 2011) and the Xinken station in China (Eichler et al., 2008). Therefore, the $\kappa$ - $D_p$ power-law fit measured at Melpitz could be applied to predict $N_{CCN}$ for these rural regions." in lines 451 to 456 in Chapter 3.3.

"The $\kappa$ - $D_p$ power-law fit presented in this study could apply to other rural regions. However, it may cause considerable deviations for different aerosol background regions." in lines 506 to 507 in Conclusion Section.

"Additionally, the seasonal difference of the $\kappa$ - $D_p$ relationship needs to be considered carefully for $N_{CCN}$ prediction. At Melpitz, if the $\kappa$ - $D_p$ power-law fit measured in summer was used for predicting $N_{CCN}$ in winter, it could cause a 13% underestimation of $N_{CCN}$ in median for all $SS$ conditions. Although the $\kappa$ - $D_p$ relationships are similar measured in rural stations, but when comparing the different urban stations (e.g., shanghai vs. Budapest in Figure 8), these relationships are clearly different and the reasons for the difference are still unclear. Thus, long-term monodisperse CCN measurements are still needed not only to obtain the $\kappa$ - $D_p$ relationships for different regions and for different seasons, but furtherly investigate the reasons for the difference of the $\kappa$ - $D_p$ relationships measured at same type of regions." in lines 513 to 522 in Conclusion Section.

[Figure]

Figure 8. Relationships between the particle hygroscopicity factor ($\kappa$) and diameter ($D_p$) observed at different stations. Lines are the power-law fits of $\kappa$ vs. $D_p$.

34)  Table 3 is very difficult to read. It is next to impossible to compare the parameters as each entry is spread over multiple lines. How about stating the equations in the Table cation and then providing only the parameters and $R^2$ values in the table. If this table stays in the main text, the error function values should be moved to the SI (see Specific Comment 16).

Response: Thanks for the suggestion. The original Table 3 has been revised and moved to SI (Table S2). The power-law fitting results are shown in new Figure 2. The error function shows in Table cation and only the parameters and R2 values shown in the table. The new Table is as follow:

Table S2. Error function fits for the relationships between activation ratio (AR) vs. supersaturation (SS), and CCN number concentration ($N_{CCN}$) vs. SS for different seasons. The equation is y=a+a*erf(ln(x/b)/c), where a, b, and c are parameters remained to be determined. The $a_{AR}$ and $a_{NCCN}$ represent the parameter a in AR vs. SS fitting and $N_{CCN}$ vs. SS fitting, respectively.

| Season | $a_{AR}$ | $a_{NCCN}$ | $b$ | $c$ | $R^2$ |
|---|---|---|---|---|---|
| Spring | 0.50 | 2637 | 0.72 | 2.33 | 0.998 |
| Summer | 0.51 | 3162 | 1.04 | 2.15 | 0.997 |
| Autumn | 0.56 | 2443 | 0.84 | 2.29 | 0.999 |
| Winter | 0.44 | 1624 | 0.29 | 1.83 | 0.999 |
| All | 0.40 | 2199 | 0.59 | 2.25 | 0.998 |

35)     The authors claim that scheme 5 (using the power law k(DP) approximation) provides an improved prediction of $N_{CCN}$. This is true when compared to schemes 1 and 2. But how much does that improvement really matter when looking at schemes 3-5? From 4 to 5, the slope decreases 0.1 on average. So, the 10% overestimation is reduced. How much will that impact, e.g., the calculation of radiative forcing or prediction of precipitation in a climate model? Is that worth the effort? Some people may argue that operating an ACSM from which $k_{chem}$ can be derived, is more feasible in a measurement station than conducting size resolved CCN measurements which are needed to obtain the k(Dp) relationship.

Response:     Thanks for your suggestion. We furtherly evaluate the effects of 10% overestimation in NCCN on radiative forcing and prediction of precipitation.

a)  From the 3$^{rd}$ and 4$^{th}$ scheme to 5$^{th}$ scheme, the slope of the linear fitting decreases 0.1 on average, meaning that the ~10% overestimation of NCCN is reduced. Theoretically, it can reduce 3.2% overestimation of cloud optical thickness, corresponding to global average difference of 1.28 Wm$^{-2}$ (assuming the cloud shortwave cooling effect of 40 Wm$^{-2}$; Lee et al., 1997), which amounts to 32% of the direct radiative forcing from a doubling CO2 (about 4 Wm$^{-2}$). Additionally, the overestimation in NCCN leads to underestimate the strength of the autoconversion process in cloud, thereby suppressing precipitation. The methods are shown in SI. Although ACSM measurements can derive kchem and thus predict NCCN, size-resolved CCN measurements are still important to obtain the k-d relationship and thus improve the prediction of NCCN and climate. In manuscript, it has been revised as follows.

    "The K3 scheme provides an improved prediction of $N_{CCN}$, which is obvious when compared to N1 and N2 schemes. Compared to K1 and K2 schemes, the K3 scheme

reduced approximately 10% overestimation of $N_{CCN}$ because the fitting slope decreased ~0.1 on average. We simply evaluate the effects of the 10% overestimation in $N_{CCN}$ on predictions of cloud radiative forcing and precipitation. The methods are in Text S2 in SI and Wang et al. (2019). Essentially, an overestimation of $N_{CCN}$ leads to overestimate the number concentration of cloud droplet ($N_C$) in models. Theoretically, it can reduce 3.2% overestimation of cloud optical thickness, corresponding to global average difference of 1.28 Wm$^{-2}$ when assuming the cloud shortwave cooling effect of 40 Wm$^{-2}$ (Lee et al., 1997), which amounts to approximately one-third of the direct radiative forcing from a doubling $CO_2$. Additionally, the overestimation in $N_{CCN}$ (and $N_C$) leads to underestimate the strength of the autoconversion process in cloud (Liu et al., 2006), thereby suppressing precipitation. Therefore, although ACSM measurements can derive $\kappa_{chem}$ and thus predict $N_{CCN}$, the monodisperse CCN measurements are still important to obtain the $\kappa$ - $D_p$ relationship and thus improve the predictions of $N_{CCN}$ (and $N_C$) and climate." In lines 435 to 450.

b) The methods shown in SI (Text S2) are as follows:

"**Text S2. Method for evaluating the impact of $N_{CCN}$ overestimation on cloud radiative forcing and autoconversion process**

Cloud optical thickness ($\tau$) can be expressed by (Stephens, 1984)

$$\tau \approx \frac{3}{2} W r_e^{-1}, \tag{1}$$

where $W$ is the liquid water path, $r_e$ is the effective radius of cloud droplets. Meanwhile $r_e$ is proportional to the volume weighted mean radius of cloud droplets ($r_v$) (Bower and Choularton, 1992) and can be expressed by

$$r_e = \beta \left( \frac{3q}{4\pi \rho_w N_c} \right)^{1/3} = \beta r_v, \tag{2}$$

where $\beta$ is the scaling factor, $q$ is the cloud liquid water content, $\rho_w$ is the density of water, and $N_c$ is the number concentration of cloud droplet. Here, to focus on the effect of $N_c$ on $r_e$, $\beta$ is specified as a fixed parameter, i.e., ignoring the dispersion effect, as assumed in many climate models (Quaas et al., 2004).

According to Liu et al. (2004, 2005), parameterization of the autoconversion process can be expressed by

$$P = TA \times P_0, \tag{3}$$

where $P$ is the autoconversion rate, $P_0$ is the rate function describing the conversion rate after the onset of the autoconversion process, and $TA$ is a function describing the threshold behavior of the autoconversion process. Meanwhile, $TA$ can be expressed by

$$TA = \left[\frac{\int_{r_c}^{\infty} r^6 n(r)dr}{\int_{0}^{\infty} r^6 n(r)dr}\right]\left[\frac{\int_{r_c}^{\infty} r^3 n(r)dr}{\int_{0}^{\infty} r^3 n(r)dr}\right], \tag{4}$$

where $r$ is the droplet radius, $n(r)$ is the cloud droplet size distribution, and $r_c$ is the critical radius of autoconversion process. The $TA$ ranges from zero to one, with a larger $TA$ indicating a greater probability that the collision process occurs in clouds. Liu et al. (2006) derived the analytical expression of $r_c$ as follows:

$$r_c \approx 4.09 \times 10^{-4} \beta_{con}^{1/6} \frac{N_c^{1/6}}{q^{1/3}}, \tag{5}$$

where $\beta_{con} = 1.15 \times 10^{23}$ s$^{-1}$ is an empirical constant.

Essentially, an overestimation of $N_{CCN}$ leads to overestimate $N_c$ in models. From the 3rd and 4th scheme to 5th scheme, the slope of the linear fitting decreases 0.1 on average, meaning that the ~10% overestimation of $N_{CCN}$ and $N_c$ is reduced. According to equations 1 and 2, it can reduce 3.1% underestimation of $r_e$ when assuming the

constant $q$ and $\beta$, thereby reducing 3.2% overestimation of $\tau$. When assuming the global

average cloud shortwave cooling effect is 40 Wm$^{-2}$ (Lee et al., 1997), the corresponding

difference is 1.28 Wm$^{-2}$, which amounts to 32% of the direct radiative forcing from a

doubling CO$_2$ (about 4 Wm$^{-2}$). Additionally, according to the equations 4 and 5, it can

reduce the overestimation of $r_c$ thus the underestimation of $TA$, indicating that the

underestimation of the strength of autoconversion process can be reduced."

36)    The k value used in scheme 3 is clearly too high. Have the authors tried to run

this scheme using the average kCCN value for their data set? How "good" is scheme 3

then?

Response: Scheme 3 sets a constant k of 0.3, which is a suggested k value for

continental regions (Andreae and Rosenfeld, 2008). We can set the mean kCCN value

for the all datasets, but the results on NCCN prediction cloud be worse than the K3 (K-

Dp power-law fitting), especially at SS of 0.1% and 0.7%. Additionally, we also

evaluated the $N_{CCN}$ predictions using the seasonally mean value of $\kappa$ over $D_p$ of 100 to

200 nm because the size dependence of $\kappa$ mainly occurs at $D_p$ of ~40 to 100 nm. The

results are shown in original Figure 10. Now, it has been moved to SI (Text S3 and

Figure S8), as follows:

**"Text S3. $N_{CCN}$ predictions using the seasonally mean value of $\kappa$ over $D_p$ of 100 to 200 nm**

The main size dependence of $\kappa$ occurs at $D_p$ of ~40 to 100 nm as shown in Figure 6a, which

would be for $SS$ larger than 0.2%. At $D_p$ of 100 to 200 nm corresponding to $SS$ less than 0.2%,

$\kappa$ almost stays constant. The mean value of $\kappa$ at $D_p$ of 100 to 200 nm is close to 0.3 for spring

and winter, and that's where deviations in Figure S7c are small. However, the mean value of $\kappa$

at $D_p$ of 100 to 200 nm overestimates the $\kappa$ for $SS$ larger than 0.2% at each season. We further

compare the $N_{CCN}$ predictions between using the seasonally mean value of $\kappa$ over $D_p$ of 100 to

200 nm and the $\kappa$ - $D_p$ power-law fit. As shown in Figure S8, at $SS = 0.1$ and 0.2%, the

seasonally mean $\kappa$ value over $D_p$ of 100 to 200 nm and $\kappa$ - $D_p$ power-law fit both predict the

$N_{CCN}$ well at each season, while the mean $\kappa$ value over $D_p$ of 100 to 200 nm leads to a significant

overestimation of $N_{CCN}$ within 10% on average at $SS = 0.3$, 0.5, and 0.7%. Therefore, to predict

the $N_{CCN}$ at a relatively low $SS$ of less than 0.2% (e.g., in fog and shallow stratiform cloud), the

mean $\kappa$ value over $D_p$ of 100 to 200 nm also works well."

[Figure]

Figure S8. Predicted vs. measured CCN number concentration ($N_{CCN}$) at different supersaturation ($SS$) conditions for different seasons. (a) results at $SS = 0.1$ and 0.2%; (b) results at $SS = 0.3$, 0.5, and 0.7%. Red cross represents the predicted $N_{CCN}$ using mean hygroscopicity factor ($\kappa$) over particle diameter ($D_p$) of 100 to 200 nm, while the blue cross represents the predicted $N_{CCN}$ using power-law fit of $\kappa$ and $D_p$. Red and blue lines are the linear fits.

37)     The Conclusions chapter is simply a summary of the previous chapters, repeating many of the numbers that were already stated. These are not "conclusions" as in interpretations or putting their findings into context. There are many things the authors bring up in this chapter. These are a few ideas that spring to my mind (some are already mentioned in other Specific Comments):

a.              How much their improved NCCN prediction may improve modelling results?

Response: The power-law scheme (K3) provides an improved prediction of $N_{CCN}$, which is obvious when compared to N1 and N2 schemes. From the K1 and K2 scheme to K3 scheme, the slope of the linear fitting decreases 0.1 on average, meaning that the ~10% overestimation of NCCN is reduced. Theoretically, it can reduce 3.2% overestimation of cloud optical thickness, corresponding to global average difference of 1.28 $Wm^{-2}$ (assuming the cloud shortwave cooling effect of 40 $Wm^{-2}$; Lee et al., 1997), which amounts to 32% of the direct radiative forcing from a doubling CO2

(about 4 Wm$^{-2}$). Additionally, the overestimation in NCCN leads to underestimate the strength of the autoconversion process in cloud, thereby suppressing precipitation.

b.      How much would using the values from the "wrong" season affect NCCN predictions? or from a wrong location (E.g. using the Budapest values for the Melpitz data set)

Response: We furtherly evaluate these effects coming from "wrong" location and seasons. Using the $\kappa$ - $D_p$ power-law fit measured in urban Budapest (Salma et al., 2021) for predicting Melpitz $N_{CCN}$, it could cause a 39% underestimation of $N_{CCN}$ in median for all $SS$ conditions. Additionally, the seasonal difference of the $\kappa$ - $D_p$ relationship needs to be considered carefully for $N_{CCN}$ prediction. At Melpitz, if the $\kappa$ - $D_p$ power-law fit measured in summer was used for predicting $N_{CCN}$ in winter, it could cause a 13% underestimation of $N_{CCN}$ in median for all $SS$ conditions.

c.      If the k ~ DP prediction works so well, do we really need continuous CCN measurements? Wouldn't it be enough to determine the representative k ~ DP fit for a few representative locations?

Response: We still need continuous CCN measurements, especially for the measurements of monodisperse CCN. As shown in Figure 8, the $\kappa$ - $D_p$ relationships are similar measured in rural stations, but when comparing the different urban stations (e.g., shanghai vs. Budapest), the $\kappa$ - $D_p$ relationships are clearly different. Thus, long-term monodisperse CCN measurements are still needed to not only obtain and correct the $\kappa$ - $D_p$ relationships for different regions and for different seasons, but also furtherly investigate the reasons for the difference of the $\kappa$ - $D_p$ relationships measured at same type of regions.

[Figure]

Figure 8. Relationships between the particle hygroscopicity factor ($\kappa$) and diameter ($D_p$) observed at different stations. Lines are the power-law fits of $\kappa$ vs. $D_p$.

d.              Or playing devil's advocate: Since the $k_{chem}$ based $N_{CCN}$ prediction is much better than the ones based on $N_{CCN}$ ~ SS or AR ~ SS, wouldn't it be better to improve composition measurements?

Response: We can only answer by the positive to this question. For example, by improving our estimation of the korg (this dependent values). In the text, we underline the importance of measurement of particle chemical composition.

"Finally for the purpose of predicting $N_{CCN}$, the measurements of monodisperse CCN and particle chemical compositions are more expected, compared to the polydisperse CCN measurements."

e.              regarding the mixing state: Why is the mixing state different between seasons?

Response: At Melpitz, the local pollution (100 km around) in winter is dominated by liquid fuel, biomass, and coal combustions mostly for house heating (e.g., van Pinxteren et al., 2016) which is more important in cold seasons. These local/regional emissions are also mixed with long-range transport aerosol particles that are quite important during eastern wind, which could cause the less internally mixing of aerosol particles

in cold seasons.

f.                    Why is k(DP) and IQR/DC(Dp) different between the seasons?

Response: As shown in Figure 6, as Dp increases, k increases and IQR/DC decrease at each season. It means that the large particles have relatively high k and high degree of internal mixing. In summer, the sensitivities of k and IQR/DC to Dp are both lowest among the four seasons. The reason for less sensitivity of k to Dp has shown in Comment 29. For the less sensitivity of IQR/DC to Dp, it could be related to the less mixing between the local emissions and long-range transport aerosol particles.

[Figure]

Figure 6. (a) Relationship between the hygroscopicity factor calculated from monodisperse CCN measurements ($\kappa_{CCN}$) and particle diameter ($D_p$), and (b) degree of external mixture (($D_{75} - D_{25})/D_c$) vs. $D_p$ at each season. The definitions of $D_{75}$ and $D_{25}$ are the $D_p$ at which 75% and 25% of the particles are activated at the given $SS$, respectively. Red lines are power-law fits. Dots represent the median values. Shaded areas represent the values in the range from $25^{th}$ to $75^{th}$ percent.

Considering the comments above, the Conclusion Section was rewritten as follows:

**"4. Conclusions**

[revised manuscript text omitted]

38)     line 565: these things are also linked to the highest kCCN and the widest spread in kCCN (i.e., least internally mixed)

Response: Thanks for the comment. In summer, the narrowest spread in kccn could not directly relate to the highest AR and NCCN values. In summer, the highest $N_{aero}$, smallest $GMD$, and lowest $\kappa_{chem}$ all contribute to the lowest AR and $N_{CCN}$ among the four seasons, and the reverse holds true in winter. Additionally, in summer, the steepest PNSD in 40-200 nm size range and the lowest $\kappa_{chem}$ causes the strongest sensitivity of $N_{CCN}$ and AR to $SS$ even though the spread in $\kappa_{CCN}$ is narrowest.

We clarified that in the text (lines 475 to 479) as follows:

"In summer, the highest $N_{aero}$, smallest $GMD$, and lowest $\kappa_{chem}$ all contribute to the lowest AR and $N_{CCN}$ among the four seasons, and the reverse holds true in winter. Additionally, in summer, the steepest PNSD in 40-200 nm size range and the lowest $\kappa_{chem}$ causes the strongest sensitivity of $N_{CCN}$ and AR to $SS$ even though the spread in $\kappa_{CCN}$ is narrowest."

**Language:**

General: In multiple locations, main clauses are attached with ";" to each other. While this is grammatically possible, it decreases readability by creating "monster sentences". Simply use a full stop and start the second main clause. Examples: line 28ff: second sentence starts at "the seasonal mean activation ratio…"

Response: Thanks for your comment. We have rewritten these sentences. Examples are as follows.

a)  "Long-term measurements of aerosol particle activation help to understand the AIEs and narrow down the uncertainties of AIEs simulation. However, they are still

scarce." (Lines 15 to 17).

b) "For instance, Sihto et al. (2011) suggested an average $\kappa$ of 0.18 to predict the CCN activation well in boreal forest conditions in Hyytiälä, Finland. A fixed $\kappa$ of 0.31 suffices to calculate the $N_{CCN}$ in a suburban site located in the center of the North China Plain (Wang et al., 2018a). The mean $\kappa$ is 0.5 in a near-coast and rural background station (CESAR Tower) in Netherlands (Schmale et al., 2018). The median $\kappa$ ranges from 0.02 to 0.16 at $SS = 0.1{-}1.0\%$ in an urban background site in Budapest, Hungary (Salma et al., 2021)." (Lines 71 to 77).

c) "Freshly formed particles are about 1 nm in diameter (Kulmala et al., 2012), which must grow to tens of nanometers in diameter to serve as the effective CCN at a relatively high $SS$ of ~1% (Dusek et al., 2006) and even larger than 200 nm to be efficient at $SS$ less than 0.1% (Deng et al., 2013)." (Lines 87 to 90).

d) "Most of the observations lasted 1–2 months or even less, mainly focusing on the effects of short-term weather processes or pollution events on aerosol particle activation" (Lines 104 to 106).

line 15 "measurements on aerosol particle activation" -> of

Response: It has been corrected.

line 20 "improving predictions": predictions of what?

Response: predictions of number concentration of cloud condensation nuclei

line 29 "twice higher" -> either "twice as high as" or "two times higher than"

Response: It has been corrected.

"At $SS = 0.1\%$, the seasonal median $N_{CCN}$ and activation ratio (AR) are 1.6 and 2.3 times higher than the summer values, respectively." (Line 27).

line 35: "the power law function" sounds as if this is a specific function with the name 'power law' change to "a power law function"

Response: Thanks for your comment. It has been corrected and also for other places in the text.

line 44 "activated cloud droplets" -> remove activated. The particles get activated to grow to cloud droplets.

Response: It has been corrected.

line 72f "should be underlined" -> no, it should not be underlined (unless you speak German ;-). change to"should be emphasised"

Response: Thanks for your comment. It has been changed to "should be emphasized".

line 137 "mixing state degree" -> sounds weird either use "degree of mixing" or

"mixing state"

Response: Thanks for your comment. It has been changed to "mixing state"

line 149f "can be found in for example, Poulain et al 2020" -> "can be found, for example, in Poulain et al 2020.

Response: It has been corrected.

line 153: "Figure 1 demonstrates" -> it is not the Figure that does something. Better use "Figure 1 shows/depicts"

Response: Thanks for your comment. It has been changed to "Figure 1 shows".

line 160: "within the diameter ranging from 5 to 800nm" -> "with a diameter range of 5 – 800 nm"

Response: It has been revised.

line 170f "respectively pass through" -> "respectively" cannot be used like that. This is also an example for a ";" monster sentence. Simply start a new sentence. "… monodisperse particle fraction. After the DMA, the flow was split to pass through a CPC […] and a CCN counter […]."

Response: Thanks for your comment. This sentence (Lines 160 to 168) has been revised as follows:

"For simultaneous measurement of particle and CCN number size distributions, dried aerosol particles were passed through the bipolar charger to establish charge equilibrium (Wiedensohler, 1988) and then through a differential mobility analyzer (DMA) for selecting a monodisperse particle fraction. After the DMA, the flow was split to pass through a condensation particle counter (CPC, model 3010, TSI) to measure the total number concentration of the selected monodisperse condensation nuclei ($N_{CN}$) and through a cloud condensation nuclei counter (CCNC, model 100, Droplet Measurement Technologies; Roberts and Nenes, 2005) to measure the $N_{CCN}$."

line 200 "was firstly corrected" -> was first corrected

Response: It has been corrected.

line 203: "thus they are falsely selected in the DMA" -> they are selected in the absolute correct way. It is the assigned diameter that is incorrect. Simply remove this phrase.

Response: Thanks for your comment. It has been removed.

line 204 "For this was corrected" -> "To correct for this, the fraction of multiple charged particles […] was subtracted […]"

Response: Thanks for your comment. This sentence (Lines 206 to 208) has been corrected as follows:

"To correct for this, the fraction of multiple charged particles as determined from the D-MPSS measurements was subtracted from each value of $N_{CCN}/N_{CN}$ in AF."

line 216 "rather than an intermittent mutation" -> do you mean "rather than displaying (?) an intermittent mutation"?

Response: Thanks for your comment. It has been changed to "rather than displaying an intermittent mutation".

line 235 "determined" -> determined feels a bit strong here. Maybe better "derived" since this is a approximation of the true k value?

Response: Thanks for your comment. It has been changed to "derived".

line 278 "…gradually peaks in summer…" -> I do not know what "gradually peaks" means in this context

Response: Thanks for your comment. We originally wanted to say that as SS increases, the Dc decreases and the CCN number size distribution peaks in the small particles, which is more noticeable in summer than other seasons. Now, it has been deleted.

line 282f "the power-law and the error function" -> should be "a".

Response: It has been corrected.

line 285 "because of more parameters" -> "due to the higher number of parameters".

Response: Thanks for your comment. This sentence has been removed.

line 298 "CCN number size distribution" -> missing "The"

Response: It has been added.

line 382: What is meant by "aerosol cluster"?

Response: It means an air mass containing an aerosol population. Now it has been removed.

line 415: "two categories of NCCN prediction approach" -> "approaches" or better "can be divided into two categories"

Response: Thanks for your comment. This sentence (Lines 388 to 389) has been changed as follows:

"Table 3 introduces the five schemes, which can be summarized into two categories."

line 417 and later "category approach" -> only "category" without approach

Response: It has been corrected.

line 444f "provide rough estimates on account of the pretty high RD" weird. RD is not causing the rough estimate it is the consequence. Better "provide rough estimate which is reflected in the high RD"

Response: Thanks for your comment. Yes, this is what we mean. It has been corrected.

line 454: "…Figure 9 further evaluates the model…"It is not the Figure that evaluates the models.

Response: Thanks for your comment. It should be that we further evaluate the model and the results are shown in Figure 9. Now, the original Figure 9 has been moved to SI and this sentence has been removed in the text.

line 458 "results are much uncertain" -> "the results have a high uncertainty"

Response: Thanks for your comment. This sentence has been removed in the text.

line 466f "the prediction results remain a high uncertainty" -> ??? "the uncertainty of the prediction results remain high"???

Response: It should be that the prediction results have a high uncertainty. Now this sentence has been removed in the text.

line 475: "NCCN is overestimated at assuming a constant k" -> "when assuming" line

Response: Thanks for your comment. This sentence has been removed in the text.

478f: "the largest median overestimation reaches to 30%" -> no "to"

Response: Thanks for your comment. This sentence has been removed in the text.

line 485f: "the 3$^{rd}$ scheme has better predictions on NCCN" ->"provides better predictions of NCCN"

Response: Thanks for your comment. This sentence has been removed in the text.

line 510: "gradually changes" really? I would not call the big improvement from scheme 1 to 2 to 3 "gradual". For the changes going from schemes 3-4-5, gradual is the correct term.

Response: Thanks for your comment. Yes, for the changes going from scheme 3 to 4 to 5, gradual is the correct term rather than from scheme 1 and 2 to 3. Now this sentence has been removed in the text.

---

## Author Response (AR2)

As the Reviewer#2 commented, the authors present a 4 year-long size resolved CCN measurement data at a central European rural background station. This type of data is very valuable for the atmospheric science community. The authors have focused on analyzing the seasonal trends and test multiple approaches to predict the number of activated particles NCCN. Based on the suggestions and comments from reviewer#2, the authors have made corresponding revisions and edits on the manuscript and making the current version of the paper being improved.

**Comment 1:**

However, I still have some concerns here, since the data are based on a 4-year measurement, I would like to suggest the authors may present a yearly trend or changes of the aerosol CCN activation, which is not shown in the revised manuscript. It would be interesting to study and analyze such trends over a regional scale.

Response: Thanks for your comment. We further investigated the yearly trends of CCN activation properties. The CCN number concentration ($N_{CCN}$) and hygroscopicity factor calculated from monodisperse CCN measurements ($\kappa_{CCN}$) measured at supersaturation ($SS$) of 0.1% and 0.7% are chosen to represent the CCN activation characteristics. The results are shown in the following Figure S7. The yearly trends in $N_{CCN}$ and $\kappa_{CCN}$ are not significant (without significant increase or decrease trends) during the measurements from August 2012 to October 2016. However, it is interesting to see that the $N_{CCN}$ measured in 2015 was significantly lower than it measured in the other four years. One of the reasons could be that the CCN measurements in 2015 concentrated in summer and autumn, lacking measurements in the spring and winter months (Figure S1). As shown in Figure 3b, $N_{CCN}$ measured at summer and autumn are lower than those measured in spring and winter due to its seasonal trend, causing the lowest $N_{CCN}$ values in 2015. Thus, the CCN measurements in 2015 may not be representative of the CCN characteristics of the whole year. Similarly, the 2012 measurements may not be representative of year-round CCN characteristics because of the lacking spring and summer measurements. Additionally, it is also hard to see the yearly trends using the only 4-year data. In order to investigate the yearly trends of CCN activation characteristics, longer periods measurements are required.

We clarify it in the text and SI as follows.

"Additionally, no significant yearly trends of the CCN activation characteristics are found during the 4-year measurements and the results are provided in SI (Text S2 and Figure S7)." Lines 314 to 316.

"**Text S2. Yearly variations of CCN activation characteristics.**

Yearly trends of CCN activation properties are investigated. The CCN number concentration ($N_{CCN}$) and hygroscopicity factor calculated from monodisperse CCN measurements ($\kappa_{CCN}$) measured at supersaturation ($SS$) of 0.1% and 0.7% are chosen to represent the CCN activation characteristics. The results are shown in Figure S7. The yearly trends in $N_{CCN}$ and $\kappa_{CCN}$ are not significant (without significant increase or decrease trends) during the measurements from August 2012 to October 2016. However, it is interesting to see that the $N_{CCN}$ measured in 2015 was significantly lower than it measured in other four years. One of the reasons could be that the CCN measurements in 2015 concentrated in summer and autumn, lacking measurements in the spring and winter months (Figure S1). As shown in Figure 3b, $N_{CCN}$ measured at summer and autumn are lower than those measured in spring and winter due to its seasonal trend, causing the lowest median NCCN values in 2015. Thus, the CCN measurements in 2015 may not be representative of the CCN characteristics of the whole year. Similarly, the 2012 measurements may not be representative of year-round CCN characteristics because of the lacking spring and summer measurements. Additionally, it is also hard to see the yearly trends of CCN activation characteristics using the only 4-year data. In order to investigate the yearly trends of CCN activation characteristics, longer-term measurements are required." in lines of 42 to 60 in SI.

[Figure]

Figure S7. Yearly variations of (a) CCN number concentration ($N_{CCN}$) at supersaturation (*SS*) of 0.1% ($N_{CCN,0.1\%}$), (b) $N_{CCN}$ at *SS* of 0.7% ($N_{CCN,0.7\%}$), (c) hygroscopicity factor calculated from monodisperse CCN measurements ($\kappa_{CCN}$) at *SS* of 0.1% ($\kappa_{CCN, 0.1\%}$), and (d) $\kappa_{CCN}$ at *SS* of 0.7% ($\kappa_{CCN, 0.7\%}$). Dots represent the median values. Shaded areas represent the values in the range from 25[th] to 75[th] percent.

[Figure]

Figure S1. Coverage of the effective data represented by the gray columns during the long-term experiment at Melpitz. CCNC — cloud condensation nuclei counter, D-MPSS — Dual-mobility particle size spectrometer, ACSM — aerosol chemical species monitor, MAAP — multi-angle

[Figure]

Figure 3. (a) Relationships between CCN number concentration ($N_{CCN}$) and supersaturation ($SS$), and relationship between activation ratios (AR) and $SS$ for different seasons. (b) Seasonal trends of $N_{CCN}$ and AR at $SS$ = 0.1% and 0.7%. Dots represent the median values of $N_{CCN}$ and AR. Shaded areas represent the values in the range from 25th to 75th percent. Red lines are power-law fittings for $N_{CCN}$ (and AR) vs. $SS$. Two parameters of the fitting results are shown in brackets.

**Comment 2:**

In addition, the authors concluded that "…. the kappa - Dp power-law fit measured at Melpitz could be applied to predict NCCN for other rural regions…." I wonder how did they can derive this, and it would be better to include a CCN closure test or if the authors can present a result by applying the kappa - Dp power-law fit measured at Melpitz to predict the NCCN at other rural sites.

Response: Thanks for your suggestion.

For predicting the NCCN using a parameterized kappa (2nd category in Table 3), the real-time particle number size distribution (PNSD) is an input and its effect on the

prediction bias is not considered. As shown in Figurer 8, the $\kappa$ - $D_p$ relationship measured at Melpitz (black line) is similar to that measured at other rural regions with similar $\kappa$ - $D_p$ power-law fitting results, e.g., the Vavihill station in Sweden (Fors et al., 2011) and the Xinken station in China (Eichler et al., 2008). Thus, we concluded that the kappa - Dp power-law fit measured at Melpitz could be applied to predict NCCN for other rural regions.

Following your suggestion, we conduct a CCN closure test to support the conclusion. However, the data of PNSD and CCN measurements in Vavihill station in Sweden (Fors et al., 2011) and the Xinken station in China (Eichler et al., 2008) are not available. Let's consider it the other way around. If the $\kappa$ - $D_p$ power-law fitting measured at other rural stations can well predict the NCCN at Melpitz, it also helps to support our conclusion. Thus, we applied the $\kappa$ - $D_p$ power-law fitting measured at Vavihill and the Xinken (green and purple lines in Figure 8) to predict the NCCN at Melpitz. The results are as shown in Figure S9. Good prediction results were obtained with mean deviations between the predicted and measured NCCN of ~1%.

We clarify it in the text as follows.

"We conducted a CCN closure test to support this conclusion. Due to lacking the data of PNSD and CCN measurements at Vavihill and Xinken stations, we applied the $\kappa$ - $D_p$ power-law fitting measured at the two rural stations (green and purple lines in Figure 8) to predict the $N_{CCN}$ at Melpitz. Good prediction results were obtained with mean deviations of ~1% (Figure S9 in SI)." Lines 459 to 463.

[Figure]

Xingtai, polluted-suburb (Wang et al., 2018a)
$\kappa = 0.35 * D_p^{0.01}$, $R^2 = 0.01$

Barbados, Coast (Kristensen et al., 2016)
$\kappa = 0.25 * D_p^{0.04}$, $R^2 = 0.02$

Xinzhou, suburb (Chen et al., 2022)
$\kappa = 0.10 * D_p^{0.23}$, $R^2 = 0.96$

Shanghai, urban (Ye et al., 2013)
$\kappa = 0.12 * D_p^{0.19}$, $R^2 = 0.88$

**Melpitz, rural (this study)**
$\boldsymbol{\kappa = 0.052 * D_p^{0.32}}$, $\boldsymbol{R^2 = 0.77}$

Xinken, rural (Eichler et al., 2008)
$\kappa = 0.06 * D_p^{0.31}$, $R^2 = 0.84$

Vavihill, rural (Fors et al., 2011)
$\kappa = 0.07 * D_p^{0.25}$, $R^2 = 0.87$

Paris, suburb (Mazoyer et al., 2019)
$\kappa = 0.003 * D_p^{0.86}$, $R^2 = 0.92$

Amazon, rainforest (Pohlker et al., 2016)
$\kappa = 0.02 * D_p^{0.49}$, $R^2 = 0.87$

Guangzhou, urban (Chen et al., 2022)
$\kappa = 0.04 * D_p^{0.31}$, $R^2 = 0.74$

Budapest, urban (Salma et al., 2021)
$\kappa = 3.4 * 10^{-5} * D_p^{1.57}$, $R^2 = 0.96$

Figure 8. Relationships between the particle hygroscopicity factor ($\kappa$) and diameter ($D_p$) observed at different stations. Lines are power-law fits of $\kappa$ vs. $D_p$.

[Figure]

Figure S9. Predicted vs. measured CCN number concentration ($N_{CCN}$) at Melpitz. (a) using the $\kappa$-$D_p$ power-law fitting measured at Xinken station in China (Eichler et al., 2008) to predict the Melpitz

$N_{CCN}$; (b) using the $\kappa$ - $D_p$ power-law fitting measured at Vavihill station in Sweden (Fors et al., 2011). Dashed line is the 1:1 line and solid line is the linear fitting.

**Comment 3:**

In Figure 8, some important references relevant to aerosols hygroscopicity dependence on Dp may be missing (Chen et al., 2022, ACP, 22, 6773–6786, https://doi.org/10.5194/acp-22-6773-2022, 2022). Adding these necessary discussions constitutes major revisions. I recommend publication after these are done, and the remaining issues listed below have been addressed.

Response: Thanks for your suggestion. The $\kappa$ vs. $D_p$ relationships measured at Guangzhou and Xinzhou stations (Chen et al., 2022) are added in Figure 8, which are in wine red and dark green. We clarify it in the text as follows.

"However, it may cause considerable deviations for different aerosol background regions, e.g., the suburb stations in Xingtai, China (Wang et al., 2018a), Xinzhou, China (Chen et al., 2022), and Paris, France (Mazoyer et al., 2019), the coast of Barbados (Kristensen et al., 2016), the amazon rainforest (Pöhlker et al., 2016), and the urban stations in Budapest, Hungary (Salma et al., 2021), Guangzhou, China (Chen et al., 2022), and Shanghai, China (Ye et al., 2013), because their $\kappa$ - $D_p$ relationships are different from that measured at Melpitz." Lines 459 to 465.

[Figure]

Figure 8. Relationships between the particle hygroscopicity factor ($\kappa$) and diameter ($D_p$) observed at different stations. Lines are power-law fits of $\kappa$ vs. $D_p$.

Chen, L., Zhang, F., Zhang, D., Wang, X., Song, W., Liu, J., Ren, J., Jiang, S., Li, X., and Li, Z.: Measurement report: Hygroscopic growth of ambient fine particles measured at five sites in China, Atmos. Chem. Phys., 22, 6773–6786, https://doi.org/10.5194/acp-22-6773-2022, 2022.

**Comment 4:**

The authors need to present the key findings in the abstract, so as to highlight their main points.

Response: Thanks for your comment. The following four points are what we would like to highlight in abstract:

1) We provide the overall CCN activation characteristics at Melpitz.

2) Aerosol particle activation is highly variable across seasons.

3) Both $\kappa$ and the mixing state are size dependent.

4) Size-resolved $\kappa$ improves the $N_{CCN}$ prediction.

In Abstract, to help readers easily see the four points, we highlight these four points by adding the serial number before each point, as follows.

"Understanding aerosol particle activation is essential for evaluating aerosol indirect effects (AIEs) on climate. Long-term measurements of aerosol particle activation help to understand the AIEs and narrow down the uncertainties of AIEs simulation. However, they are still scarce. In this study, more than 4-year aerosol comprehensive measurements were utilized at the central European research station Melpitz, Germany, to gain insight into the aerosol particle activation and provide recommendations on improving the prediction of number concentration of cloud condensation nuclei (CCN, $N_{CCN}$). (1) The overall CCN activation characteristics at Melpitz is provided. As supersaturation (*SS*) increases from 0.1% to 0.7%, the median $N_{CCN}$ increases from 399 to 2144 cm$^{-3}$, which represents 10% to 48% of the total particle number concentration with a diameter range of 10 – 800 nm, while the median hygroscopicity factor ($\kappa$) and critical diameter ($D_c$) decrease from 0.27 to 0.19 and from 176 to 54 nm, respectively. (2) Aerosol particle activation is highly variable across seasons, especially at low *SS* conditions. At *SS* = 0.1%, the median $N_{CCN}$ and activation ratio (AR) in winter are 1.6 and 2.3 times higher than the summer values, respectively. (3) Both $\kappa$ and the mixing state are size dependent. As the particle diameter ($D_p$) increases, $\kappa$ increases at $D_p$ of ~40 to 100 nm and almost stays constant at $D_p$ of 100 to 200 nm, whereas the degree of the external mixture keeps decreasing at $D_p$ of ~40 to 200 nm. The relationships of $\kappa$ vs. $D_p$ and degree of mixing vs. $D_p$ were both fitted well by a power-law function. (4) Size-resolved $\kappa$ improves the $N_{CCN}$ prediction. We recommend applying the $\kappa$ - $D_p$ power-law fit for $N_{CCN}$ prediction at Melpitz, which performs better than using the

constant $\kappa$ of 0.3 and the $\kappa$ derived from particle chemical compositions and much better than using the $N_{CCN}$ (AR) vs. $SS$ relationships. The $\kappa$ - $D_p$ power-law fit measured at Melpitz could be applied to predict $N_{CCN}$ for other rural regions. For the purpose of improving the prediction of $N_{CCN}$, long-term monodisperse CCN measurements are still needed to obtain the $\kappa$ - $D_p$ relationships for different regions and their seasonal variations."